# CHEMISTRY-INSPIRED DIFFUSION WITH NON-DIFFERENTIABLE GUIDANCE

**Yuchen Shen**[*], **Chenhao Zhang**[*], **Sijie Fu**[*], **Chenghui Zhou, Newell Washburn**[†]**, Barnabás Póczos**[†]
Carnegie Mellon University
Pittsburgh, PA 15213, USA
`{chenhao4, sijief, chenghuz, washburn}@andrew.cmu.edu`
`{yuchens3, bapoczos}@cs.cmu.edu`

## ABSTRACT

Recent advances in diffusion models have shown remarkable potential in the conditional generation of novel molecules. These models can be guided in two ways: (i) explicitly, through additional features representing the condition, or (ii) implicitly, using a property predictor. However, training property predictors or conditional diffusion models requires an abundance of labeled data and is inherently challenging in real-world applications. We propose a novel approach that attenuates the limitations of acquiring large labeled datasets by leveraging domain knowledge from quantum chemistry as a non-differentiable oracle to guide an unconditional diffusion model. Instead of relying on neural networks, the oracle provides accurate guidance in the form of estimated gradients, allowing the diffusion process to sample from a conditional distribution specified by quantum chemistry. We show that this results in more precise conditional generation of novel and stable molecular structures. Our experiments demonstrate that our method: (1) significantly reduces atomic forces, enhancing the validity of generated molecules when used for stability optimization; (2) is compatible with both explicit and implicit guidance in diffusion models, enabling joint optimization of molecular properties and stability; and (3) generalizes effectively to molecular optimization tasks beyond stability optimization.

⭘ https://github.com/A-Chicharito-S/ChemGuide

## 1 INTRODUCTION

Diffusion models have received increasing attention in molecular design. Their ability to generate novel molecules with desired properties (i.e., guided diffusion) has fostered advances in material science (Manica et al., 2023; Yang et al., 2023), chemistry (Anstine & Isayev, 2023), protein design (Watson et al., 2023; Abramson et al., 2024), etc. To achieve guided diffusion, one can *explicitly* condition the diffusion process on specific properties in training (Hoogeboom et al., 2022; Xu et al., 2023), such that the model is naturally a conditional generator during inference. Alternatively, an unconditional model can be trained without labels, and the diffusion process is guided *implicitly* at inference time (Dhariwal & Nichol, 2021; Vignac et al., 2022) using a property predictor that provides guidance gradients to steer the model toward sampling from the conditional distribution.

Both explicitly and implicitly guided diffusion require a labeled dataset to train the model or the property predictor. The acquisition of labels, however, can be expensive, time-consuming, and often impracticable[1]. When only small labeled datasets are available (Power et al., 2022), the model and the property predictor can potentially struggle to generalize beyond seen structures (see Appendix H.4), degrading the performance of the diffusion process when generating novel molecules. Implicitly guided diffusion can help address such challenges. This approach (i) requires no labels to train the model and (ii) can replace the property predictor with domain knowledge from quantum chemistry, fulfilled by quantum chemistry software such as xTB (Bannwarth et al., 2019; 2021) and Gaussian (Frisch et al., 2016). Such software can act as an expert oracle to create labeled datasets; thereby, avoiding the extrapolation shortcomings of neural networks.

Despite following certain computation procedures, quantum chemistry is by nature a non-differentiable oracle that can not be backpropagated with neural networks, as those procedures

---

[*]Equal contribution.     [†]Correspondence to: Barnabás Póczos and Newell Washburn.

[1]For example, $LD_{50}$ (Erhirhie et al., 2018) measures the lethal dose of a test substance, which is difficult to calculate from the molecular structure of the substance alone; thus, requiring animal testing.

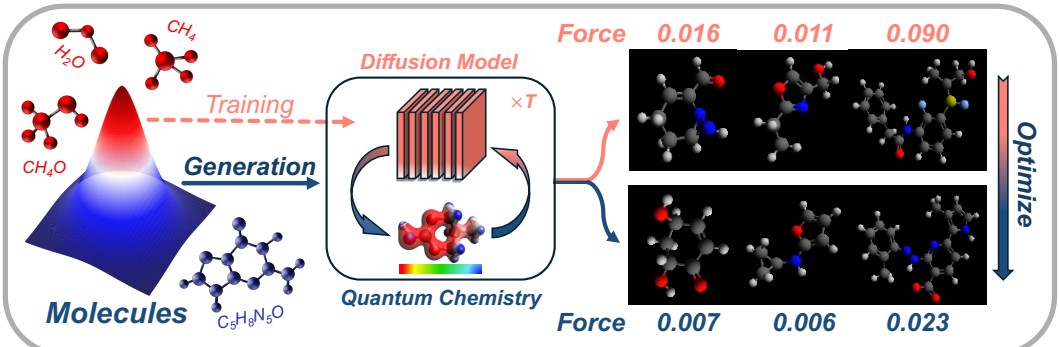

Figure 1: The overview of CHEMGUIDE. On the left, we present the space of all molecules (roughly) as a unimodal distribution, where red/blue region indicates molecules for training/novel molecules generated by the diffusion model. In the middle, CHEMGUIDE derives non-differentiable guidance from quantum chemistry to steer the diffusion process towards a conditional distribution (e.g., minimized forces). On the right, we present the average forces of 3 sets of molecules generated by GeoLDM (Xu et al., 2023) trained on *QM9* (Ramakrishnan et al.) (left two) and *GEOM* (Axelrod & Gómez-Bombarelli) (rightmost one) without (above) and with (below) CHEMGUIDE.

usually involve proprietary algorithms or extremely computationally expensive to perform, which prevents the integration of quantum chemistry to guide the diffusion models. In this work, we aim to *implicitly* guide the diffusion process with a non-differentiable oracle, and employ zeroth-order optimization methods (Nesterov & Spokoiny, 2017; Malladi et al., 2023) to estimate the necessary guidance gradients. We refer to our method as CHEMGUIDE, a method that leverages quantum chemistry to provide more accurate diffusion guidance. Specifically, given the oracle xTB with the GFN2-xTB method (Bannwarth et al., 2019) that conducts quantum chemistry calculation of potential energies and atomic forces, we first use CHEMGUIDE to improve the stability of the generated molecular geometry (see Fig. 1), by guiding the diffusion process towards sampling from a distribution where the net force on the atoms is zero. Further, we formulate diffusion guidance on both molecular property and stability as a bilevel optimization (Colson et al., 2007) problem, to demonstrate that CHEMGUIDE is compatible with both *explicitly* and *implicitly* guided diffusion. We summarize our contributions as follows:

- We propose CHEMGUIDE, which leverages zeroth-order optimization techniques to introduce a non-differentiable chemistry oracle as diffusion guidance (Section 3).
- We show in a bilevel optimization framework (Algorithm 2) that CHEMGUIDE can be utilized in conjunction with existing guided diffusion methods.
- Experiments and analyses (Section 4 and 5) demonstrate that CHEMGUIDE (i) improves the stability of generated molecules when guided with domain knowledge from the semi-empirical quantum chemical method, GFN2-xTB (Section 4.2); (ii) generalizes to molecular optimization other than stability (Section 4.5); (iii) effectively improves both molecular property and stability when used with other guided diffusion methods (Section 4.4).

## 2 PRELIMINARY

In this section, we introduce the key concepts of diffusion models and explore the architecture of a specific diffusion model designed for 3D molecule generation. We will explain how to achieve equivariance in the generated molecules and how the semi-empirical quantum chemical method GFN2-xTB can be utilized for guidance. Additionally, we will also outline our motivation for integrating CHEMGUIDE with 3D diffusion models to optimize molecular stability.

**Diffusion Models** In general, a diffusion model (Sohl-Dickstein et al., 2015; Ho et al., 2020; Song et al., 2020) consists of a forward *diffusion process* and a reverse *denoising process*. The diffusion process is a Markov chain that gradually adds Gaussian noises with a predefined variance schedule $\beta_{1:T}$ from timestep 1 to $T$ to the original datapoint $\boldsymbol{x}_0$, which is chosen such that $\boldsymbol{x}_T \sim \mathcal{N}(\mathbf{0}, \mathbf{I})$. The forward diffusion process $q$ is usually defined as a fixed schedule by the following:

$$q(\boldsymbol{x}_{1:T} \mid \boldsymbol{x}_0) = \prod_{t=1}^{T} q(\boldsymbol{x}_t \mid \boldsymbol{x}_{t-1}) \qquad q(\boldsymbol{x}_t \mid \boldsymbol{x}_{t-1}) = \mathcal{N}(\boldsymbol{x}_t; \sqrt{1-\beta_t}\boldsymbol{x}_{t-1}, \beta_t \mathbf{I}) \qquad (1)$$

The denoising process starts with $\boldsymbol{x}_T$ and recovers the original datapoint $\boldsymbol{x}_0$ by predicting the mean of $\boldsymbol{x}_{t-1}$ given $\boldsymbol{x}_t$ using a neural network denoted as $\mu_\theta(\boldsymbol{x}_t, t)$, where $\theta$ is the parameter.

$$p_\theta(\boldsymbol{x}_{0:T}) = p(\boldsymbol{x}_T) \prod_{t=1}^{T} p_\theta(\boldsymbol{x}_{t-1} \mid \boldsymbol{x}_t) \qquad p_\theta(\boldsymbol{x}_{t-1} \mid \boldsymbol{x}_t) = \mathcal{N}(\boldsymbol{x}_{t-1}; \mu_\theta(\boldsymbol{x}_t, t), \Sigma_\theta(\boldsymbol{x}_t, t)) \quad (2)$$

In practice, $\Sigma_\theta = \sigma_t^2 \mathbf{I}$ with $\sigma_t = \sqrt{\frac{1-\alpha_{t-1}^2}{1-\alpha_t^2}\beta_t}$, $\alpha_t = \sqrt{\prod_{i=1}^{t}(1 - \beta_i)}$ for all $t$. For simplicity, Ho et al. (2020) reparametrize Eq. 1 such that $\boldsymbol{x}_t = \alpha_t \boldsymbol{x}_0 + \sqrt{1 - \alpha_t^2}\epsilon$, where $\epsilon \sim N(\mathbf{0}, \mathbf{I})$. Hence, instead of predicting $\mu_\theta(\boldsymbol{x}_t, t)$, we now predict the noise $\epsilon_\theta(\boldsymbol{x}_t, t)$, where $\mu_\theta(\boldsymbol{x}_t, t)$ is parameterized as $\frac{1}{\sqrt{1-\beta_t}}(\boldsymbol{x}_t - \frac{\beta_t}{\sqrt{1-\alpha_t^2}}\epsilon_\theta(\boldsymbol{x}_t, t))$. Consequently, we have

$$\boldsymbol{x}_{t-1} \sim \mathcal{N}(\frac{1}{\sqrt{1-\beta_t}}(\boldsymbol{x}_t - \frac{\beta_t}{\sqrt{1-\alpha_t^2}}\epsilon_\theta(\boldsymbol{x}_t, t)), \sigma_t^2 \mathbf{I}) \qquad (3)$$

**Latent Diffusion Architecture for 3D Molecule Generation**  An $N$-atom molecule can be represented as a point cloud $\mathcal{G} = [\mathbf{x}, \mathbf{h}] \in \mathbb{R}^{N \times (3+d)}$, with $\mathbf{x} \in \mathbb{R}^{N \times 3}$ being the $N$ atom's 3D coordinates and $\mathbf{h} \in \mathbb{R}^{N \times d}$ as the atom features indicating atom types. A latent diffusion architecture (Rombach et al., 2022; Xu et al., 2023) consists of a Variational Autoencoder (VAE) and a diffusion model trained consecutively. Particularly, the geometric latent diffusion model (Xu et al., 2023) uses the encoder of the VAE to project molecules to a continuous latent space, on which the diffusion model is then trained. Denote the encoder as $\mathcal{E}$ and the latent variable by $\mathbf{z} \in \mathbb{R}^{N \times (3+d_z)}$, then $[\mathbf{z}_{\mathbf{x},0}, \mathbf{z}_{\mathbf{h},0}] = \mathcal{E}([\mathbf{x}, \mathbf{h}])$, with $\mathbf{z}_{\mathbf{h},0} \in \mathbb{R}^{N \times d_z}$ and $d_z < d$. Let $\mathbf{z}_t = [\mathbf{z}_{\mathbf{x},t}, \mathbf{z}_{\mathbf{h},t}]$, the latent forward diffusion process and reverse denoising process are defined by replacing $\boldsymbol{x}$ with $\boldsymbol{z}$ in Eq. 1 and 2. We denote the decoder of the VAE as $\mathcal{D}$, which maps $\mathbf{z}_0$ back to the original molecular space, such that $\mathcal{D}([\mathbf{z}_{\mathbf{x},0}, \mathbf{z}_{\mathbf{h},0}]) = [\mathbf{x}, \mathbf{h}] \in \mathbb{R}^{N \times (3+d)}$. The encoder $\mathcal{E}$ and the decoder $\mathcal{D}$ are parameterized with an equivariant graph neural network (EGNN) (Satorras et al., 2021) to translate between discrete molecular data and latent variables, such that the atom types are invariant and the positions are equivariant to transformations as the following:

$$R\mathbf{z}_{\mathbf{x},t} + T, \mathbf{z}_{\mathbf{h},t} = \mathcal{E}(R\mathbf{x}_t + T, \mathbf{h}_t) \qquad R\mathbf{x}_t + T, \mathbf{h}_t = \mathcal{D}(R\mathbf{z}_{\mathbf{x},t} + T, \mathbf{z}_{\mathbf{h},t}) \qquad (4)$$

for any rotation matrix $R$ and translation matrix $T$, where $\mathbf{z}_{\mathbf{x},t} \in \mathbb{R}^{N \times 3}$ are required to satisfy zero center gravity and have zero-mean over $N$ atoms for each position. In addition, the latent diffusion model is also parameterized by EGNN such that transitions between each timestep in the denoising process also respect the same characteristics.

**GFN2-xTB Method**  Matter such as electrons, atoms, and molecules, interacts with matter inherently to reach configurations with lower potential energies, i.e. stability. To formulate this as a molecular geometry optimization problem, let $h_1, \cdots, h_N$ be the $N$ atoms of a given molecule and $\mathbf{x_1}, \cdots, \mathbf{x_N} \in \mathbb{R}^3$ be their coordinates in the 3D space. Each atom $h_i$ is subject to a force:

$$\mathbf{f_i}(\mathcal{G}) = \frac{\partial E_p(\mathbf{x_1}, \cdots, \mathbf{x_N} \mid h_1, \cdots, h_N)}{\partial \mathbf{x_i}}, \forall i \in [N] \qquad (5)$$

where $E_p$ represents the potential energy of the conformation. The force here manifests valid physical interpretations: at a high level, an atom is pushed accordingly by the exerted force until the force reduces to zero and an equilibrium is achieved. The two necessary (but not sufficient) conditions for a stable molecular geometry are: (i) all forces on the atoms should be (close to) zero, i.e., $\forall i \in [N], \mathbf{f_i}(\mathcal{G}) = \mathbf{0}$; and (ii) the Hessian matrix, i.e. the second derivative of the potential energy, must be non-negative.

However, the exact mathematical potential energy evaluation, i.e., the solution to the Schrödinger equation, is still a "black box" (Cao et al.). Over the years, different levels of theories and methods have been developed to evaluate potential energies, such as force field (FF), semi-empirical methods (e.g., xTB), and density functional theory (DFT) methods (e.g., B3LYP/6-31G(2df,p)), listed in an order of increased accuracy and cost. After trading-off between accuracy and efficiency within a feasible computation cost, we selected GFN2-xTB, a more recent and advanced semi-empirical method (Bannwarth et al., 2019), to calculate the forces of the generated molecular geometry in the diffusion process. More details about GFN2-xTB can be found in Appendix D.

**Motivation for Stability Optimization with 3D Diffusion**  Stable molecular geometry is favored for molecule generation. A novel but unstable molecular geometry may not be physically achieved at all and thus carries no substantial meaning. To optimize stability, one can minimize the potential energy of the molecule by iteratively minimizing the forces on the atoms (Eq. 5), motivating our choice of a 3D diffusion model. More details and discussions are provided in Appendix E.

**Noisy Guidance**    The goal of neural guidance is to direct the denoising process towards a target property value $y$. Dhariwal & Nichol (2021) propose to modify the denoising process to achieve conditional generation with an unconditional model and using a scalar $s$ that controls the guidance strength:

$$\boldsymbol{x}_{t-1} \sim \mathcal{N}(\frac{1}{\sqrt{1-\beta_t}}(\boldsymbol{x}_t - \frac{\beta_t}{\sqrt{1-\alpha_t^2}}\epsilon_\theta(\boldsymbol{x}_t, t)) + s\sigma_t^2 \nabla_{\boldsymbol{x}_t} \log p_\phi(y|\boldsymbol{x}_t), \sigma_t^2 \mathbf{I}) \tag{6}$$

Here $p_\phi(y|\boldsymbol{x}_t)$ is parameterized by a classifier, and $y$ is a categorical label, such that the modification $s\sigma_t^2 \nabla_{\boldsymbol{x}_t} \log p_\phi(y|\boldsymbol{x}_t)$ shifts the mean of the sampling distribution to provide guidance. Let $y \in \mathbb{R}$ and $f_\eta : \mathcal{G} \to \mathbb{R}$ be the neural regressor for the molecular property of interest (Vignac et al., 2022). Now, assuming $y|\boldsymbol{x}_t \sim \mathcal{N}(f_\eta(\boldsymbol{x}_t), \sigma_\eta^2 \mathbf{I})$ and $\sigma_\eta^2 = 1$, we have:

$$\nabla_{\boldsymbol{x}_t} \log p_\phi(y \mid \boldsymbol{x}_t) \propto -\nabla_{\boldsymbol{x}_t} \|y - f_\eta(\boldsymbol{x}_t)\|_2^2 = -\nabla_{\boldsymbol{x}_t} \mathcal{L}(y, f_\eta(\boldsymbol{x}_t)) \tag{7}$$

where $\mathcal{L}(y, f_\eta(\boldsymbol{x}_t))$ is the Mean Square Error (MSE) between the target and the prediction of $f_\eta(\cdot)$.

However, in the early stage of the denoising process, $\boldsymbol{x}_t = \alpha_t \boldsymbol{x}_0 + \sqrt{1-\alpha_t^2}\epsilon$ might not be informative enough to predict $y$ as it consists mostly of Gaussian noise. For effective prediction during the denoising process, we estimate the denoised version of $\boldsymbol{x}_t$ as Kawar et al. (2022); Song et al. (2020):

$$t_0(\boldsymbol{x}_t) := \hat{\boldsymbol{x}}_0 = \frac{\boldsymbol{x}_t - \sqrt{1-\alpha_t^2}\epsilon_\theta(\boldsymbol{x}_t, t)}{\alpha_t} \tag{8}$$

where $f_\eta(\hat{\boldsymbol{x}}_0)$ is used in place of $f_\eta(\boldsymbol{x}_t)$ as the predicted molecular property.

**Clean Guidance**    Besides applying the gradient as guidance on noisy $\boldsymbol{x}_t$, we build insight from Bansal et al. (2023) and derive guidance in the clean (=noise free) space $\hat{\boldsymbol{x}}_0$ as:

$$\Delta\boldsymbol{x}_0 = \arg\min_\Delta \mathcal{L}(y, f_\eta(\hat{\boldsymbol{x}}_0 + \Delta)) \tag{9}$$

where $\Delta\boldsymbol{x}_0$ is approximated using $K$ steps of gradient descent starting from $\Delta = 0$. Note that $\Delta\boldsymbol{x}_0$ is in clean data space, so we need to translate it back to the noisy space while recovering $\boldsymbol{x}_t$:

$$\boldsymbol{x}_t = \alpha_t(\hat{\boldsymbol{x}}_0 + \Delta\boldsymbol{x}_0) + \sqrt{1-\alpha_t^2}\tilde{\epsilon} \tag{10}$$

where $\tilde{\epsilon}$ is the augmented noise used to to sample $\boldsymbol{x}_{t-1}$ and is thus given by:

$$\tilde{\epsilon} = \epsilon_\theta(\boldsymbol{x}_t, t) - \frac{\alpha_t}{\sqrt{1-\alpha_t^2}}\Delta\boldsymbol{x}_0 \tag{11}$$

## 3    METHODOLOGY

### 3.1    GUIDANCE FROM A NON-DIFFERENTIABLE ORACLE

We aim to tackle a challenging problem where the guidance is specified by a non-differentiable oracle (e.g., the GFN2-xTB method). We change our notation from $\boldsymbol{x}_t$ to $\boldsymbol{z}_t$ at each time step to indicate diffusion in the latent space. With the guidance target $y$ being 0, our non-differentiable function $\mathbf{f}$ (Eq. 5) for force guidance is defined as follows.

$$\mathbf{f}(\mathcal{G}) = \frac{1}{N}\sum_i^N \mathbf{f_i}(\mathcal{G}) \tag{12}$$

where the molecular graph is decoded as $\mathcal{G} = \mathcal{D}(\hat{\boldsymbol{z}}_0)$ with $\hat{\boldsymbol{z}}_0 = t_0(\boldsymbol{z}_t)$ estimated by Eq. 8, and we aim to achieve $\mathbf{f}(\mathcal{G}_0') < \mathbf{f}(\mathcal{G}_0)$. Here $\mathcal{G}_0', \mathcal{G}_0$ are provided by CHEMGUIDE and an unguided diffusion model. As $\mathbf{f}$ is non-differentiable across different molecules, we can not directly add guidance using Eq. 7. Instead, we estimate the gradient analytically. Recall that $\mathcal{L}(y, \mathbf{f}(\mathcal{G}))$ is the MSE loss, $\mathbf{f} : \mathcal{G} \to \mathbb{R}$ is the non-differentiable oracle, and $\mathcal{D}$ is the decoder. Let $\mathcal{F}$ be the composition $\mathbf{f} \circ \mathcal{D} \circ t_0$, and we estimate the gradient as:

$$\hat{\nabla}_{\boldsymbol{z}_t} \log p_\phi(y|\boldsymbol{z}_t) \propto -\nabla_{\mathcal{F}(\boldsymbol{z}_t)} \mathcal{L}(y, \mathcal{F}(\boldsymbol{z}_t)) \nabla_{\boldsymbol{z}_t} \mathcal{F}(\boldsymbol{z}_t) \tag{13}$$

$$\approx -\nabla_{\mathcal{F}(\boldsymbol{z}_t)} \mathcal{L}(y, \mathcal{F}(\boldsymbol{z}_t)) \lim_{\zeta \to 0} \frac{\mathcal{F}([\boldsymbol{z}_{\mathbf{x},t} + \zeta\mathbf{1}_{N\times3}, \boldsymbol{z}_{\mathbf{h},t}]) - \mathcal{F}([\boldsymbol{z}_{\mathbf{x},t} - \zeta\mathbf{1}_{N\times3}, \boldsymbol{z}_{\mathbf{h},t}])}{2\zeta} \tag{14}$$

---

**Algorithm 1** Bilevel guided diffusion sampling with *noisy neural guidance*

---

**Input**: a latent diffusion model $\epsilon_\theta$, a VAE decoder $\mathcal{D}$, a composition function $\mathcal{F}$, target property score $y$, oracle guidance scale $s_o$, regressor guidance scale $s_r$, SPSA perturbation $\zeta$.
**Output**: optimized molecule $[\mathbf{x}, \mathbf{h}]$
$\boldsymbol{z}_T \leftarrow \mathcal{N}(0, \mathbf{I})$
**for all** $t$ from $T$ to $1$ **do**

$\quad \mu_{t-1}, \Sigma_{t-1} \leftarrow \frac{1}{\sqrt{1-\beta_t}}(\boldsymbol{z}_t - \frac{\beta_t}{\sqrt{1-\alpha_t^2}}\epsilon_\theta(\boldsymbol{z}_t, t)), \sigma_t^2 \mathbf{I}$ ⠀⠀⠀⠀⠀⠀⠀⠀⠀⠀⠀⠀(Eq. 3)

$\quad g_{t-1,\text{regressor}} \propto -\nabla_{\boldsymbol{z}_t}\|y - f_\eta(\mathcal{D} \circ t_0(\boldsymbol{z}_t))\|^2$ ⠀⠀⠀⠀⠀⠀⠀⠀⠀⠀⠀⠀⠀(Eq. 7)
$\quad \boldsymbol{z}'_{t-1} \leftarrow \mathcal{N}(\mu_{t-1} + s_r \Sigma_{t-1} g_{t-1,\text{regressor}}, \Sigma_{t-1})$ ⠀⠀⠀⠀⠀⠀⠀⠀⠀⠀(Eq. 6)

$\quad \boldsymbol{z}'_t = \frac{\alpha_t}{\alpha_{t-1}}\boldsymbol{z}'_{t-1} + \sqrt{1 - \frac{\alpha_t^2}{\alpha_{t-1}^2}}\epsilon'$, such that $\epsilon' \sim \mathcal{N}(0, \mathbf{I})$ ⠀⠀⠀// project from $t-1$ back to $t$

$\quad \mathbf{U} \leftarrow \mathcal{N}(0, \mathbf{I})$

$\quad g_{t-1} \propto -\nabla_{\mathcal{F}(\boldsymbol{z}'_t)}\|\mathcal{F}(\boldsymbol{z}'_t)\|^2 \cdot \frac{1}{2\zeta}\left(\mathcal{F}([\boldsymbol{z}'_{\mathbf{x},t} + \zeta\mathbf{U}, \boldsymbol{z}'_{\mathbf{h},t}]) - \mathcal{F}([\boldsymbol{z}'_{\mathbf{x},t} - \zeta\mathbf{U}, \boldsymbol{z}'_{\mathbf{h},t}])\right)\mathbf{U}$ ⠀⠀(Eq. 15)

$\quad \mu'_{t-1}, \Sigma'_{t-1} \leftarrow \frac{1}{\sqrt{1-\beta_t}}(\boldsymbol{z}'_t - \frac{\beta_t}{\sqrt{1-\alpha_t^2}}\epsilon_\theta(\boldsymbol{z}'_t, t)), \sigma_t^2 \mathbf{I}$ ⠀⠀⠀⠀⠀⠀⠀⠀(Eq. 3)

$\quad \boldsymbol{z}_{t-1} \leftarrow \mathcal{N}(\mu'_{t-1} + s \cdot \sigma_{t-1}^2 g_{t-1}, \Sigma'_{t-1})$ ⠀⠀⠀⠀⠀⠀⠀⠀⠀⠀⠀⠀⠀(Eq. 6)
**end for**
$[\mathbf{x}, \mathbf{h}] \leftarrow \mathcal{D}(\boldsymbol{z}_0)$
**return** $[\mathbf{x}, \mathbf{h}]$

---

This approximation is possible because $\boldsymbol{z}_t$ is continuous and it eliminates the need to train neural regressors for properties such as forces, which comes with approximation error (Gasteiger et al., 2020). In addition, these regressors are often trained on ground state geometries (optimized stable geometries) instead of non-optimal geometries, which are the most common geometries during the denoising process, making them dubious options for property guidance. However, directly adding (or subtracting) $\zeta \mathbf{1}_{N\times3}$ to $\boldsymbol{z}_{\mathbf{x},t}$ would break the equivariance requirement in Eq. 4 on the latent variables, as it shifts the mean of the coordinates by $\zeta$. To maintain zero central gravity (Xu et al., 2023) of input to $\mathcal{F}(\cdot)$, we construct a perturbation matrix $\mathbf{U} \in \mathbb{R}^{N\times3} \sim \mathcal{N}(0, 1)$, and apply Simultaneous Perturbation Stochastic Approximation (SPSA) (Spall, 1992; Nesterov & Spokoiny, 2017) to estimate the gradient, then Eq. 14 becomes:

$$\hat{\nabla}_{\boldsymbol{z}_t} \log p_\phi(y|\boldsymbol{z}_t) \propto -\nabla_{\mathcal{F}(\boldsymbol{z}_t)}\mathcal{L}(y, \mathcal{F}(\boldsymbol{z}_t))\frac{\mathcal{F}([\boldsymbol{z}_{\mathbf{x},t} + \zeta\mathbf{U}, \boldsymbol{z}_{\mathbf{h},t}]) - \mathcal{F}([\boldsymbol{z}_{\mathbf{x},t} - \zeta\mathbf{U}, \boldsymbol{z}_{\mathbf{h},t}])}{2\zeta}\mathbf{U} \quad (15)$$

where $\zeta$ is a small perturbation scale (e.g., $10^{-6}$). The perturbed representations $[\boldsymbol{z}_{\mathbf{x},t} \pm \zeta\mathbf{U}, \boldsymbol{z}_{\mathbf{h},t}]$ obey zero central gravity required by $t_0$, as $\mathbb{E}[\zeta\mathbf{U}] = 0$. Note that, unlike neural regressor guidance, we only add guidance to the positions (i.e. $\boldsymbol{z}_{\mathbf{x},t}$) and apply no gradient to the atom types (i.e. $\boldsymbol{z}_{\mathbf{h},t}$). We do this because the force definition (Eq. 5) is only physically grounded when the set of atoms stays constant (i.e. no matter/mass is created from or reduced to void). In other words, since energy is a relative concept, any gradient that arises from atom-type change has no physical interpretation. Further discussions on the choice of SPSA for gradient estimation can be found in Appendix F.

### 3.2 COMBINE GUIDANCE FROM NEURAL NETWORK AND NON-DIFFERENTIABLE ORACLE

Apart from guided generation conditioned on one single property (e.g., the norm of static polarizability $\alpha$) with a regressor or a non-differentiable oracle (e.g., GFN2-xTB), we can perform joint optimization with the guidance specified by both the differentiable regressor and the non-differentiable oracle, which is encapsulated under the *bilevel optimization* framework. Formally, recall that $f_\eta$ represents the regressor and $\mathcal{F} = \mathbf{f} \circ \mathcal{D} \circ t_0$ is the composition of the non-differentiable oracle $\mathbf{f}$, the VAE decoder $\mathcal{D}$, the latent diffusion model, respectively, the joint optimization is formulated as:

$$\min_{\boldsymbol{z}_{x,t}} \quad \mathcal{F}([\boldsymbol{z}_{x,t}, \boldsymbol{z}^*_{h,t}]) \tag{16}$$

$$\text{s.t.} \quad [\boldsymbol{z}^*_{x,t}, \mathbf{z}^*_{h,t}] = \underset{[\boldsymbol{z}_{x,t}, \boldsymbol{z}_{h,t}]}{\arg\min} \|f_\eta([\boldsymbol{z}_{x,t}, \boldsymbol{z}_{h,t}]) - y\|^2 \tag{17}$$

where $y$ is the target property value (e.g., we hope the static polarizability $\alpha \to y$ after optimization). As detailed in Algorithm 2, we first optimize molecular property with $f_\eta$, and obtain the optimal atom positions $\boldsymbol{z}^*_{x,t}$ and features $\boldsymbol{z}^*_{h,t}$ (Eq. 17); we then optimize stability with $\mathcal{F}$ by fixing $\mathbf{z}^*_{h,t}$ and initializing $\boldsymbol{z}_{x,t}$ (Eq. 16) as $\boldsymbol{z}^*_{x,t}$. The chemistry intuition of our design is detailed in Appendix G.

## 4 EXPERIMENTS

### 4.1 EXPERIMENT SETTING

**Dataset** The models in our experiment are trained on the QM9 dataset (Ramakrishnan et al.) and the GEOM dataset (Axelrod & Gómez-Bombarelli). The QM9 dataset is a catalog with 134K small drug-like molecules consisting of up to nine heavy (non-hydrogen) atoms. The Geometric Ensemble Of Molecules (GEOM) dataset includes 450K molecules with up to 91 heavy atoms (on average, 24.9), where 37 million molecular conformations are generated and reported with their geometries, energies, and statistical weight.

**Guidance Property** We study guided generation for force and thus energy optimization on QM9 and GEOM. For neural and combined guidance, we evaluate on QM9 as there are no labels in GEOM to train the regressors. We consider the following 6 properties reported in QM9: the norm of static polarizability ($\alpha$, Bohr$^3$), the norm of dipole moment ($\mu$, Debye), heat capacity at room temperature ($C_v$, cal/(mol·K)), the energy of the electron in the highest occupied molecular orbital ($\epsilon_{\text{HOMO}}$, eV), the energy of the electron in the lowest unoccupied molecular orbital ($\epsilon_{\text{LUMO}}$, eV), and the HOMO-LUMO energy gap ($\Delta\epsilon$, eV). We choose $s$ (Eq. 6) from $\{1, 10^{-1}, 10^{-2}, 10^{-3}, 10^{-4}\}$ for all experiments, and additionally $\{2, 5, 10, 20, 25, 30, 40, 50\}$ for the 6 properties.

**Model** We integrate CHEMGUIDE with GeoLDM (Xu et al., 2023), an improvement from EDM (Hoogeboom et al., 2022). For non-differentiable guidance on force, we compare our method to unconditional EDM and GeoLDM[2], since there is no available ground-truth force to explicitly train a conditional model. For noisy neural guidance and combined guidance, we choose conditionally trained EDM (C-EDM) and GeoLDM (C-GeoLDM) as the baselines.

**Evaluation Metric** For non-differentiable guidance on the force, we use the Root Mean Square (RMS) of the forces calculated at GFN2-xTB as the evaluation metric. We also report validity, uniqueness, atom stability, molecule stability, and energy above the ground state. For differentiable neural guidance on the 6 properties ($\alpha, \mu, C_v, \epsilon_{\text{HOMO}}, \epsilon_{\text{LUMO}}, \Delta\epsilon$), we follow Xu et al. (2023) and use the Mean Absolute Error (MAE) between the target property $y$ and the predicted value $\hat{y}$ from the regressor of the generated molecule as the evaluation metric.

We add guidance to the last 400 of the 1000 diffusion steps (Han et al., 2024), and calculate the change percentage between our method and the GeoLDM/C-GeoLDM baseline. The implementation details are provided in Appendix B.

Table 1: Metrics of 500 generated molecules from *QM9* using GeoLDM with non-differentiable oracle (i.e. xTB) guidance. * and **bold** denote the overall best result and our best result, respectively. Percentage changes between our results and GeoLDM are shown in parentheses.

| Metric | Guidance Scale | | | | | Baseline | |
|---|---|---|---|---|---|---|---|
| | 0.0001 | 0.001 | 0.01 | 0.1 | 1.0 | EDM | GeoLDM |
| Force RMS (Eh/Bohr) | **0.0104**[*] (-6.76%↓) | 0.0104 | 0.0107 | 0.0108 | 0.0125 | 0.0114 | 0.0111 |
| Validity | **91.40%**[*] (1.60%↑) | 91.20% | 91.20% | 90.00% | 89.40% | 86.60% | 89.80% |
| Uniqueness | **100.00%**[*] | **100.00%**[*] | **100.00%**[*] | **100.00%**[*] | **100.00%**[*] | 100.00%[*] | 100.00%[*] |
| Atom Stability | 99.02% | 98.97% | 98.98% | **99.03%**[*] (0.10%↑) | 98.67% | 98.53% | 98.93% |
| Molecule Stability | 90.60% | 90.40% | 90.20% | **91.00%**[*] (2.20%↑) | 87.80% | 83.60% | 88.80% |
| Energy above Ground State (Eh) | **0.0042**[*] (-15.78%↓) | 0.0042 | 0.0045 | 0.0050 | 0.0061 | 0.0072 | 0.0050 |

### 4.2 NON-DIFFERENTIABLE ORACLE GUIDANCE ON FORCE

We adopt various guidance scales on QM9 and show numeric results in Table 1. It is observed that over different scales, our non-differentiable oracle guidance generates molecules with up to 6.31% lower force, 1.60% and 2.20% higher validity and molecule stability, respectively. Despite optimizing molecular stability via force instead of directly with energy, CHEMGUIDE exhibits a smaller energy change than GeoLDM with a 15.78% decrease. These results show that our method generates molecules closer to the ground state, i.e. the most stable geometry, suggesting that our methods can generate molecules with higher validity and stability. Further, we explore CHEMGUIDE on the GEOM dataset with larger molecules and report the results in Table 2, where a similar trend is observed in our favor: our method generates valid molecules closer to the ground state. Surprisingly, CHEMGUIDE performs better on larger molecules than smaller ones (e.g., a higher decrease in force RMS), which brings potential applications in large-molecule production, e.g., protein. This is partly because larger molecules have more intricate structures and conformational space; thus, are likely to suffer more from distorted geometries than smaller ones, yielding more space for improvement.

---

[2]EDM and GeoLDM are in short of the unconditional version of the models when the context is clear.

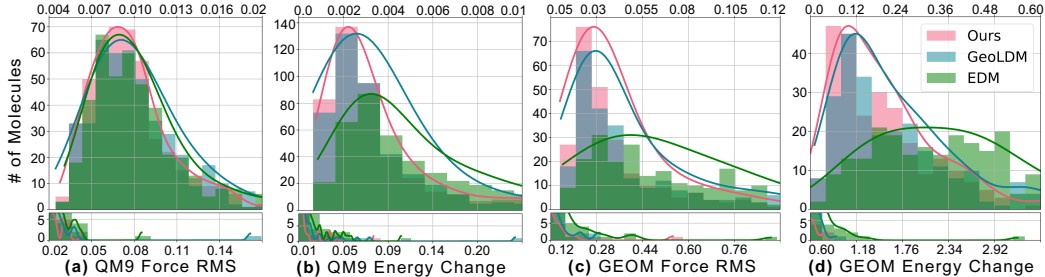

Figure 2: Histograms and distributions of force RMS and energy change of 500 generated molecules from *QM9* and *GEOM* using GeoLDM with CHEMGUIDE under scale=0.0001. Energy change refers to the energy of our generated molecule above its ground state energy.

Table 2: Metrics of 500 generated molecules from *GEOM* using unconditional GeoLDM with non-differentiable oracle (i.e. GFN2-xTB) guidance. * and **bold** denote the overall best result and our best result. Percentage changes between our results and GeoLDM are shown in parentheses.

| Metric | Guidance Scale | | | | | Baseline | |
|---|---|---|---|---|---|---|---|
| | 0.0001 | 0.001 | 0.01 | 0.1 | 1.0 | EDM | GeoLDM |
| Force RMS (Eh/Bohr) | 0.0445 | 0.0447 | 0.0434 | **0.0411**$^*$ (-14.16%↓) | 0.0513 | 0.0742 | 0.0478 |
| Validity | 47.00% | 45.20% | **51.60%**$^*$ (2.60%↑) | 50.40% | 49.40% | 46.40% | 49.00% |
| Uniqueness | **100.00%**$^*$ | **100.00%**$^*$ | **100.00%**$^*$ | **100.00%**$^*$ | **100.00%**$^*$ | 100.00%$^*$ | 100.00%$^*$ |
| Atom Stability | 84.47% | 84.49% | **84.65%**$^*$ (0.12%↑) | 84.36% | 83.45% | 81.22% | 84.53% |
| Energy above Ground State (Eh) | 0.2148 | 0.2085 | 0.2104 | **0.1935**$^*$ (-13.92%↓) | 0.2650 | 0.3742 | 0.2248 |

Fig. 2 shows the plot of the force RMS and energy change histogram and distribution of the 500 generated molecules from EDM, GeoLDM, and CHEMGUIDE on QM9 and GEOM. Our method is more concentrated around 0 and less spread to higher regions. Also, note that both EDM and GeoLDM generate outlier molecules with significantly large force and energy change, while no outlier is generated by CHEMGUIDE. For example, in Fig. 2(a), an outlier with force RMS 0.155 is generated by GeoLDM, which is ~15 times higher than the force RMS where the molecules are concentrated. It demonstrates that our method performs better both on average and molecule-wise, and we provide detailed plots of various scales in Fig. 11 in Appendix K.

To further validate the effectiveness of CHEMGUIDE, we compute stability metrics at the DFT/B3LYP/6-31G(2df,p) level of theory with *Gaussian16* (Frisch et al., 2016) for more precise and strict benchmarking. As *DFT* is computationally demanding, we only present the results of CHEMGUIDE with the best scale, GeoLDM, and EDM on QM9 in Table 3. Despite applying a less accurate but computationally feasible oracle (i.e., GFN2-xTB), CHEMGUIDE still improves validity and stability (i.e., valid molecular geometry close to optimized ground state geometry), when measured against higher standards (i.e., DFT). On the other hand, applying DFT as the non-differentiable oracle to guide the generation process is computationally expensive and impracticable.

Table 3: Metrics of 500 generated molecules from *QM9* using GeoLDM with CHEMGUIDE using scale=0.0001, calculated at DFT/B3LYP/6-31G(2df,p) level of theory, more precise but computationally expensive than GFN2-xTB. *Strict validity* is defined as generated geometries within 50 kcal/mol (0.07968 Eh) of the optimized geometries. * and **bold** denote the overall best result and our best result. Percentage changes between our results and GeoLDM are shown in parentheses.

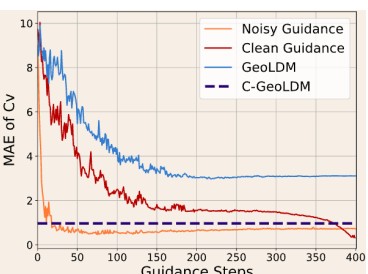

Figure 3: MAE trajectories of GeoLDM, with noisy/clean guidance, and C-GeoLDM on $C_v$.

| Metric | Ours | EDM | GeoLDM |
|---|---|---|---|
| Validity | **96.2%**$^*$ (2.6%↑) | 89.0% | 93.6% |
| Strict Validity | **95.6%**$^*$ (2.4%↑) | 88.4% | 93.2% |
| Energy above Ground State (Eh) | **0.004051**$^*$ (-9.6%↓) | 0.006314 | 0.004480 |
| Energy above GS (kcal/mol) | **2.542**$^*$ (-9.6%↓) | 3.962 | 2.811 |

### 4.3 NEURAL NETWORK GUIDANCE ON PROPERTIES

#### 4.3.1 UNCONDITIONAL DIFFUSION WITH NOISY NEURAL GUIDANCE

We present the results of noisy guidance (Section 2) in Table 4 and provide further details in Table 16 in Appendix L. For all the properties $\alpha$, $\mu$, $C_v$, $\epsilon_{HOMO}$, $\epsilon_{LUMO}$, and $\Delta\epsilon$, noisy neural guidance outperforms **unconditional** GeoLDM by 57.92%, 37.24%, 77.09%, 23.18%, 36.87%, and 37.88%

Table 4: MAE of 500 generated molecules from *QM9* using GeoLDM with noisy/clean neural guidance, where the result from the best configuration is reported. We use C-EDM, C-GeoLDM, and GeoLDM as baselines. $*$ and **bold** denote the overall best and the best within different scales, respectively. Percentage changes between our results and C-GeoLDM are shown in parentheses.

| Property | $\alpha$ | $\mu$ | $C_v$ | $\epsilon_{\text{HOMO}}$ | $\epsilon_{\text{LUMO}}$ | $\Delta\epsilon$ |
|---|---|---|---|---|---|---|
| Units | Bohr$^3$ | D | $\frac{\text{cal}}{\text{mol}}$K | meV | meV | meV |
| Conditional EDM | 2.6308 | 1.1257 | 1.0804 | 0.3207 | 0.5940 | 0.6301 |
| Conditional GeoLDM | 2.5551 | 1.1084 | 0.9703 | 0.3327 | 0.5518$^*$ | 0.5878 |
| GeoLDM | 5.6732 | 1.6461 | 3.1046 | 0.6151 | 1.1778 | 1.2022 |
| **GeoLDM w/ Noisy Guidance** | **2.3870**$^*$ (-6.58%↓) | 1.0331 | 0.7112 | 0.4725 | 0.7436 | 0.7468 |
| **GeoLDM w/ Clean Guidance** | 5.4798 | **0.2031**$^*$ (-81.68%↓) | **0.3151**$^*$ (-67.53%↓) | **0.3109**$^*$ (-3.06%↓) | 0.5688 | **0.3830**$^*$ (-34.84%↓) |

respectively, which verifies its effectiveness. Furthermore, GeoLDM with noisy neural guidance yields comparable performance with C-GeoLDM and C-EDM, provided that conditional models are explicitly trained for property optimization; surprisingly, we use unconditional GeoLDM as the backbone and outperforms C-GeoLDM in $\alpha$, $\mu$, and $C_v$. To understand the details of guided generation, we plot the MAE of $C_v$ on sampled molecules at each guidance step in Fig. 3. Compared to GeoLDM which oscillates in the first 200 steps and converges slowly to a high MAE, the trajectory of noisy neural guidance drops fast in the first 50 steps and the amplitude of the oscillation is less intense, which suggests the guidance smooths the MAE change and quickly steers the model toward better properties. The property MAE trajectories are detailed in Fig. 12 in Appendix L.

### 4.3.2 UNCONDITIONAL DIFFUSION WITH CLEAN NEURAL GUIDANCE

As presented in Table 4, clean neural guidance outperforms unconditional GeoLDM on all properties and beats C-GeoLDM in $\mu$, $C_v$, $\epsilon_{\text{HOMO}}$, and $\Delta\epsilon$ by 81.68%, 67.53%, 3.06%, and 34.84%. Interestingly, it seems that noisy and clean guidance complement each other on different properties (e.g., $\alpha, \Delta\epsilon$). In Fig. 3, clean neural guidance shows a similar oscillation trend in the first 200 guidance steps compared to GeoLDM, then exhibits a sharp drop in the last 50 steps and achieves the lowest MAE, which is also observed on other properties shown in Fig. 12 in Appendix L. We hypothesize this is due to that molecules are less noisy as denoising steps accumulate; thus, guidance optimization in the clean space manifests its strongest power when $z_t$ is less dependent on the one-step estimation function $t_0(\cdot)$ in later steps, where it is closer and more similar to $z_0$ (Eq. 8).

### 4.4 COMBINED GUIDANCE ON FORCE AND PROPERTIES

We present the results for the bilevel optimization on molecular property and stability, with MAE shown in Table 5, and the force RMS and energy change in Fig. 4. We sample 200 molecules for combined guidance, as there are 6 properties to optimize, and force optimization with xTB only supports CPU computation. We choose conditional GeoLDM (C-GeoLDM) as the baseline.

Table 5: MAE of 200 generated molecules using C-GeoLDM with CHEMGUIDE and GeoLDM with noisy neural guidance, where the result from the best configuration is reported. We use C-EDM and C-GeoLDM as baselines. $*$ and **bold** denote the overall best and the best within different scales, respectively. Percentage changes between our results and C-GeoLDM are shown in parentheses.

| Property | $\alpha$ | $\mu$ | $C_v$ | $\epsilon_{\text{HOMO}}$ | $\epsilon_{\text{LUMO}}$ | $\Delta\epsilon$ |
|---|---|---|---|---|---|---|
| Units | Bohr$^3$ | D | $\frac{\text{cal}}{\text{mol}}$K | meV | meV | meV |
| C-EDM | 2.5089 | 1.0571 | 1.0624 | 0.3304 | 0.5046 | 0.6191 |
| C-GeoLDM | 2.5593 | 1.1582 | 0.9646 | 0.3407 | 0.5927 | 0.4982$^*$ |
| **Bilevel Optimization w/ Explicit Guidance** | **2.4430**$^*$ (-4.54%↓) | 1.0483 | 0.9393 | **0.3404**$^*$ (-0.09%↓) | **0.5077**$^*$ (-14.34%↓) | **0.5225** (4.88%↑) |
| **Bilevel Optimization w/ Noisy Guidance** | 2.4860 | **0.7246**$^*$ (-37.43%↓) | **0.9369**$^*$ (-2.87%↓) | 0.4445 | 0.8099 | 0.7041 |

### 4.4.1 EXPLICITLY GUIDED DIFFUSION WITH CHEMGUIDE

We employ C-GeoLDM with CHEMGUIDE, which resembles the bilevel optimization framework as C-GeoLDM automatically optimizes molecular property during the denoising process. Compared to C-GeoLDM alone, we expect our method to reduce the force while not hurting MAE. We present

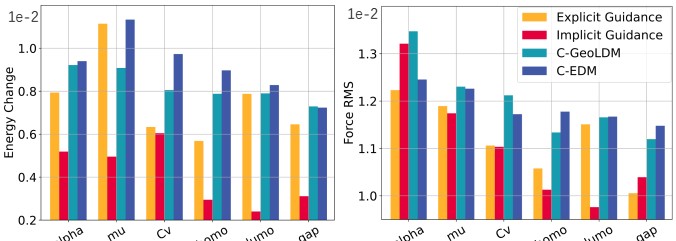 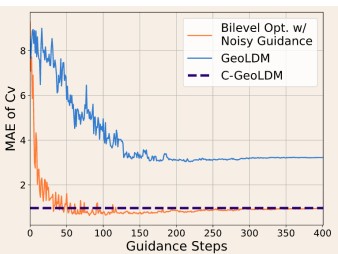

Figure 4: Force RMS and energy change of 200 generated molecules using explicit bilevel optimization, implicit bilevel optimization with noisy neural guidance, C-EDM, and C-GeoLDM. Energy change refers to energy above ground state.

Figure 5: MAE trajectories of GeoLDM, C-GeoLDM, and our bilevel optimization on $C_v$.

the results in Fig. 4 and Table 5 (details are provided in Table 17 and Table 18 in Appendix M). For the 6 properties ($\alpha$, $\mu$, $C_v$, $\epsilon_{HOMO}$, $\epsilon_{LUMO}$, $\Delta\epsilon$), our method reduces the force RMS by 9.21%, 3.33%, 8.75%, 6.69%, 1.27%, and 10.19%, and for the first 5, MAEs decrease 4.54%, 9.49%, 2.62%, 0.09%, and 14.34%. The results suggest it is feasible and effective to combine explicitly guided diffusion (e.g., C-GeoLDM) with CHEMGUIDE, which goes beyond our expectation as both force RMS and property MAE are reduced.

### 4.4.2 IMPLICITLY GUIDED DIFFUSION WITH CHEMGUIDE

We apply two aforementioned neural network guidance approaches (noisy/clean guidance, Section 2) for the property optimization step in our bilevel optimization framework (Section 3.2). We expect that both the property MAE and the force metrics are better than conditional GeoLDM.

**Bilevel optimization with noisy guidance** As shown in Table 5, our method outperforms C-GeoLDM in $\alpha$, $\mu$, and $C_v$ by 2.86%, 37.43%, and 2.87% in MAE. It is also observed in Fig. 4 that, for the six properties, our method achieves lower force RMS than C-GeoLDM by 1.94%, 4.57%, 8.97%, 10.66%, 16.26%, and 7.18%, with smaller energy change compared to baselines. Details of MAE and force metrics are provided in Table. 18 and Table. 19 in Appendix M. Combining property optimization with force optimization, noisy guidance with CHEMGUIDE reduces MAE and improves the stability of generated molecules, which suggests that the combined guidance prompts the diffusion model towards generating stabler molecules with better properties.

To further understand how our bilevel framework optimizes property, we provide the MAE trajectory of $C_v$ in Fig. 5. The trajectories of our noisy neural guidance method (Fig.3) and bilevel optimization behave similarly, such that the MAE drops fast in the first 50 steps and oscillates less compared to GeoLDM. However, bilevel optimization is slightly worse than the noisy neural guidance method (i.e. it converges closer to C-GeoLDM than noisy guidance). We suspect it is because guided by gradients from the neural regressor and then CHEMGUIDE, the generation process leans less toward a lower MAE, as it keeps a balance between property and stability optimization. Details and explanations of the MAE trajectories of each property can be found in Fig. 13 in Appendix M.

**Bilevel optimization with clean guidance** We present the details of this section in Appendix J.

### 4.5 CHEMGUIDE BEYOND STABILITY

In this section, we explore the potential of using CHEMGUIDE to optimize molecular properties other than force and energy. We sample 200 molecules using **unconditional** GeoLDM with CHEMGUIDE on $\alpha$, $\mu$, $\epsilon_{HOMO}$, $\epsilon_{LUMO}$, and $\Delta\epsilon$, which can also be calculated at GFN2-xTB[3], and the results are shown in Table 6. Specifically, since the property label of a molecule from QM9 (calculated at DFT) may be different from the xTB computation, we obtain property scores of all QM9 molecules with xTB as a distribution where we sample the guidance target, which is the reason that GeoLDM has a different MAE than Table 5, and *GEOM* is processed similarly. It is observed that CHEMGUIDE on properties improves the baseline in $\mu$, $\epsilon_{LUMO}$, and $\Delta\epsilon$. Our method performs less satisfyingly on $\alpha$ and $\epsilon_{HOMO}$ because they are inherently more challenging to optimize, as observed in Table 4. We can see that CHEMGUIDE is general and applicable to a broad range of molecular optimization tasks other than stability across different datasets. Full results can be found in Table 14 and 15 in Appendix K, and our discussion on the difference between CHEMGUIDE and neural guidance for properties is presented in Appendix H.4.

---

[3] $C_v$ is skipped as its calculation requires the Hessian matrix, which is expensive to compute.

Table 6: CHEMGUIDE for properties ($\alpha$, $\mu$, $\epsilon_{HOMO}$, $\epsilon_{LUMO}$, $\Delta\epsilon$) with 200 generated molecules on *QM9* and *GEOM*, where the result from the best configuration is reported. We use GeoLDM as the baseline. $*$ and **bold** denote the overall best and the best within different scales, respectively. Percentage changes between our results and GeoLDM are shown in parentheses.

| | Property | $\alpha$ | $\mu$ | $\epsilon_{HOMO}$ | $\epsilon_{LUMO}$ | $\Delta\epsilon$ |
| --- | --- | --- | --- | --- | --- | --- |
| | Units | Bohr$^3$ | D | meV | meV | meV |
| *QM9* | GeoLDM | 4.7555$^*$ | 1.5490 | 0.6264$^*$ | 2.1790 | 2.1568 |
| | GeoLDM w/ CHEMGUIDE | **4.8781** | **1.5295**$^*$ | **0.6392** | **2.0091**$^*$ | **1.9318**$^*$ |
| | | (2.58%↑) | (-1.26%↓) | (2.04%↑) | (-7.80%↓) | (-10.43%↓) |
| *GEOM* | GeoLDM | 24.3424 | 3.3676 | 1.1922$^*$ | 5.0903 | 1.9956 |
| | GeoLDM w/ CHEMGUIDE | **24.2168**$^*$ | **3.3412**$^*$ | **1.2939** | **5.0802**$^*$ | **1.9264**$^*$ |
| | | (-0.52%↓) | (-0.78%↓) | (8.53%↑) | (-0.20%↓) | (-3.47%↓) |

## 5 ANALYSES

In Appendix H, we discuss acceleration methods for CHEMGUIDE (Section H.1); different guidance steps (Section H.2); neural guidance for stability optimization (Section H.3); generalization analyses of property regressor (Section H.4) and conditional diffusion model (Section H.5); the effect of guidance scales for noisy/clean guidance (Section H.6); the effect of optimization steps for clean guidance (Section H.7).

## 6 RELATED WORK

Due to space limits, we present the discussion of related works in Appendix C.

## 7 CONCLUSION & FUTURE WORK

In this work, we study conditional generation for 3D diffusion models on molecules. We propose CHEMGUIDE, which guides the generation process of an unconditionally trained diffusion model using an external, typically non-differentiable, chemistry oracle. By analytically estimating the guidance gradients using zeroth-order optimization methods, while ensuring the equivariance and invariance requirements for 3D diffusion models, we aim to enhance the stability of generated molecules without relying on labeled data to train an additional property predictor. We observe reduced atomic forces and improved molecular validity across both small and large molecule datasets when the model is equipped with CHEMGUIDE, highlighting the effectiveness of our approach. Further experiments incorporating both CHEMGUIDE and neural guidance within a bilevel optimization framework demonstrate improvements in both molecular properties and stability. This suggests that our non-differentiable guidance is compatible with existing methods. Moreover, when CHEMGUIDE is used for property optimization alone, it shows strong generalization beyond force minimization.

However, since xTB runs on CPU and neural guidance requires gradients that heavily utilize GPU resources, our method may be constrained by limited or insufficient computational resources. Future work could explore ways to accelerate the guidance process while reducing computational costs without compromising performance. Additionally, we observe that the guidance scale plays a critical role in the generative process of guided diffusion for molecules, but it is generally difficult to determine in advance. This opens the possibility for research into more effective methods for selecting guidance scales, including automated scale scheduling that optimizes guidance strength.

In summary, we propose CHEMGUIDE, which introduces guidance signals for the diffusion process from a non-differentiable quantum chemical oracle. We hope this work sparks further exploration into how domain knowledge from physics and chemistry can be more effectively integrated into neural networks to enhance molecular design.

## 8 ACKNOWLEDGMENTS

We are grateful to our anonymous ICLR reviewers, with whose help this work has been greatly improved. The work was partially supported by the NSF Award 2307698.

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

# A    LIST OF CONTENTS

We detail the contents in the appendix below.

- **Implementation Details** (Section B) includes the hardware and running time, the models, the evaluation metrics for our experiments, and the link to the code repository.
- **Related Work** (Section C) includes related literature on molecule generation and guided diffusion.
- **GFN2-xTB** (Section D) introduces the method used by CHEMGUIDE to perform quantum chemistry calculation.
- **Motivation for Stability Optimization** (Section E) explains the importance and motivation of stability optimization for molecule generation.
- **Motivation for Gradient Estimation with SPSA** (Section F) explains the motivation of choosing SPSA as the zeroth-order optimization method to estimate gradients.
- **Chemistry Reasoning behind Bilevel Property Optimization** (Section G) explains the chemistry intuition for the bilevel optimization framework on molecular property and stability.
- **Analyses** (Section H) discusses acceleration methods for CHEMGUIDE (Section H.1); different guidance steps (Section H.2); neural guidance for stability optimization (Section H.3); generalization analyses of property regressor (Section H.4) and conditional diffusion model (Section H.5); the effect of guidance scales for noisy/clean guidance (Section H.6); the effect of optimization steps for clean guidance (Section H.7).
- **Other Optimization Algorithms** (Section I) discusses the possibility of conditional generation and stability optimization with evolutionary algorithm, and how it can be used together with CHEMGUIDE.
- **Bilevel Optimization with Clean Guidance** (Section J) discusses using clean guidance and CHEMGUIDE in the bilevel optimization framework.
- **Comprehensive Results on Non-differentiable Oracle Guidance** (Section K) presents detailed results of CHEMGUIDE for stability and molecular property optimization.
- **Comprehensive Results on Neural Guidance** (Section L) presents detailed results of noisy and clean guidance for molecular property optimization.
- **Comprehensive Results on Bilevel Optimization** (Section M) presents detailed results for molecular property and stability optimization.
- **Molecule Visualization** (Section N) visualizes the generated molecules.

# B    IMPLEMENTATION DETAILS

**Hardware & Time**    We use a 48 GiB A6000 GPU with AMD EPYC 7513 32-Core Processors for our experiments. For molecular optimization conducted with xTB (e.g., force, $\alpha$), it takes around 6 and 18 hours to sample 100 molecules on the QM9 and GEOM datasets, respectively.

**Model**    For unconditional EDM (Hoogeboom et al., 2022) and GeoLDM (Xu et al., 2023), we use the checkpoints available in the official implementations. For the conditional models (i.e., C-EDM, C-GeoLDM) and the property regressors, we follow the instructions and train them with the hyperparameters specified by the authors.

**Evaluation Metric**    We present the metrics to evaluate molecular stability and property in our experiment as follows.

For stability optimization, we report the following metrics:

- **force RMS** (Eh/Bohr): we compute the root mean square (RMS) of the forces on the atoms of a molecule; it is the *lower the better*.
- **validity**: it measures how many percent of the generated molecules are valid, and is calculated by either xTB or DFT and specified in the context; it is the *higher the better*.

- **uniqueness**: it is the percentage of unique molecules among the generated molecules (Xu et al., 2023); it is the *higher the better*.

- **atom stability**: it is the percentage of atoms that have the correct valency (Xu et al., 2023); it is the *higher the better*.

- **molecule stability**: it is the percentage of generated molecules whose atoms are all stable (Xu et al., 2023); it is the *higher the better*.

- **energy above ground state** (Eh): it is the difference between the energy before and after using xTB or DFT (specified in the context) to optimize the geometry of the generated molecules, and a smaller energy above ground state indicates the geometry is closer to the stable ground state of the molecule; it is the *lower the better*.

- **strict validity**: it is defined as generated geometries within 50 kcal/mol (0.07968 Eh) of the optimized geometries; it is the *higher the better*.

For property optimization on $\alpha$ (Bohr$^3$), $\mu$ (Debye), $C_v$ cal/(mol·K), $\epsilon_{\text{HOMO}}$ (eV), $\epsilon_{\text{LUMO}}$ (eV), $\Delta\epsilon$ (eV) , we report the following metrics:

- **MAE**: we compute the Mean Absolute Error (MAE) between the target property $y$ and the predicted value $\hat{y}$ by either the regressor or xTB (specified in the context) of the generated molecules; it is the *lower the better*.

We use one step of guidance/optimization for our experiments (e.g., clean guidance in Section 2) by default. Our implementation is available at https://github.com/A-Chicharito-S/ChemGuide.

## C  RELATED WORK

**Molecule Generation**   Previous works focus on generating molecules as SMILES strings (Kusner et al., 2017; Gómez-Bombarelli et al., 2018; Segler et al., 2018), and 2D graphs (Li et al., 2018; Jo et al., 2022), where various models, such as VAE (Simonovsky & Komodakis, 2018; Jin et al., 2018; 2019), GAN (De Cao & Kipf, 2018; Prykhodko et al., 2019), diffusion models (Vignac et al., 2022; Kong et al., 2023; Zhao et al., 2024a), have been proposed to serve as the generative backbone. To model the spatial structure of molecules, G-Schnet (Gebauer et al., 2019) and G-SphereNet (Luo & Ji, 2022) use autoregressive models to generate the 3D coordinates of the molecules, however, they are less powerful compared with diffusion models. Hoogeboom et al. (2022) propose to use the equivariant diffusion model (EDM) for 3D molecule generation, which utilizes an equivariant graph neural network (Satorras et al., 2021) to model the molecules as a 3D graph, with coordinates and atom types as its node features. GeoLDM (Xu et al., 2023) further extends EDM to latent diffusion, and has shown improvements in stability and validity of the generated molecules. However, to achieve conditional generation, both EDM and GeoLDM need to be re-trained with labels, where the target property value is appended to the feature space as auxiliary information, which guides the generation toward fulfilling certain molecular property requirements.

**Guided Diffusion**   Guided diffusion has shown promising results in vision (Bansal et al., 2023), where the guidance can be provided by texts (Rombach et al., 2022), poses and edges (Zhang et al., 2023), classifiers (Dhariwal & Nichol, 2021), etc. Similar approaches are adapted for diffusion-based molecule generation, where *explicitly* guided diffusion (Hoogeboom et al., 2022; Xu et al., 2023) trains a conditional model with labels, and *implicitly* guided diffusion (Vignac et al., 2022; Bao et al., 2022) combines an unconditional model with a conditional property network, which provides guidance as gradients to the diffusion process at inference. Gruver et al. (2023) propose guided discrete diffusion for multi-objective protein sequence optimization, and Han et al. (2024) achieve multi-property guidance by modeling property relations as probabilistic graphs. Weiss et al. (2023) investigate guided diffusion and demonstrate the capability to design molecules beyond the seen property distribution, while MOOD (Lee et al., 2023) and GCD (Klarner et al., 2024) incorporate out-of-distribution control to improve generalization of the property predictor. However, unlike CHEMGUIDE, all of the above works require a labeled dataset to train a property network; thus, inevitably constrained by the generalization ability of the neural networks. Beyond molecule optimization, diffusion guidance has been discussed in the broader literature, such as protein design (Wu et al., 2023), large language models (Zhao et al., 2024b), and music generation (Huang et al., 2024).

## D  GFN2-XTB

GFN stands for, respectively, **g**eometry optimization, vibrational **f**requencies, and **n**on-covalent interactions. xTB refers to e**x**tended **t**ight **b**inding, and 2 refers to the version. In the GFN2-xTB method, the total energy expression is given by Bannwarth et al. (2021; 2019)

$$E_{\text{GFN2-xTB}} = E_{\text{rep}} + E_{\text{disp}} + E_{\text{EHT}} + E_{\text{IES+IXC}} + E_{\text{AES}} + E_{\text{AXC}} + G_{\text{Fermi}} \tag{18}$$

where $E_{\text{rep}}$ is the repulsive energy contribution from short-range interactions, $E_{\text{disp}}$ is the dispersion energy contribution from long-range interactions, $E_{\text{EHT}}$ is the energy contribution from the extended Hückel theory (EHT), $E_{\text{IES+IXC}}$ is the isotropic electrostatic (IES) energy contribution and the isotropic exchange-correlation (IXC) energy contribution, $E_{\text{AES}}$ is the anisotropic electrostatic (AES) energy contribution, $E_{\text{AXC}}$ is the anisotropic exchange-correlation (AXC) energy contribution, and $G_{\text{Fermi}}$ is the entropic contribution of an electronic free energy at finite electronic temperature $T_{\text{el}}$ due to Fermi smearing.

Its accuracy and efficiency come strictly from the element-specific and global parameters for all elements up to radon (Z = 86) (Bannwarth et al., 2019), hence the semi-empiricism. The pre-computed tight-binding parameters and empirical corrections are utilized to approximate the electronic structure and calculate energy contributions efficiently.

## E  MOTIVATION FOR STABILITY OPTIMIZATION

The geometries of the generated molecules have domain-specific implications—a molecule's stability and properties depend significantly on its preferred quantum geometric states, i.e., atomic and molecular geometries. For example, polarity is closely related to molecular geometry. A water molecule $H_2O$ has a stable V-shaped H-O-H geometry of 104.5 degrees and is thus polar. A generated $H_2O$ molecule with a linear H-O-H geometry would instead be nonpolar but unstable. Therefore, when we discuss molecules and their properties, it is essential to start from their preferred stable geometries (ground states), which motivates our work to improve the stability of the generated molecular geometries in the diffusion process.

## F  MOTIVATION FOR GRADIENT ESTIMATION WITH SPSA

To achieve equivariance to transformations (Eq. 4), the 3D diffusion models must satisfy the *zero center gravity* requirement that $\mathbf{z}_{\mathbf{x},t} \in \mathbb{R}^{N \times 3}$ should have zero-mean over $N$ atoms on the $x, y, z$ axis. To derive guidance, one can estimate $\nabla_{\mathbf{z}_t} \mathcal{F}(\mathbf{z}_t)$ in the gradient using finite-different approximation (from Eq. 14) as:

$$\nabla_{\mathbf{z}_t} \mathcal{F}(\mathbf{z}_t) \approx \lim_{\zeta \to 0} \frac{\mathcal{F}([\mathbf{z}_{\mathbf{x},t} + \zeta \mathbf{1}_{N \times 3}, \mathbf{z}_{\mathbf{h},t}]) - \mathcal{F}([\mathbf{z}_{\mathbf{x},t} - \zeta \mathbf{1}_{N \times 3}, \mathbf{z}_{\mathbf{h},t}])}{2\zeta} \tag{19}$$

which however violates *zero center gravity* as the perturbed variable $\mathbf{z}_{\mathbf{x},t} \pm \zeta \mathbf{1}_{N \times 3}$ is shifted by $\pm \zeta \mathbf{1}_{N \times 3}$ over the zero-mean of $\mathbf{z}_{\mathbf{x},t}$.

To obey the zero-mean requirements, one can employ SPSA (Simultaneous Perturbation Stochastic Approximation) (Spall, 1992) by perturbing $\mathbf{z}_{\mathbf{x},t}$ with a zero-mean random variable $\Delta_t$ as follows.

$$\nabla_{\mathbf{z}_t} \mathcal{F}(\mathbf{z}_t) \approx \lim_{\zeta \to 0} \frac{\mathcal{F}([\mathbf{z}_{\mathbf{x},t} + \zeta \Delta_t, \mathbf{z}_{\mathbf{h},t}]) - \mathcal{F}([\mathbf{z}_{\mathbf{x},t} - \zeta \Delta_t, \mathbf{z}_{\mathbf{h},t}])}{2\zeta \Delta_t} \tag{20}$$

where we abuse the notation by using dividing $\Delta_t$ to denote the element-wise operation. Spall (1992) assumes $\mathbf{z}_{\mathbf{x},t}$ to have bounded inverse moments, which excludes $\Delta_t$ being Gaussian. However, the Gaussian perturbation $[\mathbf{z}_{\mathbf{x},t} \pm \zeta \mathbf{U}, \mathbf{z}_{\mathbf{h},t}]$ with $\mathbf{U} \in \mathbb{R}^{N \times 3} \sim \mathcal{N}(0,1)$ would be favored, since it is one of the commonly used distributions that obey *zero central gravity* with $\mathbb{E}[\zeta \mathbf{U}] = 0$, while only introduce small computational overhead compared to finite-different methods, as sampling from $\mathcal{N}(0,1)$ and matrix multiplication are fast in modern machine learning practices on GPUs.

To achieve so, one can adopt the *random gradient-free oracles* (Nesterov & Spokoiny, 2017) to estimate the gradient for differentiable $f : E \to R$. Given the random vector $u \in E$ being Gaussian

with correlation operator $B^{-1}$ (e.g., $B = I$) and $\mu \geq 0$, the gradient at $x \in E$ is estimated as follows.

$$\hat{g}_u(x) = \frac{f(x + \mu u) - f(x - \mu u)}{2\mu} \cdot Bu \tag{21}$$

which leads to our estimation for the guidance gradient in Eq. 15.

## G  CHEMISTRY REASONING BEHIND BILEVEL PROPERTY OPTIMIZATION

The task of generating molecules with target properties is closely related to the field of rational chemical design such as drug design. Domain knowledge is widely used to guide the design process. For example, benzene rings that contain oxygen (e.g., dioxins) and sulfur (e.g., thiophenes) are typically known to be toxic and carcinogenic. Therefore, when designing drugs, it is typical to stay away from such species prior to performing further property analysis such as bind-affinity to a specific protein or DNA. In such design processes, the overall structures are considered by experts before further target property analysis.

The conflict behind generating stable molecules with target properties is that properties are closely related to both atom types $\mathbf{h}$ and atom coordinates $\mathbf{x}$. We are mostly concerned with the properties of a stable molecule with an optimized geometry but it is not known *a priori* given the atom types. In the diffusion process to generate molecules, the atom types are generated and stabilized during earlier steps, and their coordinates are further optimized in later steps. Since our target is to generate stable molecules with desired properties, there is a conflict between stability-first and property-first in the generation process.

The *bilevel* property and stability optimization proposed here takes inspiration from the rational design process: higher-level overall structure before lower-level property detail analysis. An overall molecular structure is generated with desired properties in earlier steps, where the diffusion model is considered as chemistry/biology/etc. experts performing the initial structure-property analysis. In the later steps, the geometries are further optimized.

## H  ANALYSES

### H.1  ACCELERATION METHODS FOR CHEMGUIDE

GPUs can significantly accelerate computation for neural network inference and training, but xTB operates on the CPU, which lowers the running speed of the inference. Hence, we propose a *skip-step* acceleration method for xTB-guided optimization. Specifically, suppose we set the skip-step to be $k$, then we only calculate gradients from xTB for every $k$ step. We use the best scale (=0.0001) found in Table 1 for the skip-step acceleration experiment, and the results are shown in Table 7. As we skip more steps, the performance becomes worse and approaches unconditional GeoLDM without any guidance. Hence, there is a trade-off between time and performance: the more guidance from xTB we add to the generation, the better performance we can end with. This also demonstrates the effectiveness and necessity of the xTB guidance: more guidance almost monotonically enhances performance.

Table 7: Metrics of 500 generated molecules from *QM9* using GeoLDM with non-differentiable oracle (i.e. xTB) guidance. * and **bold** denote the overall best result and our best result, respectively. Percentage changes between our results and GeoLDM are shown in parentheses.

| Metric | Every N step (s) | | | Baseline | |
|---|---|---|---|---|---|
| | 1 | 3 | 5 | EDM | GeoLDM |
| Force RMS (Eh/Bohr) | **0.0104**\* (-6.76%↓) | 0.0106 | 0.0011 | 0.0114 | 0.0111 |
| Validity | **91.40%**\* (1.60%↑) | 91.40% | 89.40% | 86.60% | 89.80% |
| Uniqueness | **100.00%**\* | 99.79% | 100.00% | 100.00% | 100.00% |
| Atom Stability | 99.02% | 99.01% | 98.63% | 98.53% | 98.93% |
| Molecule Stability | 90.60% | 90.60% | 87.20% | 83.60% | 88.80% |
| Energy above ground state (Eh) | **0.0042**\* (-15.78%↓) | 0.0045 | 0.0055 | 0.0072 | 0.0050 |

Table 8: Metrics of 500 generated molecules from *QM9* using GeoLDM with non-differentiable oracle (i.e. xTB) guidance with various guidance steps (100, 200, 300, 400) and guidance scales (0.0001, 0.001, 0.01, 0.1, 1.0). * and **bold** denote the overall best result and our best result, respectively. Percentage changes between our results and GeoLDM are shown in parentheses.

| Metric | Guidance Scale (100 Guidance Steps) | | | | | Baseline | |
|---|---|---|---|---|---|---|---|
| | 0.0001 | 0.001 | 0.01 | 0.1 | 1.0 | EDM | GeoLDM |
| Force RMS (Eh/Bohr) | **0.0110*** (-0.88%↓) | **0.0110*** (-0.88%↓) | 0.0110 | 0.0113 | 0.0120 | 0.0114 | 0.0111 |
| Validity | 90.40% | 90.40% | 90.40% | **90.80%*** (1.00%↑) | 90.60% | 86.60% | 89.80% |
| Uniqueness | **100.00%*** | **100.00%*** | **100.00%*** | **100.00%*** | **100.00%*** | 100.00%* | 100.00%* |
| Atom Stability | **98.94%*** (0.01%↑) | **98.94%*** (0.01%↑) | **98.94%*** (0.01%↑) | 98.91% | 98.92% | 98.53% | 98.93% |
| Molecule Stability | 89.40% | 89.40% | 89.40% | **89.60%*** (0.80%↑) | **89.60%*** (0.80%↑) | 83.60% | 88.80% |
| Energy above ground state (Eh) | 0.0051 | **0.0051*** (0.87%↑) | 0.0051 | 0.0055 | 0.0057 | 0.0072 | 0.0050 |
| Metric | Guidance Scale (200 Guidance Steps) | | | | | Baseline | |
| | 0.0001 | 0.001 | 0.01 | 0.1 | 1.0 | EDM | GeoLDM |
| Force RMS | 0.0119 | **0.0106*** (-4.28%↓) | 0.0119 | 0.0114 | 0.0116 | 0.0114 | 0.0111 |
| Validity | 90.00% | **91.00%*** (1.20%↑) | 90.00% | 90.60% | 90.60% | 86.60% | 89.80% |
| Uniqueness | **100.00%*** | **100.00%*** | **100.00%*** | **100.00%*** | **100.00%*** | 100.00%* | 100.00%* |
| Atom Stability | 98.87% | 98.72% | 98.87% | **98.90%*** (-0.03%↓) | 98.86% | 89.53% | 98.93% |
| Molecule Stability | 89.00% | 88.40% | 89.00% | **89.40%*** (0.60%↑) | 89.00% | 83.60% | 88.80% |
| Energy above ground state (Eh) | 0.0054 | **0.0045*** (-9.32%↓) | 0.0054 | 0.0049 | 0.0050 | 0.0072 | 0.0050 |
| Metric | Guidance Scale (300 Guidance Steps) | | | | | Baseline | |
| | 0.0001 | 0.001 | 0.01 | 0.1 | 1.0 | EDM | GeoLDM |
| Force RMS | 0.0109 | 0.0109 | **0.0107*** (-3.98%↓) | 0.0108 | 0.0115 | 0.0114 | 0.0111 |
| Validity | **92.80%*** (3.00%↑) | **92.80%*** (3.00%↑) | 92.60% | 92.20% | 88.80% | 86.60% | 89.80% |
| Uniqueness | 99.57% | **100.00%*** | 99.58% | 99.36% | **100.00%*** | 100.00%* | 100.00%* |
| Atom Stability | 98.93% | **99.09%*** (0.15%↑) | 99.00% | 98.99% | 98.88% | 98.53% | 98.93% |
| Molecule Stability | 89.80% | **90.80%*** (2.00%↑) | 89.60% | 89.60% | 88.00% | 83.60% | 88.80% |
| Energy above ground state (Eh) | 0.0049 | 0.0052 | **0.0045*** (-10.64%↓) | 0.0046 | 0.0056 | 0.0072 | 0.0050 |
| Metric | Guidance Scale (400 Guidance Steps) | | | | | Baseline | |
| | 0.0001 | 0.001 | 0.01 | 0.1 | 1.0 | EDM | GeoLDM |
| Force RMS (Eh/Bohr) | **0.0104*** (-6.76%↓) | 0.0104 | 0.0107 | 0.0108 | 0.0125 | 0.0114 | 0.0111 |
| Validity | **91.40%*** (1.60%↑) | 91.20% | 91.20% | 90.00% | 89.40% | 86.60% | 89.80% |
| Uniqueness | **100.00%*** | **100.00%*** | **100.00%*** | **100.00%*** | **100.00%*** | 100.00%* | 100.00%* |
| Atom Stability | 99.02% | 98.97% | 98.98% | **99.03%*** (0.10%↑) | 98.67% | 98.53% | 98.93% |
| Molecule Stability | 90.60% | 90.40% | 90.20% | **91.00%*** (2.20%↑) | 87.80% | 83.60% | 88.80% |
| Energy above Ground State (Eh) | **0.0042*** (-15.78%↓) | 0.0042 | 0.0045 | 0.0050 | 0.0061 | 0.0072 | 0.0050 |

## H.2 EFFECT OF GUIDANCE STEPS

As mentioned in Appendix B, adding guidance to the last 400 of the 1000 diffusion steps is time-consuming, so we explore relaxations in this section, where we add guidance to later steps such as the last 300, 200, and 100 steps. The results are presented in Table 8 for QM9 and Table 9 for GEOM. It is observed that more guidance steps yield better results in terms of force RMS and energy above ground state while the effect on validity and stability does not seem to be related to the number of guidance steps. This is reasonable because we are optimizing the force during the diffusion process, so adding more guidance steps provides more guidance towards lower force and smaller energy above ground state.

## H.3 CHEMGUIDE VS. NEURAL GUIDANCE ON FORCE

In this section, we replace the non-differentiable oracle (i.e. xTB) guidance with a neural network that predicts energy and force. We employ a machine learning interatomic potential (MLIP), AIMNet2 (Anstine et al., 2024), a recent neural network model trained on 20 million hybrid quantum chemical calculations to efficiently predict energy, force, and properties. By combining ML-parameterized short-range and physics-based long-range terms, AIMNet2 achieves better generalization across diverse molecules. We sample 500 molecules from QM9 and 50 molecules from GEOM, and present the results in Table 10 for QM9 and Table 11 for GEOM, for reference and comparison, we also present the metrics of CHEMGUIDE with the best scales.

Using AIMNet2 as guidance doesn't improve over xTB in terms of time, because it introduces a prohibitive memory constraint that makes sampling almost linear. Recall from Section 2, the molecule position is represented as $[N, 3]$; thus, the force is $[N, 3]$, and the derivative (i.e. Hessian) of the force on the position is $[N, 3, N, 3]$, which is required when calculating the guidance gradient via backpropagation. It requires a large amount of GPU memory and is problematic when sampling large molecules (e.g., GEOM). In CHEMGUIDE, only force is required to estimate gradient, and 100 molecules are sampled at a time on a 48GiB GPU, however, with AIMNet2 guidance only 5 molecules can be sampled at a time on the same machine. This is because calculating Hessian scales up the memory by $N \times 3$, which is a major drawback of using neural networks on force guidance as it requires more memory.

Table 9: Metrics of 500 generated molecules from *GEOM* using GeoLDM with non-differentiable oracle (i.e. xTB) guidance with various guidance steps (100, 200, 300, 400) and guidance scales (0.0001, 0.001, 0.01, 0.1, 1.0). * and **bold** denote the overall best result and our best result, respectively. Percentage changes between our results and GeoLDM are shown in parentheses.

| Metric | Guidance Scale (100 Guidance Steps) | | | | | Baseline | |
|---|---|---|---|---|---|---|---|
| | 0.0001 | 0.001 | 0.01 | 0.1 | 1.0 | EDM | GeoLDM |
| Force RMS (Eh/Bohr) | 0.0434 | 0.0434 | 0.0432 | 0.0430 | **0.0414*** (-13.36%↓) | 0.0742 | 0.0478 |
| Validity | 49.20% | 49.40% | 49.20% | **50.00%*** (1.00%↑) | 48.00% | 46.40% | 49.00% |
| Uniqueness | **100.00%*** | **100.00%*** | **100.00%*** | **100.00%*** | **100.00%*** | 100.00%* | 100.00%* |
| Atom Stability | 84.70% | **84.70%*** (0.17%↑) | 84.68% | 84.67% | 84.67% | 81.22% | 84.53% |
| Energy above Ground State (Eh) | 0.2043 | 0.2039 | **0.2031*** (-9.66%↓) | 0.2049 | 0.2037 | 0.3742 | 0.2248 |
| Metric | Guidance Scale (200 Guidance Steps) | | | | | Baseline | |
| | 0.0001 | 0.001 | 0.01 | 0.1 | 1.0 | EDM | GeoLDM |
| Force RMS (Eh/Bohr) | 0.0433 | **0.0425*** (-11.13%↓) | 0.0445 | 0.0433 | 0.0476 | 0.0742 | 0.0478 |
| Validity | 48.80% | 48.40% | 50.20% | 47.80% | **51.00%*** (2.00%↑) | 46.40% | 49.00% |
| Uniqueness | **100.00%*** | **100.00%*** | **100.00%*** | **100.00%*** | **100.00%*** | 100.00%* | 100.00%* |
| Atom Stability | 84.31% | 84.28% | 84.28% | **84.45%*** (-0.08%↓) | 83.91% | 81.22% | 84.53% |
| Energy above Ground State (Eh) | 0.2224 | 0.2181 | 0.2254 | **0.2142*** (-4.72%↓) | 0.2297 | 0.3742 | 0.2248 |
| Metric | Guidance Scale (300 Guidance Steps) | | | | | Baseline | |
| | 0.0001 | 0.001 | 0.01 | 0.1 | 1.0 | EDM | GeoLDM |
| Force RMS (Eh/Bohr) | 0.0462 | 0.0468 | 0.0468 | **0.0440*** (-7.96%↓) | 0.0478 | 0.0742 | 0.0478 |
| Validity | 47.20% | 48.50% | 48.60% | **51.80%*** (2.80%↑) | 50.00% | 46.40% | 49.00% |
| Uniqueness | **100.00%*** | **100.00%*** | **100.00%*** | **100.00%*** | **100.00%*** | 100.00%* | 100.00%* |
| Atom Stability | 83.73% | 83.59% | 83.75% | **83.80%*** (-0.73%↓) | 83.38% | 81.22% | 84.53% |
| Energy above Ground State (Eh) | 0.2190 | 0.2248 | 0.2223 | **0.2071*** (-7.88%↓) | 0.2520 | 0.3742 | 0.2248 |
| Metric | Guidance Scale (400 Guidance Steps) | | | | | Baseline | |
| | 0.0001 | 0.001 | 0.01 | 0.1 | 1.0 | EDM | GeoLDM |
| Force RMS (Eh/Bohr) | 0.0445 | 0.0447 | 0.0434 | **0.0411*** (-14.16%↓) | 0.0513 | 0.0742 | 0.0478 |
| Validity | 47.00% | 45.20% | **51.60%*** (2.60%↑) | 50.40% | 49.40% | 46.40% | 49.00% |
| Uniqueness | **100.00%*** | 100.00% | 100.00% | 100.00% | 100.00% | 100.00%* | 100.00%* |
| Atom Stability | 84.47% | 84.49% | **84.65%*** (0.12%↑) | 84.36% | 83.45% | 81.22% | 84.53% |
| Energy above Ground State (Eh) | 0.2148 | 0.2085 | 0.2104 | **0.1935*** (-13.92%↓) | 0.2650 | 0.3742 | 0.2248 |

For QM9, we sample 500 molecules, and using AIMNet2 as guidance yields much worse results in all metrics than CHEMGUIDE. For GEOM, we sample 50 molecules due to the memory constraint, and AIMNet2 achieves compatible results as CHEMGUIDE for guidance, where the force RMS, validity, and stability are close.

However, on GEOM, CHEMGUIDE achieves much lower energy above ground state compared to AIMNet2 guidance, indicating that CHEMGUIDE generated molecules closer to their corresponding ground states. Recall that in the energy curve of a molecule, the x-axis and the y-value refer to the molecule configuration and its energy, where the derivative of energy gives the force, and we aim to optimize the molecule towards its ground state (i.e. the global minimum of the energy curve). Hence, although AIMNet2 guidance reduces the force similarly to CHEMGUIDE, it is farther from the global energy minimum, which demonstrates that guidance directly from quantum chemistry provides more intrinsic optimization than neural networks. Additionally, we sample 500 molecules in Table 2 while we only sample 50 molecules for AIMNet, which might not be comparable as smaller populations are more likely to be biased.

To better understand the difference, we plot the cosine similarities of the gradients from AIMNet2 and CHEMGUIDE over 400 guidance steps on 10 molecules from QM9, as shown in Figure 6. The results show that the cosine similarities are small and concentrated around zero, suggesting that the gradients from AIMNet2 and CHEMGUIDE are nearly orthogonal. This observation helps explain the significant performance differences reported in Table 10: CHEMGUIDE achieves much better results than AIMNet2-guided optimization on QM9 because CHEMGUIDE provides more effective and stable guidance, whereas the guidance from AIMNet2 negatively impacts stability.

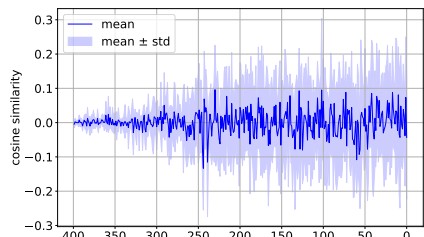

Figure 6: Cosine similarity between the guidance gradients of AIMNet and CHEMGUIDE of 10 molecules sampled on QM9 for the last 400 steps.

Table 10: Metrics of 500 generated molecules from *QM9* using GeoLDM with neural network guidance (i.e. AIMNet2) on force. * and **bold** denote the overall best result and our best result, respectively. Percentage changes between our results and GeoLDM are shown in parentheses. For ease of comparison, we report the results using the best scale of CHEMGUIDE from Table 1.

| Metric | Guidance Scale | | | | | CHEMGUIDE | Baseline | |
|---|---|---|---|---|---|---|---|---|
| | 0.0001 | 0.001 | 0.01 | 0.1 | 1.0 | scale = 0.0001 | EDM | GeoLDM |
| Force RMS (Eh/Bohr) | 0.0113 | 0.0113 | 0.0112 | 0.0114 | 0.0110 (-1.00%↓) | **0.0104*** (-6.76%↓) | 0.0114 | 0.0111 |
| Validity | 89.20% (-0.60%↓) | 89.20% (-0.60%↓) | 89.00% | 88.60% | 89.00% | **91.40%*** (1.60%↑) | 86.60% | 89.80% |
| Uniqueness | **100.00%*** | **100.00%*** | **100.00%*** | 100.00%* | 100.00%* | **100.00%*** | 100.00%* | 100.00%* |
| Atom Stability | 98.72% | 98.72% | 98.71% | 98.74% | 98.75% (-0.18%↓) | **99.02%*** (0.09%↑) | 98.53% | 98.93% |
| Molecule Stability | 87.80% | 87.80% | 87.60% | 87.80% | 88.00% (-0.80%↓) | **90.60%*** (1.80%↑) | 83.60% | 88.80% |
| Energy above ground state (Eh) | 0.0056 | 0.0057 | 0.0056 | 0.0059 | 0.0055 (9.30%↑) | **0.0042*** (-15.78%↓) | 0.0072 | 0.0050 |

Table 11: Metrics of 50 generated molecules from *GEOM* using GeoLDM with neural network guidance (i.e. AIMNet2) on force. * and **bold** denote the overall best result and our best result, respectively. Percentage changes between our results and GeoLDM are shown in parentheses. For ease of comparison, we report the results using the best scale of CHEMGUIDE from Table 2.

| Metric | Guidance Scale | | | | | CHEMGUIDE | Baseline | |
|---|---|---|---|---|---|---|---|---|
| | 0.0001 | 0.001 | 0.01 | 0.1 | 1.0 | scale = 0.1 | EDM | GeoLDM |
| Force RMS (Eh/Bohr) | 0.0418 | 0.0418 | 0.0418 | 0.0409 | **0.0406*** (-15.14%↓) | 0.0411 (-14.16%↓) | 0.0742 | 0.0478 |
| Validity (xTB) | 50.00% (1.00%↑) | 50.00% | 50.00%* | 50.00% | 50.00% | **50.40%*** (1.40%↑) | 46.40% | 49.00% |
| Uniqueness | **100.00%*** | **100.00%*** | **100.00%*** | 100.00%* | 100.00%* | **100.00%*** | 100.00%* | 100.00%* |
| Atom Stability | 85.01% | 85.01% | 85.01% | 85.13% | **85.26%*** (0.73%↑) | 84.36% (-0.17%↓) | 81.22% | 84.53% |
| Energy above ground state (Eh) | 0.2392 | 0.2392 | 0.2392 | 0.2356 | 0.2299 (2.26%↑) | **0.1935***(-13.92%↓) | 0.3742 | 0.2248 |

## H.4 GENERALIZATION
### ANALYSIS OF NEURAL REGRESSORS

As observed in Table 16 in Appendix L and Table 18 in Appendix M, our results are not always stable and sometimes collapse (see the collapse analysis in Section H.6). In this section, we provide an analysis of the neural network used for gradient calculation, with the results shown in Fig. 7. We sampled 1,000 molecules from QM9 and added Gaussian noise with varying variances (1e-4, 1e-3, 1e-2, 1.0, 2.0, 3.0, 4.0) to their positions. We then evaluated the neural regressor and xTB, calculating the percentage change in predicted values from those predicted when no noise was added. For visualization purposes, we clipped the maximum percentage change to be less than 500%. The line represents the mean of the percentage changes, while the shaded area (magnitude) represents the range from maximum to minimum. As the noise scale increases, the regressors become increasingly unstable, with percentage changes reaching as high as 100%. This instability can explain the fluctuating MAE in our results, as we use the neural regressor during the generation process, where the molecules are noisier and less optimized than those in the QM9 dataset. Consequently, the regressor may provide suboptimal or even incorrect guidance during the early stages of the generation.

By comparing the performance of the regressor and xTB, it is clear that xTB provides more stable predictions, as both the mean and the magnitude of instability remain lower than those of the regressor. This helps explain the

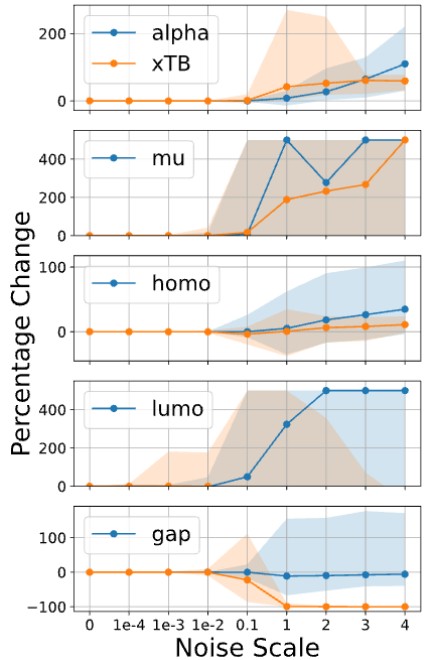

Figure 7: Percentage change of $\alpha, \mu, \epsilon_{\text{HOMO}}, \epsilon_{\text{LUMO}}, \Delta_\epsilon$ predicted by the neural regressor when adding various noise.

performance differences in Table 4 and Table 6, where CHEMGUIDE performs worse than $\alpha$: the regressor produces more stable results in $\alpha$. However, CHEMGUIDE performs better in $\Delta\epsilon$ because the regressor becomes more stable at larger noise scales, and xTB, on average, begins to "prune" invalid molecules when the noise scale is large, ceasing to provide guidance (i.e., gradients) to these invalid molecules.

This behavior is evident from the "peak" in xTB prediction magnitude around noise scale = 1.0, after which the magnitude quickly decays and converges to the mean, while the magnitude of the regressor's performance continues to increase. This trend is more obvious in $\Delta\epsilon$, where a -100% change is observed, indicating that xTB either throws an error on invalid molecules, in which case the gradient is 0, or returns negligible values for distorted molecules, and its guidance diminishes and stops further changes to molecular structures.

This validates our motivation for using a non-differentiable chemistry oracle to guide generation and optimization. A chemistry oracle, based on deterministic chemistry-rule calculations, can better capture the properties of molecules, even for previously unseen ones, and will not return high scores on invalid molecules, which further distort the molecular structures.

### H.5 GENERALIZATION ANALYSIS OF CONDITIONAL GEOLDM

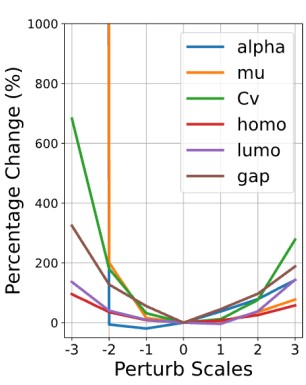

Figure 8: Percentage changes of MAE of 200 generated molecules using C-GeoLDM with various perturbed scales.

As discussed in the introduction (Section 1), explicit guidance requires training a conditional model with labels; thus, the model is likely to be not robust and does not generalize well to distribution shifts. It performs poorly when conditioned on property scores that were not encountered during training. To demonstrate this, we perturb the sampled context from the property distribution on which C-GeoLDM is conditioned and generate 200 molecules. Since the sampled property scores are standardized with z-score normalization, we shift the context by $\pm 1, \pm 2, \pm 3$ (which means we look at the $\pm 1 \cdot \sigma$, $\pm 2 \cdot \sigma$, and $\pm 3 \cdot \sigma$ regions of the property score distribution). We then plot the percentage change in MAE compared to the case with no distribution shift (i.e., perturb scale=0) in Fig. 8. For percentage changes greater than 1000%, we clip the values (for example, in the case of $\mu$, the percentage change exceeds 1000%, so it is not fully displayed). The percentage change in MAE can be as high as 50% even within the standard deviation of $\pm 1 \cdot \sigma$, and it increases significantly when the distribution is shifted by $\pm 2\sigma$, which suggests C-GeoLDM is prone to be affected by distribution shifts; thus, unlikely to generalize well when conditioned on scores beyond the property distribution seen during training.

### H.6 ANALYSIS OF GRADIENT COLLAPSE

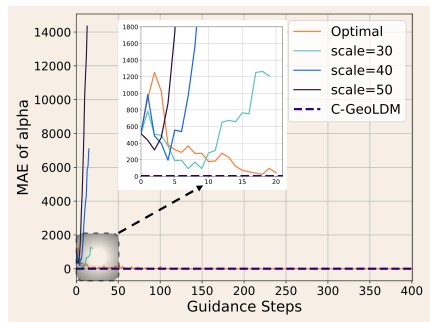

Figure 9: Gradient collapse of $\alpha$ using GeoLDM with noisy neural guidance.

It's observed in Table. 16 in Appendix L that some gradients of noisy neural guidance and clean neural guidance collapse for some scales. For example, scale 5, 20, 25, 30 when optimizing $\alpha$, so we explain why such collapses happen in this section via a case study of optimizing $\alpha$ with scale 10 (optimal scale with no collapse), 30, 40, and 50. As we increase the scale, the MAE grows quickly in the first 50 steps and the molecules collapse because the added gradient is so high that it pulls the atom apart and distorts the molecule structure. Hence, the guidance scale for each property must be carefully searched and selected so that it provides strong enough guidance while maintaining a valid molecule structure.

### H.7 MORE CLEAN GUIDANCE OPTIMIZATION STEPS

Recall that the clean guidance is provided in Eq.10, where we use K steps of gradient descent (GD). In this section, we experiment with more GD steps. For each property, we select the scale that results in the lowest MAE without collapse, as shown in Table16. For example, we choose a scale of 0.1 for $\alpha$. The results are presented in Fig. 10.

To investigate the effect of additional recurrent steps and to prevent a single collapse from stopping the entire generation process, we divide the molecules into groups of 50 and calculate the MAE on a group-wise basis. In the figure, "High" indicates that the MAE is too large to be displayed properly, and a circle (○) means that the MAE is either too low or too high; thus, it is classified as

an outlier. For instance, in the case of $\mu$, most MAEs are high when GD step=5, except for one group with an MAE of 0.12. In terms of the lowest MAE a group can achieve, using more GD steps tends to decrease the MAE, as observed for $\alpha$ and $\Delta\epsilon$, where the lowest MAE is reduced by approximately 30% for $\alpha$ and 55% for $\mu$ (even though the lowest MAE for $\mu$ is already small, it can still be significantly reduced from 0.20 with GD step=1 to 0.09 with GD steps=3). However, in terms of average MAE, since clean guidance inherently has higher instability during generation (with details and analysis of this instability available in Fig.3 and Section L), adding more GD steps amplifies this effect. This is most evident in the case of $C_v$, where all groups with GD steps=3 and 10 collapses, and the MAE becomes exceptionally high when GD=5. Moreover, adding more steps does not guarantee a monotonically better MAE, as seen in $\epsilon_{LUMO}$, where 3 steps yield the optimal result while steps=5 increase the variance and mean. Therefore, selecting the number of GD steps and guidance scales carefully for each property is essential, which we leave for future work.

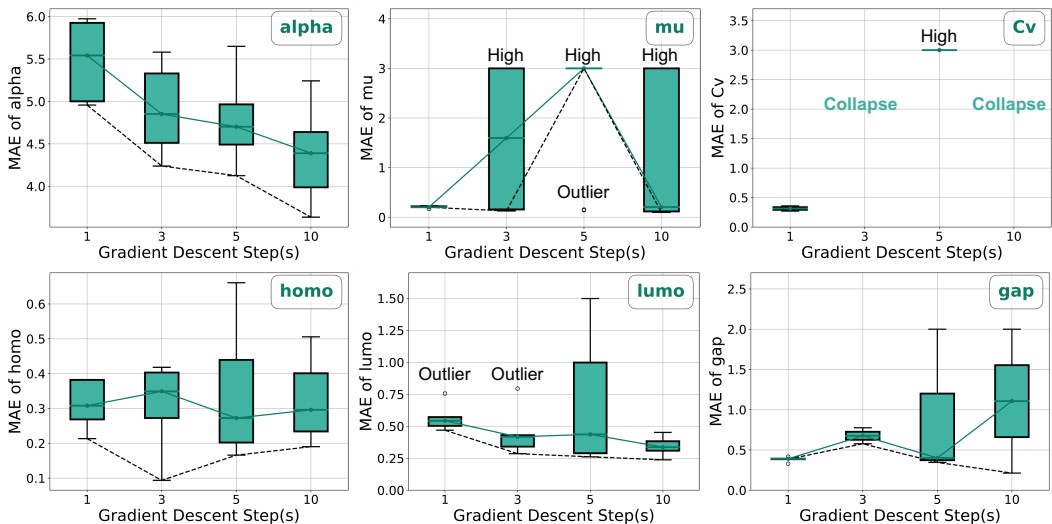

Figure 10: Box plots of MAEs of 500 generated molecules using clean guidance with more GD steps to optimize all 6 properties. The first, second (median), and third quantiles are plotted.

## I   OTHER OPTIMIZATION ALGORITHMS

In this section, we compare our proposed CHEMGUIDE with an alternative class of gradient-free methods that incorporate non-differentiable oracles into the diffusion process: evolutionary algorithms (Schneuing et al., 2022; Huang et al., 2024). The evolutionary algorithm, detailed in Algorithm 2, has two main parameters: the variant size ($k$) and the evolution interval ($E$). During the diffusion process, the population is preserved and evolution is performed at fixed intervals. At each evolution step, $k - 1$ noise perturbations are added to the population, resulting in $k$ variants for each molecule, where the unperturbed molecule is treated as a variant. The best variant is then selected as the new population based on evaluations from the non-differentiable oracle (i.e. xTB). Specifically, the oracle calculates the force RMS, and the variant with the lowest force RMS is selected.

We explored several combinations of variant size and evolution interval on QM9, and the results are summarized in Table 12. For ease of comparison, the results using the best scales of CHEMGUIDE are also provided in Table 1. We can observe CHEMGUIDE is on par with the evolutionary algorithm in validity and stability, and significantly outperforms it in terms of force RMS and energy above the ground state. This advantage can be attributed to the fact that gradients provide informative guidance by indicating the correct direction for optimizing stability. In contrast, the evolutionary algorithm operates without relying on gradients, which results in a lack of directional guidance. Consequently, our approach enables controllable optimization with this gradient-based guidance. While the evolutionary algorithm selects the best variant at each evolutionary step, its optimization direction is essentially random, making the process less controllable. Additionally, the gradient inherently adjusts the direction of the generative process, whereas the evolutionary algorithm does not

---

**Algorithm 2** CHEMGUIDE diffusion sampling with *evolutionary algorithm*

---

**Input**: a latent diffusion model $\epsilon_\theta$, a VAE decoder $\mathcal{D}$, a composition function $\mathcal{F}$, target property score $y$, variant size $k$, variant scale $s_v$, evolution interval $E$.
**Output**: optimized molecule $[\mathbf{x}, \mathbf{h}]$
$\boldsymbol{z}_T \leftarrow \mathcal{N}(0, \mathbf{I})$
$population \leftarrow [\boldsymbol{z}_t]$
**for all** $t$ from $T$ to 1 **do**
    // Add variants
    **if** $t \% E = 0$ **then**
        $population$.extend($[\boldsymbol{z}_t + s_v \cdot \epsilon_1, \boldsymbol{z}_t + s_v \cdot \epsilon_2, \cdots, \boldsymbol{z}_t + s_v \cdot \epsilon_{k-1}]$), where $\epsilon_1, \cdots, \epsilon_{k-1} \sim \mathcal{N}(0, \mathbf{I})$
    **end if**
    **for all** $\boldsymbol{z}_{t,i} \in population$ **do**

$$\mu_{t-1,i}, \Sigma_{t-1,i} \leftarrow \frac{1}{\sqrt{1-\beta_t}}(\boldsymbol{z}_{t,i} - \frac{\beta_t}{\sqrt{1-\alpha_t^2}}\epsilon_\theta(\boldsymbol{z}_{t,i}, t)), \sigma_t^2 \mathbf{I} \qquad \text{(Eq. 3)}$$

$$\boldsymbol{z}_{t-1,i} \leftarrow \mathcal{N}(\mu_{t-1,i}, \Sigma_{t-1,i}) \qquad \text{(Eq. 6)}$$

        Replace $\boldsymbol{z}_{t,i}$ in $population$ by $\boldsymbol{z}_{t-1,i}$
    **end for**
    $\boldsymbol{z}_{t-1}^* \leftarrow \arg\min_{\boldsymbol{z}_{t-1} \in population} \|\mathcal{F}(\boldsymbol{z}_{t-1})\|^2$      // Select the best variant
    $population \leftarrow [\boldsymbol{z}_{t-1}^*]$
**end for**
$[\mathbf{x}, \mathbf{h}] \leftarrow \mathcal{D}(\boldsymbol{z}_0^*)$
**return** $[\mathbf{x}, \mathbf{h}]$

---

Table 12: Metrics of 500 generated molecules from *QM9* using GeoLDM with evolutionary algorithm (Algorithm 2). $k, E$ denote the variant size and evolution interval. * and **bold** denote the overall best result and our best result, respectively. Percentage changes between our results and GeoLDM are shown in parentheses. For ease of comparison, we report the results using the best scale of CHEMGUIDE from Table 1.

| Metric | (variant size, evolution interval) | | | | CHEMGUIDE | Baseline | |
|---|---|---|---|---|---|---|---|
| | $(k=3, E=20)$ | $(k=3, E=50)$ | $(k=5, E=20)$ | $(k=5, E=50)$ | scale = 0.0001 | EDM | GeoLDM |
| Force RMS (Eh/Bohr) | 0.0112 | 0.0111 | 0.0110 | 0.0109 (-1.75%↓) | **0.0104**\* (-6.76%↓) | 0.0114 | 0.0111 |
| Validity | 91.60% | 87.60% | 90.00% | **92.40%**\* (2.60%↑) | 91.40% (1.60%↑) | 86.60% | 89.80% |
| Uniqueness | **100.00%**\* | **100.00%**\* | 99.78% | **100.00%**\* | **100.00%**\* | 100.00%\* | 100.00%\* |
| Atom Stability | 98.95% | 98.76% | 98.66% | **99.14%**\* (0.21%↑) | 99.02% (0.09%↑) | 98.53% | 98.93% |
| Molecule Stability | 90.00% | 87.60% | 88.00% | **91.20%**\* (2.40%↑) | 90.60% (1.80%↑) | 83.60% | 88.80% |
| Energy above ground state (Eh) | 0.0048 (-4.45%↓) | 0.0053 | 0.0056 | 0.0054 | **0.0042**\* (-15.78%↓) | 0.0072 | 0.0050 |

dynamically adapt its direction. Therefore, our CHEMGUIDE offers a more generic and inherently guided optimization process.

## J  BILEVEL OPTIMIZATION WITH CLEAN GUIDANCE

As shown in Section 4.3.2, our clean guidance method can achieve significantly lower MAE than the baseline. However, its ability to drastically lower MAE also contributes to unstable molecular geometries. To illustrate this, we pick two properties as a case study from Table 4 which are easiest ($\mu$) and hardest ($\alpha$) to optimize using clean guidance, and the results are shown in Table 13. Comparing the MAE and force RMS across two experiments in each property optimization, we observe that the bilevel optimization framework helps reduce force RMS and MAE at the same time, indicating that optimizing force/property yields better results on the other. However, we can observe that the force RMS of bilevel optimization is still higher than the baseline (right block of Table 13, with 200 molecules generated), but this is because the force RMS of clean guidance is much higher than the baseline (left block of Table 13, with 500 molecules generated); thus, CHEMGUIDE in bilevel optimization can't minimize the force significantly to beat the baseline, which is the natural hardness of optimizing force after clean guidance as the clean guidance is so strong that it significantly changes the molecular structures, making it challenging for xTB to optimize. We notice a trade-off between force and property optimization when comparing the performance across $\alpha$ and $\mu$ in bilevel optimization: lower force yields higher MAE while lower MAE brings higher force. Hence, given

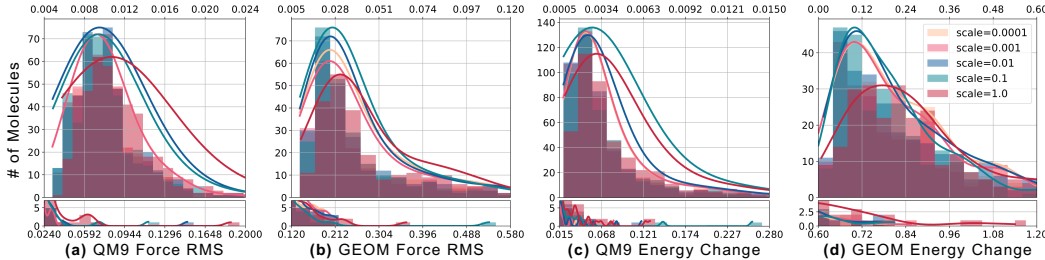

Figure 11: Histograms and distributions of force RMS and energy above the ground state of 500 generated molecules from *QM9 dataset* using unconditional GeoLDM with CHEMGUIDE using various scales. Energy change refers to energy above ground state.

the difficulty of optimizing force and MAE at the same time to beat the baseline, it's important and tricky to pick the best scales or schedules for force and property optimization, which we leave to future work. Similarly, we report full numerical results of the case study on $\alpha$ and $\mu$ optimization to Table 20 in Appendix M.

Table 13: Comparisons of force RMS and MAE of generated molecules from *QM9* using baselines, unconditional GeoLDM with clean neural guidance, bilevel optimization with clean neural guidance.

| | Models | Clean Neural Guidance | | | | Bilevel w/ Clean Neural Guidance | | | |
|---|---|---|---|---|---|---|---|---|---|
| | | $\alpha$ (Bohr$^3$) | | $\mu$ (D) | | $\alpha$ (Bohr$^3$) | | $\mu$ (D) | |
| | | Force RMS (Eh/Bohr) | MAE | Force RMS (Eh/Bohr) | MAE | Force RMS (Eh/Bohr) | MAE | Force RMS (Eh/Bohr) | MAE |
| Baselines | C-EDM | 0.0118 | 2.6308 | 0.0122 | 1.1257 | 0.0125 | 2.5089 | 0.0123 | 1.0571 |
| | C-GeoLDM | 0.0117 | 2.5551 | 0.0120 | 1.1084 | 0.0135 | 2.5593 | 0.0123 | 1.1582 |
| Ours | | 0.0326 | 4.9992 | 0.3962 | 0.4025 | 0.0274 | 4.2058 | 0.1348 | 0.3150 |

## K    COMPREHENSIVE RESULTS ON NON-DIFFERENTIABLE ORACLE GUIDANCE

We report the distributions of force RMS and energy above ground state of 500 generated molecules from QM9 dataset and GEOM dataset using unconditional GeoLDM with CHEMGUIDE using various scales (0.0001, 0.001, 0.01, 0.1, and 1.0) in Fig. 11, numerical results are shown in Table 1 and Table 2 in Section 4.2. We observe that smaller scales produce better results while larger scales (i.e. 1.0) produce worse results than the baselines. This is because the position values of the molecules are small (less than 1.0); thus, when the scale of the gradients is similar to position values, adding too much guidance results in "over-optimizing" the position and distorting the molecule structures. This can be validated in Fig. 11: as the scale increases, the molecules become less concentrated around 0, spreading out and exhibiting higher energy above ground states and force RMS. The distribution shifts downward, indicating lower validity, and its peak moves away from 0, suggesting reduced stability. The performance, worse than the baselines, implies that the molecular structures are being distorted due to excessive guidance. The results for GeoLDM with CHEMGUIDE on properties are presented in Table 14.

## L    COMPREHENSIVE RESULTS ON NEURAL GUIDANCE

We present full numeric results of **unconditional** GeoLDM using noisy and clean neural guidance with various scales in Table 16. It is observed that larger scales yield better results while smaller scales produce results closer to unconditional GeoLDM. This is intuitive because larger scales provide more guidance to the generation, which brings better properties; on the other hand, since we are using unconditional GeoLDM as the backbone, the guidance becomes more negligible when the scale is small and thus the performance converges to unconditional GeoLDM. ✗ in Table 16 means that the molecules collapse in the middle of the generation, so there are no output molecules. Explanation and case study of collapse can be found in Section H.6 and Section H.4. To better investigate the effect of larger scales, we split 500 generated molecules into 10 groups so that a single

Table 14: MAE of 200 generated molecules on *QM9* using GeoLDM with CHEMGUIDE on all six properties, with GeoLDM as the baseline. ∗ and **bold** denote the overall best and the best within different scales. Percentage changes between our results and GeoLDM are shown in parentheses.

| Property | $\alpha$ | $\mu$ | $\epsilon_{\text{HOMO}}$ | $\epsilon_{\text{LUMO}}$ | $\Delta\epsilon$ |
|---|---|---|---|---|---|
| Units | Bohr$^3$ | D | meV | meV | meV |
| GeoLDM | 4.7555∗ | 1.5490 | 0.6264∗ | 2.1790 | 2.1568 |
| GeoLDM w/ CHEMGUIDE scale = $1.0 \times 10^{-4}$ | 4.9045 | 1.5335 | 0.7228 | **2.0091**∗ (-7.80%↓) | 1.9321 |
| scale = $1.0 \times 10^{-3}$ | 4.8790 | 1.5311 | 0.7182 | 2.0183 | **1.9318**∗ (-10.43%↓) |
| scale = $1.0 \times 10^{-2}$ | **4.8781** (2.58%↑) | **1.5295**∗ (-1.26%↓) | 0.7449 (150) | 2.0548 | 1.9428 |
| scale = $1.0 \times 10^{-1}$ | 4.9025 | 1.5979 | 0.6961 | 2.0136 | 1.9607 |
| scale = $1.0 \times 10^{+0}$ | 5.2251 | 1.6631 | **0.6392** (2.04%↑) | 2.0298 | 2.0416 |

Table 15: MAE of 200 generated molecules on *GEOM* using GeoLDM with CHEMGUIDE on all six properties, with GeoLDM as the baseline. ∗ and **bold** denote the overall best and the best within different scales. Percentage changes between our results and GeoLDM are shown in parentheses.

| Property | $\alpha$ | $\mu$ | $\epsilon_{\text{HOMO}}$ | $\epsilon_{\text{LUMO}}$ | $\Delta\epsilon$ |
|---|---|---|---|---|---|
| Units | Bohr$^3$ | D | meV | meV | meV |
| GeoLDM | 24.3424 | 3.3676 | 1.1922∗ | 5.0903 | 1.9956 |
| GeoLDM w/ CHEMGUIDE scale = $1.0 \times 10^{-4}$ | 26.2032 | 3.3632 | **1.2939** (8.53%↑) | **5.0802**∗ (-0.20%↓) | 1.9381 |
| scale = $1.0 \times 10^{-3}$ | 26.1301 | 3.3881 | 1.2974 | 5.1477 | 1.9616 |
| scale = $1.0 \times 10^{-2}$ | 25.9853 | **3.3412**∗ (-0.78%↓) | 1.3032 | 5.2268 | **1.9264**∗ (-3.47%↓) |
| scale = $1.0 \times 10^{-1}$ | 25.8031 | 3.4581 | 1.3002 | 5.3704 | 1.9445 |
| scale = $1.0 \times 10^{+0}$ | **24.2168**∗ (-0.52%↓) | 3.7300 | 1.2992 | 5.1828 | 2.1446 |

collapse will not stop the generation of all molecules, we put the remaining number of molecules in the parenthesis in Table 16 and report the MAE of the molecules in groups without collapse. For example, 3.2309 (450) in $\alpha$ of scale=5 means there are 450 molecules in groups with no collapse and their MAE is 3.2309.

We plot the MAE trajectories of optimizing all six properties in Fig. 12. Applying noisy neural guidance results in much smoother trajectories compared to the baseline. It reduces MAE significantly during the first 50 epochs and then stabilizes, making noisy guidance most effective in the early stages of generation. Clean neural guidance also lowers MAE in the first 100 steps to some extent, but it fluctuates more than noisy guidance (and sometimes more than the baseline, as in the trajectory of optimizing $\epsilon_{LUMO}$). However, clean guidance becomes more powerful in the final 50 steps, where we observe a sharp drop in MAE across all six properties. The difference in smoothness is because, in noisy guidance, the gradient is propagated through the denoising estimation (Eq.8) and the VAE decoder (Algorithm2), inherently accounting for the effect of noise in the gradient calculation, which leads to consistent guidance throughout the generation steps. On the contrary, in clean guidance, the gradient is computed in the clean space (Eq.11) and then projected back into the generation process by adding noise (Eq.10). Since the gradient does not propagate through the denoising estimation and VAE decoder, it is more sensitive to noise, resulting in greater fluctuations and inconsistency during guidance. Noisy guidance stabilizes and doesn't exhibit a sharp drop in later steps because the loss decreases as the MAE reduces in the early stages, so the gradient is small, which mainly serves to maintain the current molecular structure, preventing the MAE from converging to that of GeoLDM. This can be observed in Table 16, where the MAE for smaller scales is closer to that of unconditional GeoLDM. Although clean guidance helps reduce MAE in earlier steps of generation, it reaches its full potential in the later steps, when the molecules are more similar and close to decoded molecules, leading to a sharp drop in MAE.

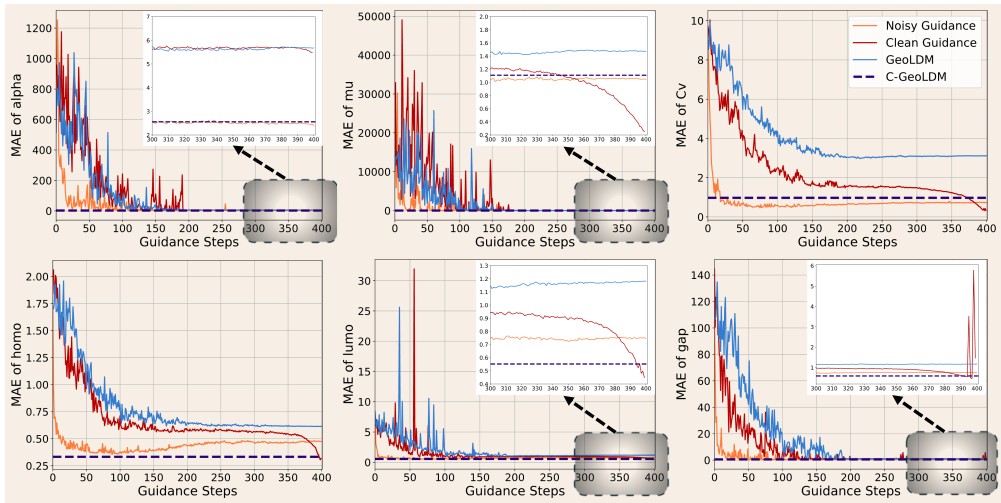

Figure 12: MAE trajectories of 500 generated molecules using GeoLDM and **GeoLDM with clean&noisy guidance** with optimal configuration. We use the scales with no collapse as optimal scales; for example, the optimal scale of optimizing $\alpha$ is 10.0 instead of 25.0.

## M    COMPREHENSIVE RESULTS ON BILEVEL OPTIMIZATION

In this section, we present MAE trajectories and detailed results of force metrics and MAE for explicit bilevel optimization and implicit bilevel optimization with noisy guidance, and we provide full numeric results of MAE and force metrics of implicit bilevel optimization with clean guidance. Recall that we only sample 200 molecules for experiments in this section, as mentioned in Section 4.4.

**Explicit guided diffusion with CHEMGUIDE**    The MAE and force metrics are shown in Table 18 and Table 17 respectively. Since we only add CHEMGUIDE to C-GeoLDM, our expectation is that we can achieve lower forces without increasing MAE. However, the results go beyond our expectations: our method not only achieves better force RMS, energy above ground state, and validity but also reduces MAE in the first 5 properties. This shows the effectiveness of combining explicit property optimization with CHEMGUIDE, and verifies the feasibility of our bilevel optimization framework.

**Implicit guided diffusion with noisy guidance and CHEMGUIDE**    The MAE and force metrics of bilevel optimization with noisy guidance are shown in Table 18 and Table 19 respectively. For ease of searching scales for noisy guidance and to better explore larger scales, we employ an ensemble method: based on the results from unconditional GeoLDM with noisy guidance in Table 16, we pick the **top-4** scales that give the lowest MAE, and then sample 50 molecules using each scale and aggregate the results. For instance, for $\alpha$, we pick scales of 25.0 (Top 1), 20.0 (Top 2), 10.0 (Top 3), and 5.0 (Top 4). Overall, we achieve better force metrics than the baselines. For each property, there exists at least one scale with a large improvement in force metrics; for $\alpha$, $\mu$, and $C_v$, we can find scales that produce improvement in both force and MAE such as scale=20 for $\alpha$, scale=20.0 for $\mu$, and scale=25 for $C_v$, indicating that our bilevel optimization with noisy guidance does succeed in optimizing both property and force. The results for MAE are consistent with GeoLDM with noisy guidance in the sense that top scales in general yield lower MAE and, the same as Table 16, our bilevel optimization with noisy guidance performs better than the baseline in $\alpha$, $\mu$, and $C_v$ while worse than the baseline in the other three properties. Moreover, the percentage change of the best scale in both tables are similar yet bilevel optimization with noisy guidance yields slightly worse results than GeoLDM with noisy guidance, which suggests adding guidance for force will slightly contradict the guidance for MAE as the the guidance tries to find a balance between property and stability optimization.

To illustrate how bilevel optimization works, we provide MAE trajectories of each property in Fig. 13. Compared to MAE trajectories in unconditional GeoLDM with noisy guidance in Fig. 12,

Table 16: MAE of all six properties of 500 sampled molecules from **GeoLDM with noisy and clean neural guidance**. We use C-EDM, C-GeoLDM, and GeoLDM as baselines. ∗ and **bold** denote the overall best and the best within different scales, respectively. ✗ marks the situation where the guidance collapses. Percentage changes between our results and C-GeoLDM are shown in parenthesis

| Property | $\alpha$ | $\mu$ | $C_v$ | $\epsilon_{\text{HOMO}}$ | $\epsilon_{\text{LUMO}}$ | $\Delta\epsilon$ |
|---|---|---|---|---|---|---|
| Units | Bohr$^3$ | D | $\frac{\text{cal}}{\text{mol}}$K | meV | meV | meV |
| Conditional EDM | 2.6308 | 1.1257 | 1.0804 | 0.3207 | 0.5940 | 0.6301 |
| Conditional GeoLDM | 2.5551 | 1.1084 | 0.9703 | 0.3327 | 0.5518 | 0.5878 |
| GeoLDM | 5.6732 | 1.6461 | 3.1046 | 0.6151 | 1.1778 | 1.2022 |
| **GeoLDM w/ Noisy Guidance** scale = $1.0 \times 10^{-4}$ | 5.6890 | 1.4784 | 3.1046 | 0.6136 | 1.1834 | 1.1725 |
| scale = $1.0 \times 10^{-3}$ | 5.6703 | 1.4743 | 3.1002 | 0.6155 | 1.1864 | 1.1683 |
| scale = $1.0 \times 10^{-2}$ | 5.6547 | 1.4669 | 3.0923 | 0.6190 | 1.1739 | 1.1770 |
| scale = $1.0 \times 10^{-1}$ | 5.1373 | 1.5102 | 2.7785 | 0.6183 | 1.1509 | 1.1319 |
| scale = $1.0 \times 10^{+0}$ | 4.1372 | 1.3248 | 1.7806 | 0.5965 | 1.0523 | 1.1033 |
| scale = $2.0 \times 10^{+0}$ | 3.6602 | 1.2153 | 1.5683 | 0.5763 | 0.9916 | 1.0504 |
| scale = $5.0 \times 10^{+0}$ | 3.2309 (450) | 1.0331 | 1.2369 | 0.5644 | 0.9834 | 1.0362 |
| scale = $1.0 \times 10^{+1}$ | 2.3870 | 0.8201 (450) | 1.0358 | 0.5667 | 0.8554 | 0.9102 |
| scale = $2.0 \times 10^{+1}$ | 1.9312 (400) | 0.6251 (350) | 0.8804 | 0.4735 | 0.8061 | 0.8325 |
| scale = $2.5 \times 10^{+1}$ | **1.5284 (100)**∗ (-40.1824%↓) | 0.5770 (300) | 0.7112 | 0.4796 | 0.7436 | 0.8596 |
| scale = $3.0 \times 10^{+1}$ | ✗ | **0.4733 (250)** (-57.2988%↓) | **0.7094 (450)** (-26.8886%↓) | 0.4725 | 0.8115 | 0.7468 |
| scale = $4.0 \times 10^{+1}$ | ✗ | ✗ | ✗ | **0.4153 (450)** (24.8272%↑) | 0.7496 | **0.7101 (200)** (20.8064%↑) |
| scale = $5.0 \times 10^{+1}$ | ✗ | ✗ | ✗ | 0.4325 (300) | **0.7015 (150)** (27.1294%↑) | ✗ |
| **GeoLDM w/ Clean Guidance** scale = $1.0 \times 10^{-4}$ | 5.7236 | 1.3925 | 2.9780 | 0.6380 | 1.1587 | 1.2056 |
| scale = $1.0 \times 10^{-3}$ | 5.7373 | 1.3914 | 2.9779 | 0.6374 | 1.1578 | 1.2055 |
| scale = $1.0 \times 10^{-2}$ | 5.6894 | 1.4102 | 2.9734 | 0.6367 | 1.1558 | 1.2024 |
| scale = $1.0 \times 10^{-1}$ | 5.4798 | 1.3384 | 2.9506 | 0.6343 | 1.1576 | 1.2096 |
| scale = $1.0 \times 10^{+0}$ | **2.4980 (200)** (-2.23%↓) | 0.7144 | 2.7440 | 0.6203 | 1.1292 | 1.1345 |
| scale = $2.0 \times 10^{+0}$ | 1320.7350 | 0.4475 | 2.3713 | 0.6040 | 1.0853 | 1.0609 |
| scale = $5.0 \times 10^{+0}$ | 5309.9470 (200) | 0.2031 | 1.6211 | 0.5533 | 0.9519 | 0.8676 |
| scale = $1.0 \times 10^{+1}$ | ✗ | **0.1631 (100)**∗ (-85.29%↓) | 0.9668 | 0.4817 | 0.7369 | 0.6690 |
| scale = $2.0 \times 10^{+1}$ | ✗ | 2179.0210 (100) | 0.5311 | 0.3740 | 0.5688 | 0.4313 (300) |
| scale = $2.5 \times 10^{+1}$ | ✗ | 2010.2822 (100) | 0.4524 (250) | 0.3359 | 0.5580 (300) | 0.3830 |
| scale = $3.0 \times 10^{+1}$ | ✗ | ✗ | 0.3151 | 0.3186 | 0.5747 | **0.2852 (100)**∗ (-51.48%↓) |
| scale = $4.0 \times 10^{+1}$ | ✗ | ✗ | 0.6815 (250) | 0.3109 | **0.5374 (300)** (-2.61%↓) | 28.5487 (200) |
| scale = $5.0 \times 10^{+1}$ | ✗ | ✗ | **0.2044 (250)**∗ (-78.93%↓) | **0.2386 (300)**∗ (-28.28%) | 0.7159 (100) | 0.4831 (100) |

the trajectories of bilevel optimization with noisy guidance behave similarly in the sense that the MAE is reduced significantly during the first 50 guidance steps and stabilizes. The difference is that the trajectories of bilevel optimization with noisy guidance have slightly larger fluctuation during the early steps of guidance. We reason that in the early steps, the molecules are less stable, so CHEMGUIDE provides larger guidance such that the guidance from CHEMGUIDE and noisy guidance interact with each other and result in fluctuated trajectories. The trajectories are also more converged to GeoLDM because the generation process leans less toward a lower MAE, as it keeps a balance between property and stability optimization, which can be verified from 19.

**Implicit guided diffusion with clean guidance** Finally, we present the algorithm of bilevel optimization with clean guidance and CHEMGUIDE in Algorithm 3. The detailed results of optimizing $\alpha$ and $\mu$ are shown in Table 20. Recall that $\alpha$ is the most challenging to optimize and $\mu$ is the easiest according to Table 4. In both properties, it's obvious that increasing guidance scales results in lower MAE yet larger force RMS, which is because the effect of clean guidance exceeds that of CHEMGUIDE, so it's a trade-off between optimizing force and property. Comparing the two methods of the two properties, we notice that optimizing $\alpha$ yields better force RMS yet higher MAE while optimizing $\mu$ results in worse force RMS yet lower MAE, which is another trade-off between optimizing force and property. In each property, although we can't directly compare the performance of GeoLDM with clean guidance and bilevel optimization with clean guidance because we sample 500 molecules from the former and 200 from the latter, we can compare them with their respective baselines. We notice that by adding CHEMGUIDE on force in bilevel optimization, we

---

**Algorithm 3** Bilevel guided diffusion sampling with *clean neural guidance*

---

**Input**: a latent diffusion model $\epsilon_\theta$, a VAE decoder $\mathcal{D}$, a composition function $\mathcal{F}$, target property score $y$, oracle guidance scale $s$, SPSA perturbation $\zeta$, number of gradient descent steps $K$.
**Output**: optimized molecule $[\mathbf{x}, \mathbf{h}]$
$\boldsymbol{z}_T \leftarrow \mathcal{N}(0, \mathbf{I})$
**for all** $t$ from $T$ to 1 **do**
$\quad \hat{\boldsymbol{z}}_0 \leftarrow t_0(\boldsymbol{z}_t)$
$\quad \Delta \boldsymbol{z}_0 \leftarrow \arg\min_\Delta \|y - f_\eta(\hat{\boldsymbol{z}}_0 + \Delta)\|^2$ using $K$ steps of gradient descend $\qquad$ (Eq. 10)
$\quad \tilde{\boldsymbol{z}}_0 \leftarrow \hat{\boldsymbol{z}}_0 + \Delta \boldsymbol{z}_0$
$\quad \mathbf{U} \leftarrow \mathcal{N}(0, \mathbf{I})$
$\quad g_{t-1} \propto -\nabla_{\mathcal{F}(\tilde{\boldsymbol{z}}_0)} \|\mathcal{F}(\tilde{\boldsymbol{z}}_0)\|^2 \cdot \frac{1}{2\zeta} \left( \mathcal{F}([\tilde{\boldsymbol{z}}_{\mathbf{x},0} + \zeta\mathbf{U}, \tilde{\boldsymbol{z}}_{\mathbf{h},0}]) - \mathcal{F}([\tilde{\boldsymbol{z}}_{\mathbf{x},0} - \zeta\mathbf{U}, \tilde{\boldsymbol{z}}_{\mathbf{h},0}]) \right) \mathbf{U}$ $\quad$ (Eq. 15)
$\quad \tilde{\epsilon} \leftarrow \epsilon_\theta(\boldsymbol{z}_t, t) - \frac{\alpha_t}{\sqrt{1-\alpha_t^2}} \Delta \boldsymbol{z}_0$ $\qquad$ (Eq. 11)
$\quad \mu_{t-1}, \Sigma_{t-1} \leftarrow \frac{1}{\sqrt{1-\beta_t}} (\boldsymbol{z}_t - \frac{\beta_t}{\sqrt{1-\alpha_t^2}} \tilde{\epsilon}), \sigma_t^2 \mathbf{I}$ $\qquad$ (Eq. 3)
$\quad \boldsymbol{z}_{t-1} \leftarrow \mathcal{N}(\mu_{t-1} + s \cdot \sigma_{t-1}^2 g_{t-1}, \Sigma_{t-1})$ $\qquad$ (Eq. 6)
**end for**
$[\mathbf{x}, \mathbf{h}] \leftarrow \mathcal{D}(\boldsymbol{z}_0)$
**return** $[\mathbf{x}, \mathbf{h}]$

---

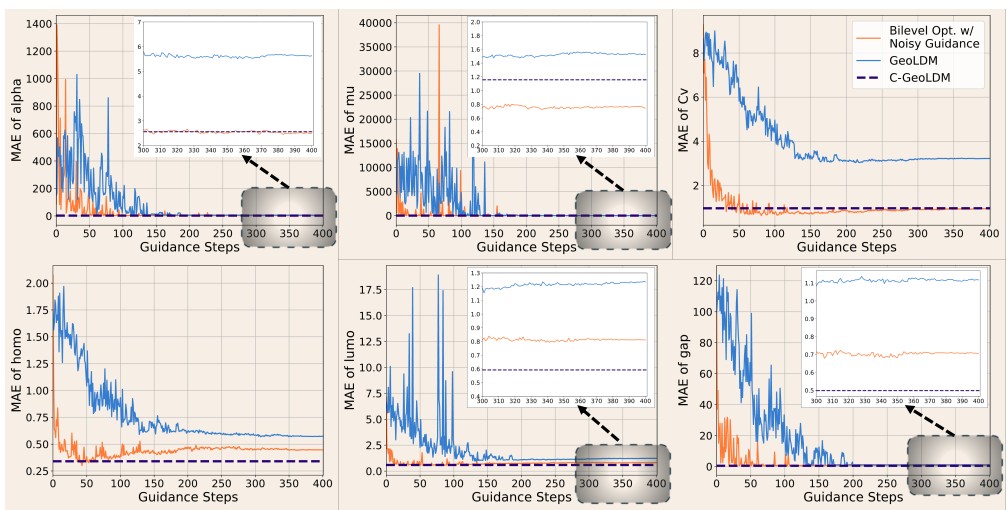

Figure 13: MAE trajectories of 200 generated molecules using GeoLDM and **bilevel optimization with noisy guidance on GeoLDM** with optimal configuration. We use the scales with no collapse as optimal scales.

achieve lower force RMS percentage changes from the baselines. Surprisingly, the MAE of bilevel optimization with clean guidance is lower than that of GeoLDM with clean guidance, which seems to conflict with our findings in bilevel optimization with noisy guidance, we suspect that this is because, as the molecules are optimized to be more stable and thus more similar to the denoised ones, clean guidance can obtain more power in reducing the MAE, as explained in Section L. However, there is a collapse in the bilevel optimization of $\mu$ with scale 10 while it doesn't happen in GeoLDM with clean guidance, we suspect that this is due to the exceptionally high force RMS in GeoLDM with clean guidance, so CHEMGUIDE provides much higher gradient and the guidance from CHEMGUIDE (pushing toward lower force RMS) and clean guidance (pushing towards higher force RMS) conflict, resulting in the collapse of molecular structures. Nevertheless, although the force RMS of bilevel optimization with clean guidance is still higher than the baselines, it is already high in GeoLDM with clean guidance and we can reduce it by adding CHEMGUIDE, which, together with lower MAE, indicates the feasibility of combining CHEMGUIDE with clean guidance and our success in obtaining both better property and force metrics.

Table 17: Force metrics of 200 generated molecules from QM9 dataset using C-EDM, C-GeoLDM, and **explicit bilevel optimization**. Each sub-table contains results for one property that the model is conditioned on. From up to bottom: $\alpha$, $\mu$, $C_v$, $\epsilon_{HOMO}$, $\epsilon_{LUMO}$, $\Delta\epsilon$. $*$ and **bold** denote the overall best result and our best result, respectively. Percentage changes between our results and GeoLDM are shown in parenthesis.

| Metric | Guidance scale ($\alpha$) | | | | | Baseline ($\alpha$) | |
|---|---|---|---|---|---|---|---|
| | 0.0001 | 0.001 | 0.01 | 0.1 | 1.0 | C-EDM | C-GeoLDM |
| Force RMS (Eh/Bohr) | 0.0123 | 0.0124 | **0.0122**$^*$ (-9.21%↓) | 0.0127 | 0.0131 | 0.0125 | 0.0135 |
| Validity | **90.00%**$^*$ (7.00%↑) | 90.00% | 88.00% | 87.50% | 89.50% | 83.50% | 83.00% |
| Uniqueness | **100.00%**$^*$ | **100.00%**$^*$ | **100.00%**$^*$ | **100.00%**$^*$ | 99.47% | 100.00%$^*$ | 100.00%$^*$ |
| Atom Stability | **98.87%**$^*$ (0.63%↑) | 98.81% | 98.76% | 98.57% | 98.87% | 98.44% | 98.23% |
| Molecule Stability | **89.50%**$^*$ (8.50%↑) | 89.00% | 87.50% | 87.50% | 87.00% | 82.50% | 81.00% |
| Energy above ground state (Eh) | 0.0083 | 0.0085 | **0.0079**$^*$ (-13.93%↓) | 0.0084 | 0.0104 | 0.0094 | 0.0092 |

| Metric | Guidance scale ($\mu$) | | | | | Baseline ($\mu$) | |
|---|---|---|---|---|---|---|---|
| | 0.0001 | 0.001 | 0.01 | 0.1 | 1.0 | C-EDM | C-GeoLDM |
| Force RMS (Eh/Bohr) | 0.0120 | 0.0120 | 0.0122 | **0.0119**$^*$ (-3.33%↓) | 0.0136 | 0.0123 | 0.0123 |
| Validity | 82.00% | 82.00% | 83.50% | **85.00%** (-1.50%↓) | 83.50% | 83.00% | 86.50%$^*$ |
| Uniqueness | **100.00%**$^*$ | **100.00%%**$^*$ | **100.00%%**$^*$ | **100.00%%**$^*$ | **100.00%%**$^*$ | 100.00%$^*$ | 100.00%$^*$ |
| Atom Stability | 98.18% | 98.18% | 98.23% | **98.32%**$^*$ (0.03%↑) | 97.96% | 97.48% | 98.29% |
| Molecule Stability | 80.00% | 80.00% | 81.50% | **82.00%** (-2.00%↓) | 77.50% | 79.00% | 84.00%$^*$ |
| Energy above ground state (Eh) | **0.0106** (16.72%↑) | 0.0106 | 0.0112 | 0.0111 | 0.0116 | 0.0113 | 0.0091$^*$ |

| Metric | Guidance scale ($C_v$) | | | | | Baseline ($C_v$) | |
|---|---|---|---|---|---|---|---|
| | 0.0001 | 0.001 | 0.01 | 0.1 | 1.0 | C-EDM | C-GeoLDM |
| Force RMS (Eh/Bohr) | 0.0111 | 0.0111 | **0.0111**$^*$ (-8.75%↓) | 0.0112 | 0.0123 | 0.0117 | 0.0121 |
| Validity | 86.00% | 86.00% | 86.50% | **88.50%**$^*$ (0.50%↑) | 87.00% | 75.80% | 88.00% |
| Uniqueness | **100.00%**$^*$ | **100.00%**$^*$ | **100.00%**$^*$ | **100.00%**$^*$ | **100.00%**$^*$ | 100.00%$^*$ | 100.00%$^*$ |
| Atom Stability | 98.48% | 98.48% | 98.57% | **98.68%**$^*$ (0.11%↑) | 98.57% | 97.43% | 98.57% |
| Molecule Stability | **85.00%** (-1.00%↓) | 85.00% | 85.50% | 84.50% | 84.00% | 74.50% | 86.00%$^*$ |
| Energy above ground state (Eh) | 0.0067 | 0.0066 | **0.0063**$^*$ (-21.39%↓) | 0.0066 | 0.0092 | 0.0097 | 0.0081 |

| Metric | Guidance scale ($\epsilon_{HOMO}$) | | | | | Baseline ($\epsilon_{HOMO}$) | |
|---|---|---|---|---|---|---|---|
| | 0.0001 | 0.001 | 0.01 | 0.1 | 1.0 | C-EDM | C-GeoLDM |
| Force RMS (Eh/Bohr) | **0.0106**$^*$ (-6.69%↓) | 0.0106 | 0.0108 | 0.0128 | 0.0118 | 0.0113 |
| Validity | 86.00% | 86.00% | 87.00% | **90.00%**$^*$ (3.00%↑) | 85.50% | 82.00% | 87.00% |
| Uniqueness | **100.00%**$^*$ | **100.00%**$^*$ | **100.00%**$^*$ | **100.00%**$^*$ | 99.46% | 100.00%$^*$ | 100.00%$^*$ |
| Atom Stability | 98.81% | 98.76% | **98.84%**$^*$ (0.33%↑) | 98.79% | 98.40% | 98.17% | 98.51% |
| Molecule Stability | 87.50% | 87.00% | **88.00%**$^*$ (4.00%↑) | 88.00% | 85.00% | 80.50% | 84.00% |
| Energy above ground state (Eh) | **0.0057**$^*$ (-27.87%↓) | 0.0057 | 0.0061 | -0.0060 | 0.0125 | 0.0090 | 0.0079 |

| Metric | Guidance scale ($\epsilon_{LUMO}$) | | | | | Baseline ($\epsilon_{LUMO}$) | |
|---|---|---|---|---|---|---|---|
| | 0.0001 | 0.001 | 0.01 | 0.1 | 1.0 | C-EDM | C-GeoLDM |
| Force RMS (Eh/Bohr) | 0.0119 | 0.0117 | **0.0115**$^*$ (-1.27%↓) | 0.0124 | 0.0141 | 0.0117 | 0.0117 |
| Validity | 81.50% | 81.50% | 82.00% | 79.50% | **86.50%**$^*$ (2.00%↑) | 83.00% | 84.50% |
| Uniqueness | **100.00%**$^*$ | **100.00%**$^*$ | 99.45% | **100.00%**$^*$ | **100.00%**$^*$ | 100.00%$^*$ | 100.00%$^*$ |
| Atom Stability | 98.12% | 98.15% | 98.18% | 97.74% | **98.62%**$^*$ (0.30%↑) | 98.20% | 98.32% |
| Molecule Stability | 79.50% | 80.00% | 80.50% | 78.00% | **84.00%** (-0.50%↓) | 81.50% | 84.50%$^*$ |
| Energy above ground state (Eh) | **0.0079**$^*$ (-0.60%↓) | 0.0079 | 0.0079 | 0.0102 | 0.0096 | 0.0083 | 0.0079 |

| Metric | Guidance scale ($\Delta\epsilon$) | | | | | Baseline ($\Delta\epsilon$) | |
|---|---|---|---|---|---|---|---|
| | 0.0001 | 0.001 | 0.01 | 0.1 | 1.0 | C-EDM | C-GeoLDM |
| Force RMS (Eh/Bohr) | 0.0104 | 0.0106 | 0.0104 | **0.0101**$^*$ (-10.19%↓) | 0.0121 | 0.0115 | 0.0112 |
| Validity | 88.50% | **89.00%** (-1.50%↓) | 88.50% | 87.50% | 83.50% | 86.00% | 90.50%$^*$ |
| Uniqueness | **100.00%**$^*$ | **100.00%**$^*$ | **100.00%**$^*$ | **100.00%**$^*$ | **100.00%**$^*$ | 100.00%$^*$ | 100.00%$^*$ |
| Atom Stability | **98.87%**$^*$ (0.08%↑) | 98.79% | 98.76% | 98.46% | 98.04% | 98.33% | 98.79% |
| Molecule Stability | **88.50%**$^*$ | 88.00% | 88.00% | 86.00% | 81.00% | 81.50% | 88.50% |
| Energy above ground state (Eh) | **0.0061**$^*$ (-16.41%↓) | 0.0065 | 0.0064 | -0.0065 | 0.0094 | 0.0072 | 0.0073 |

Table 18: MAE of all 6 properties of 200 sampled molecules from **explicit bilevel optimization** and **implicit bilevel optimization with noisy guidance**. The "Top" scales of each property can be found in Table. 16. We use C-EDM and C-GeoLDM as baselines. $*$ and **bold** denote the overall best and the best within different scales, respectively. ✗ marks the situation where the guidance collapses. Percentage changes between our results and C-GeoLDM are shown in parenthesis

| Property | | $\alpha$ | $\mu$ | $C_v$ | $\epsilon_{HOMO}$ | $\epsilon_{LUMO}$ | $\Delta\epsilon$ |
|---|---|---|---|---|---|---|---|
| Units | | Bohr$^3$ | D | $\frac{cal}{mol}$K | meV | meV | meV |
| Conditional EDM | | 2.5089 | 1.0571 | 1.0624 | 0.3304 | 0.5046 | 0.6191 |
| Conditional GeoLDM | | 2.5593 | 1.1582 | 0.9646 | 0.3407 | 0.5927 | 0.4982 |
| GeoLDM | | 5.7481 | 1.4821 | 3.1630 | 0.6042 | 1.1602 | 1.1967 |
| **Explicit Bilevel Optimization** | scale = $1.0 \times 10^{-4}$ | 2.5449 | 1.1151 | 0.9468 | 0.3554 | 0.5173 | 0.5327 |
| | scale = $1.0 \times 10^{-3}$ | 2.5304 | 1.1180 | **0.9393** (-2.62%↓) | 0.3554 | 0.5306 | **0.5225** (4.88%↑) |
| | scale = $1.0 \times 10^{-2}$ | 2.4684 | 1.1246 | 0.9727 | 0.3517 | **0.5077** (-14.34%↓) | 0.5354 |
| | scale = $1.0 \times 10^{-1}$ | 2.5394 | **1.0483** (-9.49%↓) | 0.9843 | **0.3404** (-0.09%↓) | 0.5317 | 0.5400 |
| | scale = $1.0 \times 10^{+0}$ | **2.4430** (-4.54%↓) | 1.0493 | 0.9694 | 0.3690 | 0.5288 | 0.5695 |
| **Implicit Bilevel Optimization w/ Noisy Guidance** | Top 1 | 2.1493 | **0.6364**$^*$ (-45.05%↓) | 0.9290 | 0.4358 | 0.8359 | 0.7890 |
| | Top 2 | **1.9703**$^*$ (-23.01%↓) | 0.7012 | **0.7649**$^*$ (-20.70%↓) | **0.4217** (23.17%↑) | 0.8037 | 0.6428 |
| | Top 3 | 2.8100 | 0.7712 | 0.9795 | 0.4375 | **0.7598** (28.19%↑) | 0.7663 |
| | Top 4 | 3.0144 | 0.8594 | 1.0742 | 0.4832 | 0.8404 | **0.6182** (24.09%↑) |
| | All | 2.4860 (-2.86%↓) | 0.7246 (-37.43%↓) | 0.9369 (-2.87%↓) | 0.4445 (30.47%↑) | 0.8099 (36.64%↑) | 0.7041 (41.33%↑) |

## N   MOLECULE VISUALIZATION

We present the visualization of generated molecules with CHEMGUIDE for stability optimization in Fig. 14 and 15, and bilevel optimization for molecular property and stability with noisy neural guidance and CHEMGUIDE in Fig. 16.

Table 19: Force Metrics of all 6 properties of 200 sampled molecules from **implicit bilevel optimization with noisy guidance**. The "Top" scales of each property can be found in Table. 16. We use C-EDM and C-GeoLDM as baselines. ∗ and **bold** denote the overall best and the best within different scales, respectively. Percentage changes between our results and C-GeoLDM are shown in parenthesis

| Metric | Guidance scale ($\alpha$) | | | | | Baseline ($\alpha$) | |
|---|---|---|---|---|---|---|---|
| | Top 1 (25.0) | Top 2 (20.0) | Top 3 (10.0) | Top 4 (5.0) | All | C-EDM | C-GeoLDM |
| Force RMS (Eh/Bohr) | 0.0181 | 0.0132 | 0.0114 | **0.0099**∗ (-26.47%↓) | 0.0132 (-1.94%↓) | 0.0125 | 0.0135 |
| Validity | **92.00%**∗ (9.00%↑) | 80.00% | 88.00% | 88.00% | 87.00% (4.00%↑) | 83.50% | 83.00% |
| Uniqueness | **100.00%**∗ | **100.00%**∗ | **100.00%**∗ | **100.00%**∗ | **100.00%**∗ | 100.00%∗ | 100.00%∗ |
| Atom Stability | 97.87% | 98.54% | 98.77% | **99.44%**∗ (1.21%↑) | 98.66% (0.42%↑) | 98.44% | 98.23% |
| Molecule Stability | 88.00% | 86.00% | 88.00% | **92.00%**∗ (11.00%↑) | 88.50% (7.50%↑) | 82.50% | 81.00% |
| Energy above ground state (Eh) | 0.0087 | 0.0047 | 0.0043 | **0.0028**∗ (-69.26%↓) | 0.0052 (-43.72%↓) | 0.0094 | 0.0092 |

| Metric | Guidance scale ($\mu$) | | | | | Baseline ($\mu$) | |
|---|---|---|---|---|---|---|---|
| | Top 1 (30.0) | Top 2 (25.0) | Top 3 (20.0) | Top 4 (10.0) | All | C-EDM | C-GeoLDM |
| Force RMS (Eh/Bohr) | 0.0122 | 0.0153 | **0.0095**∗ (-23.12%↓) | 0.0098 | 0.0117 (-4.57%↓) | 0.0123 | 0.0123 |
| Validity | 92.00% | 94.00% | 84.00% | **96.00%**∗ (9.50%↑) | 91.50% (5.00%↑) | 83.00% | 86.50% |
| Uniqueness | **100.00%**∗ | **100.00%**∗ | **100.00%**∗ | **100.00%**∗ | **100.00%**∗ | 100.00%∗ | 100.00%∗ |
| Atom Stability | 97.48% | 97.88% | **98.55%**∗ (0.26%↑) | 97.71% | 97.91% (-0.38%↓) | 97.48% | 98.29% |
| Molecule Stability | 84.00% | 84.00% | **86.00%**∗ (2.00%↑) | 84.00% | 84.50% (0.50%↑) | 79.00% | 84.00% |
| Energy above ground state (Eh) | 0.0035 | 0.0108 | 0.0027 | **0.0025**∗ (-72.31%↓) | 0.0050 (-45.45%↓) | 0.0113 | 0.0091 |

| Metric | Guidance scale ($C_v$) | | | | | Baseline ($C_v$) | |
|---|---|---|---|---|---|---|---|
| | Top 1 (30.0) | Top 2 (25.0) | Top 3 (20.0) | Top 4 (10.0) | All | EDM | GeoLDM |
| Force RMS (Eh/Bohr) | 0.0131 | 0.0105 | 0.0106 | **0.0099**∗ (-18.40%↓) | 0.0110 (-8.97%↓) | 0.0117 | 0.0121 |
| Validity | **92.00%**∗ (4.00%↑) | 82.00% | 88.00% | **92.00%**∗ (4.00%↑) | 88.50% (0.50%↑) | 75.80% | 88.00% |
| Uniqueness | **100.00%**∗ | **100.00%**∗ | **100.00%**∗ | **100.00%**∗ | **100.00%**∗ | 100.00%∗ | 100.00%∗ |
| Atom Stability | 96.95% | 96.73% | 98.32% | **98.79%**∗ (0.22%↑) | 97.69% (-0.88%↓) | 97.43% | 98.57% |
| Molecule Stability | 80.00% | 72.00% | 84.00% | **88.00%**∗ (2.00%↑) | 81.00% (-5.00%↓) | 74.50% | 86.00% |
| Energy above ground state (Eh) | 0.0139 | 0.0036 | 0.0037 | **0.0025**∗ (-68.78%↓) | 0.0060 (-24.96%↓) | 0.0097 | 0.0081 |

| Metric | Guidance scale ($\epsilon_{HOMO}$) | | | | | Baseline ($\epsilon_{HOMO}$) | |
|---|---|---|---|---|---|---|---|
| | Top 1 (40.0) | Top 2 (50.0) | Top 3 (30.0) | Top 4 (20.0) | All | EDM | GeoLDM |
| Force RMS (Eh/Bohr) | 0.0102 | 0.0101 | **0.0099**∗ (-12.33%↓) | 0.0103 | 0.0101 (-10.66%↓) | 0.0118 | 0.0113 |
| Validity | **92.00%**∗ (5.00%↑) | 84.00% | **92.00%**∗ (5.00%↑) | 84.00% | 88.00% (1.00%↑) | 82.00% | 87.00% |
| Uniqueness | **100.00%**∗ | **100.00%**∗ | **100.00%**∗ | **100.00%**∗ | **100.00%**∗ | 100.00%∗ | 100.00%∗ |
| Atom Stability | **98.77%**∗ (0.26%↑) | 97.87% | **98.77%**∗ (0.26%↑) | 98.54% | 98.49% (-0.02%↓) | 98.17% | 98.51% |
| Molecule Stability | 88.00% | 86.00% | **90.00%**∗ (6.00%↑) | 86.00% | 87.50% (3.50%↑) | 80.50% | 84.00% |
| Energy above ground state (Eh) | 0.0032 | 0.0028 | 0.0030 | **0.0028**∗ (-64.91%↓) | 0.0029 (-62.59%↓) | 0.0090 | 0.0079 |

| Metric | Guidance scale ($\epsilon_{LUMO}$) | | | | | Baseline ($\epsilon_{LUMO}$) | |
|---|---|---|---|---|---|---|---|
| | Top 1 (50.0) | Top 2 (25.0) | Top 3 (40.0) | Top 4 (30.0) | All | EDM | GeoLDM |
| Force RMS (Eh/Bohr) | 0.0096 | 0.0100 | 0.0100 | **0.0095**∗ (-18.39%↓) | 0.0098 (-16.26%↓) | 0.0117 | 0.0117 |
| Validity | **96.00%**∗ (11.50%↑) | 94.00% | **96.00%**∗ (11.50%↑) | 92.00% | 94.50% (10.00%↑) | 83.00% | 84.50% |
| Uniqueness | **100.00%**∗ | **100.00%**∗ | **100.00%**∗ | **100.00%**∗ | **100.00%**∗ | 100.00%∗ | 100.00%∗ |
| Atom Stability | 99.08% | 99.22% | 97.88% | **99.54%**∗ (1.22%↑) | 98.92% (0.61%↑) | 98.20% | 98.32% |
| Molecule Stability | **94.00%** ∗(9.50%↑) | 90.00% | 84.00% | **94.00%**∗ (9.50%↑) | 90.50% (6.00%↑) | 81.50% | 84.50% |
| Energy above ground state (Eh) | **0.0021**∗ (-73.79%↓) | 0.0024 | 0.0026 | 0.0025 | 0.0024 (-69.62%↓) | 0.0083 | 0.0079 |

| Metric | Guidance scale ($\Delta\epsilon$) | | | | | Baseline ($\Delta\epsilon$) | |
|---|---|---|---|---|---|---|---|
| | Top 1 (40.0) | Top 2 (30.0) | Top 3 (20.0) | Top 4 (25.0) | All | EDM | GeoLDM |
| Force RMS (Eh/Bohr) | 0.0104 | 0.0104 | 0.0109 | **0.0099**∗ (-11.47%↓) | 0.0104 (-7.18%↓) | 0.0115 | 0.0112 |
| Validity | 88.00% | **96.00%**∗ (5.50%↑) | 92.00% | 92.00% | 92.00% (1.50%↑) | 86.00% | 90.50% |
| Uniqueness | **100.00%**∗ | **100.00%**∗ | **100.00%**∗ | **100.00%**∗ | **100.00%**∗ | 100.00%∗ | 100.00%∗ |
| Atom Stability | 97.47% | 98.04% | 98.73% | **99.02%**∗ (0.23%↑) | 98.32%(-0.47%↓) | 98.33% | 98.79% |
| Molecule Stability | 80.00% | **88.00%**∗ (-0.50%↓) | 88.00 | **88.00%**∗ | 86.00%(-2.50%↓) | 81.50% | 88.50%∗ |
| Energy above ground state (Eh) | 0.0034 | 0.0036 | **0.0026**∗ (-64.64%↓) | 0.0028 | 0.0031 (-57.32%↓) | 0.0072 | 0.0073 |

Table 20: MAE and force RMS of $\alpha$ and $\mu$ of 500 sampled molecules from **GeoLDM with clean guidance** and 200 sampled molecules from **implicit bilevel optimization with clean guidance**. The "Top" scales of each property can be found in Table. 16. We use C-EDM and C-GeoLDM as baselines. $*$ and **bold** denote the overall best and the best within different scales, respectively. ✗ marks the situation where the guidance collapses.

| Model ($\alpha$) | | Guidance scale | | | | | Baseline ($\alpha$) | |
|---|---|---|---|---|---|---|---|---|
| | | Top 1 (1.0) | Top 2 (0.1) | Top 3 (0.01) | Top 4 (0.0001) | All | C-EDM | C-GeoLDM |
| Clean Guidance | Force RMS (Eh/Bohr) | 0.1086 | 0.0179 | 0.0112 | 0.0112 | 0.0326 | 0.0118 | 0.0117 |
| | MAE | 2.4980 | 5.4798 | 5.6894 | 5.7236 | 4.9992 | 2.6308 | 2.5551 |
| Bilevel Optimization w/ Clean Guidance | Force RMS (Eh/Bohr) | 0.0972 | 0.0205 | 0.0101 | 0.0111 | 0.0274 | 0.0125 | 0.0135 |
| | MAE | 1.6182 | 4.2294 | 4.9078 | 4.9764 | 4.2058 | 2.5089 | 2.5593 |
| Model ($\mu$) | | Guidance scale | | | | | Baseline ($\mu$) | |
| | | Top 1 (10.0) | Top 2 (5.0) | Top 3 (2.0) | Top 4 (1.0) | All | C-EDM | C-GeoLDM |
| Clean Guidance | Force RMS (Eh/Bohr) | 1.1042 | 0.2684 | 0.2760 | 0.0836 | 0.3962 | 0.0122 | 0.0120 |
| | MAE | 0.1631 | 0.2031 | 0.4475 | 0.7144 | 0.4025 | 1.1257 | 1.1084 |
| Bilevel Optimization w/ Clean Guidance | Force RMS (Eh/Bohr) | ✗ | 0.1454 | 0.1524 | 0.1109 | 0.1348 (150) | 0.0123 | 0.0123 |
| | MAE | ✗ | 0.1335 | 0.2452 | 0.5420 | 0.3150 (150) | 1.0571 | 1.1582 |

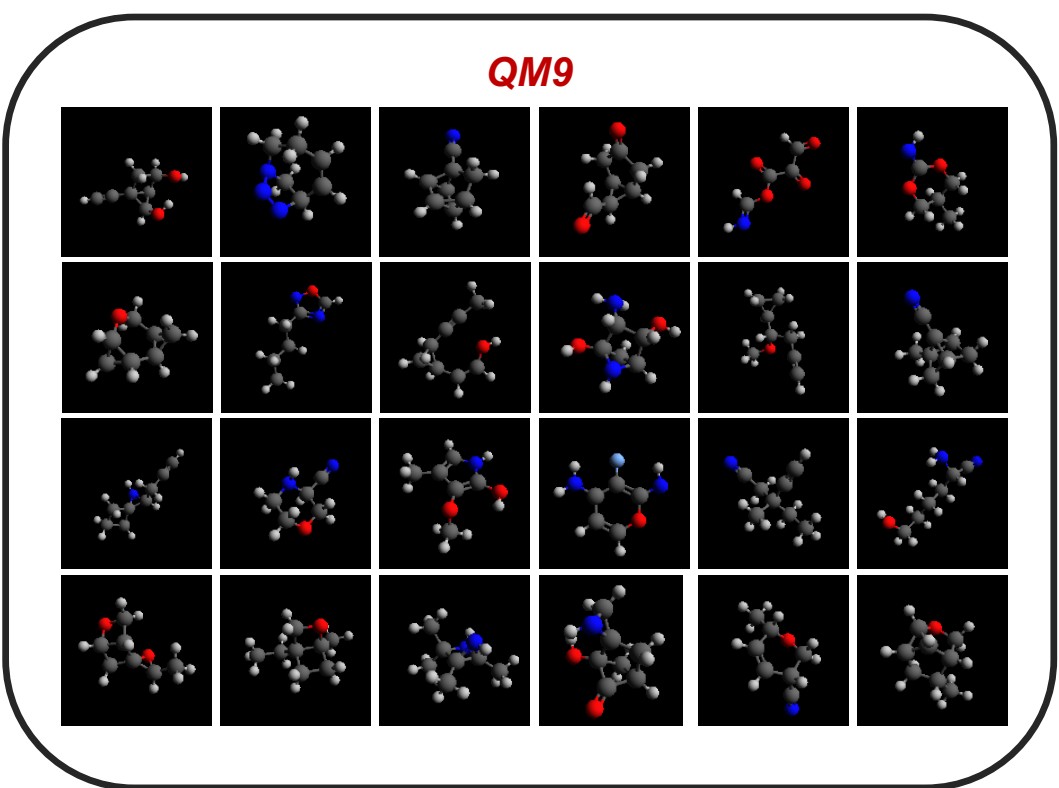

Figure 14: Molecules generated from *QM9* using unconditional GeoLDM with CHEMGUIDE.

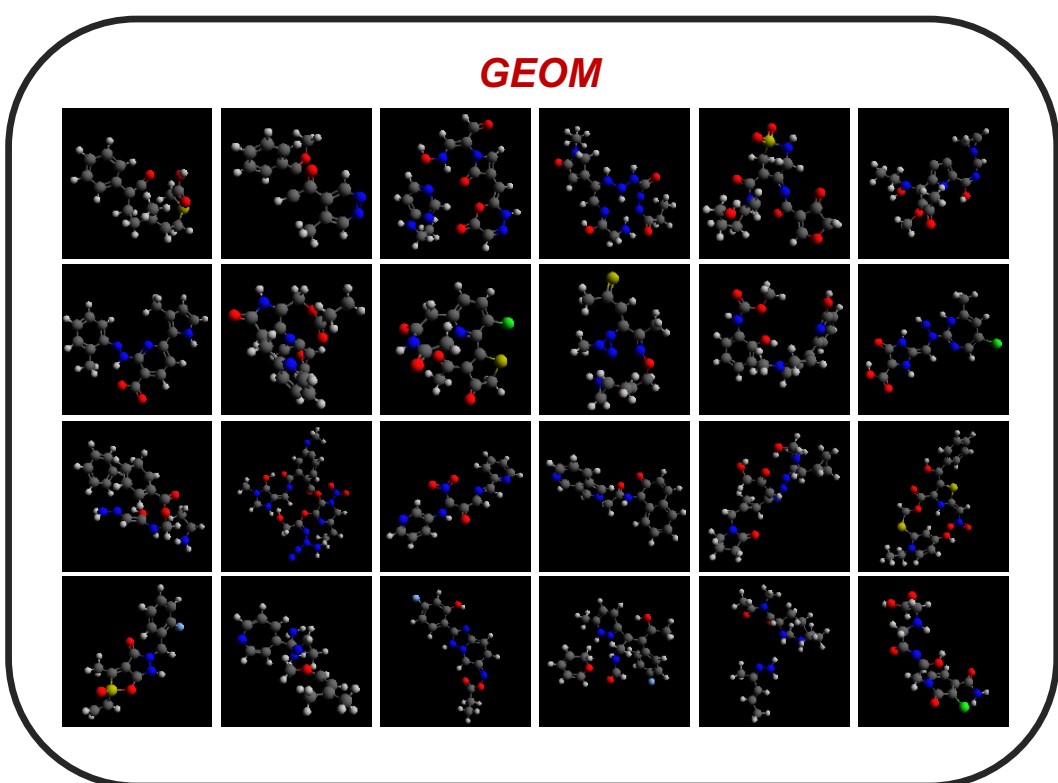

Figure 15: Molecules generated from *GEOM* using unconditional GeoLDM with CHEMGUIDE.

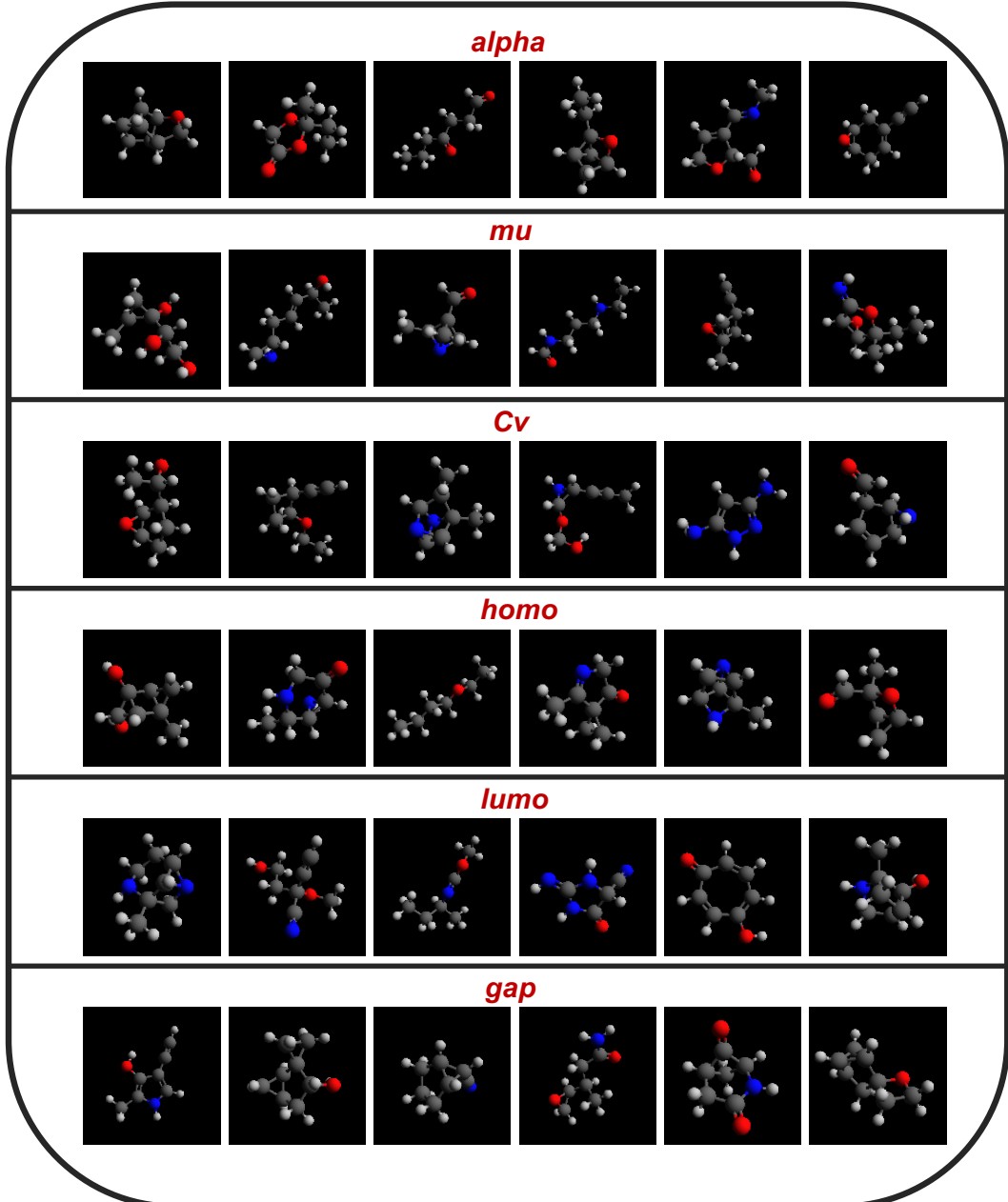

Figure 16: Molecules generated from *QM9* using bilevel optimization with noisy neural guidance and CHEMGUIDE for unconditional GeoLDM.

