# OpenReview forum: "Chemistry-Inspired Diffusion with Non-Differentiable Guidance"
_ICLR.cc/2025/Conference — ICLR 2025 Poster_

### Official Review · Reviewer_wXP8 · 2024-10-30

**Soundness:** 4
**Presentation:** 4
**Contribution:** 3
**Rating:** 8
**Confidence:** 4

**Summary:**

The authors propose ChemGuide, a method for estimating diffusion guidance (i.e. gradients) from a non-differentiable property predictor. The goal is to eliminate the need for labeled training data typically required for property prediction networks. They demonstrate their approach in the context of 3D molecular generation. ChemGuide enables an unconditional diffusion model to generate more stable 3D molecular structures by incorporating guidance from quantum chemistry calculations (GFN2-xTB) that serve as a non-differentiable oracle.

**Strengths:**

- **Strong and relevant contribution**: The idea proposed in the paper has clearly relevant applications in a number of domains (i.e. any field with expensive oracles) and naturally extends existing efforts in the field of diffusion models.
- **Novelty**: The method is conceptually simple yet novel, opening the door for various applications which could benefit from the guidance of a non-differentiable oracle.
- **Thorough empirical evaluation**: The paper presents thorough empirical analysis, effectively demonstrating both the strengths and limitations of the proposed method. The experiments span multiple datasets (QM9 and GEOM), various molecular properties, and different guidance approaches (explicit, implicit, and combined).
-  **Extensive analysis**: the empirical observations are grounded in real-world chemistry insights, with careful analysis of failure cases and performance trade-offs.
- **Clarity of presentation**: the paper is well-written and includes many relevant and well-designed figures.

**Weaknesses:**

- **Justification of the zeroth-order optimization and comparison to other non-gradient optimization methods**: The paper lacks clear justification for choosing SPSA as the zeroth-order optimization method. Section 3.2 would benefit from a discussion of alternative approaches (e.g., finite differences, evolution strategies) and justification for their specific choice in terms of computational efficiency and accuracy trade-offs, and suitability for this particular application of guiding molecular diffusion models. A short pragaraph would suffice here.

- **Assessing the quality of the gradients obtained with ChemGuide vs a differentiable regressor** While the paper shows final performance metrics, it lacks direct analysis comparing the gradients estimated via zeroth-order optimization to those from a differentiable regressor. Such comparison could provide insights into how reliable are CHEMGUIDE's estimated gradients compared to differentiated gradients. For example, the authors could plot the cosine similarity between CHEMGUIDE's estimated gradients and those from a differentiable regressor across different timesteps of the diffusion process

- **Explain which guidance method is suitable for which property**:  The authors observe that noisy and clean guidance methods appear complementary, with properties poorly optimized by one often well-optimized by another (e.g., α vs ∆ϵ). However, the paper would be more practically useful if it provided explanations for these differences, helping practitioners choose the appropriate method for their specific use case.

**Questions:**

**clarifications**
- Can you clarify which aspects of your method implement SPSA? Equation 15 combines two ideas: 1) introducing random perturbations for atom coordinates from a standard normal distribution, and 2) using these perturbations in a finite-difference approximation of the gradient. It would be helpful to explicitly state which of these constitutes SPSA and how it differs from standard finite differences.

- What are the theoretical requirements for SPSA convergence? The paper mentions continuity of z as a requirement, but are there other conditions (e.g., smoothness, bounded variance) needed for the gradient estimates to be reliable?

**suggestions**
- Given the extensive appendix (sections A-K), adding a table of contents at its beginning would improve navigation.

- Consider adding a brief discussion of computational overhead introduced by SPSA compared to standard finite differences.

**nitpicks**
- Stability' is unnecessarily capitalized on page 6, section 4.2

---

> ### Author Response · Authors · 2024-11-23
>
> Thank you for the detailed suggestions on the presentation of our paper! We address the weaknesses and questions as follows.
>
> ### **1. Justification of SPSA & Comparison to other methods**
>
> - **SPSA justification**: we have added more descriptions and our motivation for SPSA in **Appendix F** Motivation for Gradient Estimation with SPSA. We previously considered using finite difference to calculate the gradient as $\frac{F([z_{x, t}+\zeta 1_{N \times 3}, z_{h, t}])-F([z_{x, t}-\zeta 1_{N \times 3}, z_{h, t}])}{2\zeta}$, however, it violates the zero-mean requirements for 3D diffusion models to be equivariant (i.e.,  $z_{x, t}\pm\zeta 1_{N \times 3}$ shifts the mean by $\pm\zeta$), which motivates the choice of our SPSA based estimation (i.e., the perturbation is sampled from the normal distribution).
>
> - **Alternative methods**: we have conducted new experiments and added the results for evolutionary algorithms in the (global) official comments above. (please see **Appendix I** in our current revised paper). It turns out that ChemGuide outperforms the evolutionary algorithm in terms of force RMS and energy about ground state.
>
> ### **2. Quality of the guidance gradients**
>
> We employed a neural network potential AIMNet2 [1] to calculate the force, and compared its backpropagated gradient with ChemGuide measured in cosine similarity in Appendix H3 ChemGuide vs. Neural Guidance on Force. We can observe that the cosine similarity over time oscillates around 0, suggesting that the guidance from ChemGuide and neural networks are different, hence the discrepancy in performance.
>
> We report the results of neural network guidance on force in the (global) official comments above. (**Appendix H3** in our current revised paper). We observe that using neural networks for force guidance  **underperforms ChemGuide**, and significantly increases memory demands, which makes guidance on large molecules extremely difficult (e.g., a 48 GiB GPU can only generate 5 molecules at a time on GEOM, making sampling a large amount of molecules almost linear in time).
>
> ### **3. Compatibility between guidance methods and property**
>
> From the chemistry definitions, there is no direct evidence to suggest that one property would be harder or easier than another for the neural regressor to learn. However, we can observe from Figure 7 on the generalization analysis of neural regressor, that on property $\alpha$ the regressor is more robust in terms of perturbations compared to other properties, meanwhile, in Table 4, noisy guidance performs better than clean guidance on $\alpha$, and vice the versa on other properties.
> We hypothesize that this is because noisy guidance requires backpropagation through the diffusion model, while clean guidance translates the optimized representation $x_0$ in the clean space back to $x_t$ with pre-defined schedules. Thus, with a robust regressor (e.g., on $\alpha$), noisy guidance might be preferred over clean guidance as it directly backpropagates to $x_t$; given a less precise regressor on out-of-distribution molecules, clean guidance would be favored as the errors are more likely to be broadcast through both the VAE decoder and the latent diffusion model for noisy guidance.
>
> However, our analyses are primitive and do not include many factors (e.g., different architectures of the diffusion models), and further investigation in this direction would be very valuable.

---

> > ### Author Response · Authors · 2024-11-23
> >
> > ### **4. Clarification on SPSA and theoretical results**
> >
> > For a differentiable $f: f: E\in R^d \rightarrow R$, finite-difference estimates the gradient at $x$ as:
> > $$\hat{g}(x)=\frac{f(x+\epsilon)-f(x-\epsilon)}{2\epsilon}$$
> > and SPSA [2] estimates the gradient with some scale $\mu>0$ and a zero-mean perturbation vector $\Delta$ as:
> > $$\hat{g}(x)=\frac{f(x+\mu\Delta)-f(x-\mu\Delta)}{2\mu\Delta}$$
> > where we abuse the notation by using dividing $\Delta$ to denote the element-wise operation (e.g., $\Delta_i$ for each dimension $i=1, 2, …, d$)
> >
> > The key difference between finite difference and SPSA is the requirement of $\Delta$ being zero-mean, which made it possible for us to obey the zero center gravity requirement of 3D diffusion models to be equivariant to transformations. However, SPSA indeed can be seen as 1. sample perturbation from the normal distribution; 2. apply finite-difference to approximate the gradients.
> >
> > Further, SPSA [2] requires $\Delta$ to have bounded inverse moments, which rules out $\Delta$ being Gaussian. In order to sample $\Delta$ from the standard normal distribution, which is one of the frequently used distributions with zero mean, we follow [3] and estimate the gradient (which can be seen as a variant of the original SPSA) as follows:
> > $$\hat{g}_{\Delta}(x)=\frac{f(x+\mu \Delta)-f(x-\mu Delta)}{2\mu}\cdot \Delta$$
> >
> > We mention $z$ to be “continuous” in the intention as opposed to “discrete” variables (i.e. molecules in the molecular space), such that we can add the perturbation vector directly to $z$. In the context of gradient estimation, $f$ to be differentiable would suffice, however, if one were to use the estimated gradient $\hat{g}_{\Delta}(x)$ for optimization problems (e.g., finding the minimizer of $f$), [3] assumes $f$ is convex and studied the properties and convergence of such algorithms under various settings (e.g., $f$ is smooth/non-smooth)
> >
> > In terms of the practices of such an estimation, SPSA has empirically demonstrated its effectiveness in deep learning applications, such as in Large Language Models [4], and other domains [5].
> >
> > ### **5. Response to suggestions & nitpicks**
> >
> > We have added a table of contents at the beginning of the appendix, and changed the capitalized “Stability” accordingly in our current revised submission. Thank you for your detailed inputs to make our paper more accessible!
> >
> > We include the discussion on the computation overhead of SPSA in **Appendix F** Motivation for Gradient Estimation with SPSA. There are two extra operations from SPSA: 1. sampling $U$ from Gaussian; 2. matrix multiplication with $U$, where the extra computation overhead of SPSA compared to finite-difference methods is ignorable during the diffusion process, since both operations are fast and well-optimized on GPUs in modern machine learning practices.
> >
> > ### **References**
> >
> > [1] Dylan Anstine, Roman Zubatyuk, and Olexandr Isayev. Aimnet2: a neural network potential to meet your neutral, charged, organic, and elemental-organic needs. 2024.
> >
> > [2] James C. Spall. Multivariate stochastic approximation using a simultaneous perturbation gradient approximation. IEEE Transactions on Automatic Control, 37:332–341, 1992. URL https://api.semanticscholar.org/CorpusID:122365276.
> >
> > [3] Yurii Nesterov and Vladimir Spokoiny. Random gradient-free minimization of convex functions. Foundations of Computational Mathematics, 17(2):527–566, 2017
> >
> > [4] Sadhika Malladi, Tianyu Gao, Eshaan Nichani, Alexandru Damian, Jason D. Lee, Danqi Chen, and Sanjeev Arora. Fine-tuning language models with just forward passes. NeurIPS 2023
> >
> > [5] Sijia Liu, Pin-Yu Chen, Bhavya Kailkhura, Gaoyuan Zhang, Alfred O Hero III, and Pramod K Varshney. A primer on zeroth-order optimization in signal processing and machine learning: Principals, recent advances, and applications. IEEE Signal Processing Magazine, 37(5):43–54, 2020.

---

> > > ### Author Response · Authors · 2024-11-23
> > >
> > > We hope the above information helps address your concerns and questions. We are happy to answer any future questions you may have. Thank you!
> > >
> > > We also included additional results and summaries in <https://openreview.net/forum?id=4dAgG8ma3B&noteId=UrUPt5fBhg>.

---

> ### Comment · Reviewer_wXP8 · 2024-11-25
>
> Thank you for providing new experiments and revising your manuscript to incorporate the changes requested.
>
> Overall, while I agree with the other reviewers about the existence of multiple non-differentiable guidance methods, I find the simplicity of the current method (using zeroth-order optimization to approximate the gradients of the oracle) appealing, especially in light of the discussion the authors provided when comparing to evolutionary algorithms and a property predictor neural network.
>
> I will thus maintain my score.

---

> > ### Author Response · Authors · 2024-12-04
> >
> > Thank you very much for your engagement in the discussion and valuable suggestions to make our paper better.

---

### Official Review · Reviewer_tzoc · 2024-10-31

**Soundness:** 2
**Presentation:** 2
**Contribution:** 1
**Rating:** 5
**Confidence:** 5

**Summary:**

This paper studies how to achieve derivative-free molecular optimization with diffusion models. The main motivation for this work is that many real-world molecular properties are sophisticated and can only be evaluated through expensive experiments or non-differential simulations. In this paper, a zero-order optimization method is constructed by perturbing the input molecular conformation and the effect on the molecular properties. The effectiveness of the proposed methods is validated on a set of quantum mechanical properties for small molecules.

**Strengths:**

* This paper studies an important and timely problem, how to move beyond generative models (randomly sample from learned distributions) to efficient search and optimization over the space with guidance is a pressing question in molecular generation.
* Derivative-free guidance is very well-motivated and I agree that it is of great importance to problems in real-world molecular discovery process.

**Weaknesses:**

I have several concerns about this paper, mostly coming from the claim, related work, and experiments.

* First of all, the idea of derivative-free guidance is not new, in molecular optimization, evolutionary algorithms have been used [1], twisted sequential Monte Carlo for protein design [2], twisted SMC for large language model inference [3], and stochastic optimal control method for music generation [4]. I believe the claim for this work to be the first of its kind is inappropriate, neither for derivative-free guidance nor in molecular design.

* Given this is not new, the related work section in the Appendix only discusses the general molecule generation literature and part of the guided diffusion model literature, but misses the critical relevant literature both in molecular generation and other domains.

* The experimental results are weak, even if I agree on the general motivation of derivative-free guidance, (1) there are works such as simple evolutionary algorithms [1] and twisted SMC [2] available for comparison; even if you do not want to compare against them, you need to compare with gradient-based method --- if you think about the experiment budget, you can always construct a classifier by the samples you have evaluated, e.g. a trained neural network. Despite this may not generalize OOD or perform badly, but you may still include them as baselines. For more potential baselines to compare against, you can check this benchmark [5].

[1] Schneuing, A., Du, Y., Harris, C., Jamasb, A., Igashov, I., Du, W., Blundell, T., Lió, P., Gomes, C., Welling, M. and Bronstein, M., 2022. Structure-based drug design with equivariant diffusion models. arXiv preprint arXiv:2210.13695.

[2] Wu, L., Trippe, B., Naesseth, C., Blei, D. and Cunningham, J.P., 2024. Practical and asymptotically exact conditional sampling in diffusion models. Advances in Neural Information Processing Systems, 36.

[3] Zhao, S., Brekelmans, R., Makhzani, A. and Grosse, R.B., Probabilistic Inference in Language Models via Twisted Sequential Monte Carlo. In Forty-first International Conference on Machine Learning.

[4] Huang, Y., Ghatare, A., Liu, Y., Hu, Z., Zhang, Q., Sastry, C.S., Gururani, S., Oore, S. and Yue, Y., Symbolic Music Generation with Non-Differentiable Rule Guided Diffusion. In Forty-first International Conference on Machine Learning, 2024.

[5] https://github.com/brandontrabucco/design-bench

**Questions:**

Most of my questions are about experiments, I feel the current experimental comparisons are too weak (only comparing with unconditional generation), see weaknesses.

---

> ### Author Response · Authors · 2024-11-23
>
> Thank you for taking the time and we appreciate your input to help make our paper a better work! We address the weaknesses and questions as follows.
>
> ### **1. Novelty and motivation**
> We would like to re-emphasize that our focus is on, **for molecule optimization**, how to achieve controllable generation at inference time from quantum chemistry to bypass the requirement to train a neural property predictor or a conditional diffusion model, which **require a large amount of labels** that are hard to acquire. We choose stability (defined only on 3D) to explore this motivation, and hence experiment on 3D diffusion models.
>
> We are not proposing a new guidance method (e.g., as in [2]) nor are claiming to be the first to introduce derivative-free guidance to diffusion models.
> In fact, our method builds on previous proposed guided diffusion methods [3, 4], and the use of zeroth-order optimization techniques [5, 6] for gradient estimation is to overcome the difficulty that we can not backpropagate the chemistry software.
>
> The evolution algorithm [1] relies on stochastic mutations and pruning, which requires a large search space to be effective and do not directly change the underlying **conditional** distribution from which the diffusion model samples.  Gradients provide informative guidance by indicating the correct direction for optimizing stability. Although the evolutionary algorithm selects the best variant at each evolutionary step, it doesn’t rely on the gradient, so optimization direction is essentially random, making the process less controllable.
>
> For tSMC[2], it proposes a novel guidance method and **requires $p_{y|x^0}(y|x^0)$, which is a classifier that is trained on labels** (refer to Section 5.2 and 6.2 in [2] on the choice of such likelihood $p$) and thus not directly comparable to our works, further, they experiment on protein binding, which is intrinsically different from our task in molecule optimization.
>
> We have excluded the sentence “To the best of our knowledge, this is the first work of its kind.” in the introduction to avoid further confusion in our current revised paper.
>
> ### **2. Related works**
> For the broader discussion of guided diffusion beyond the scope of our work on molecule optimization, we have added the papers you suggested to enlighten further discussions.
>
> ### **3. Additional experiments**
> As stated above, our motivation is not derivative-free guidance but to bypass the need to train a neural predictor or a conditional diffusion model to achieve controllable generation.
> - **Evolutionary algorithm**: we added the results using the evolutionary algorithm [1,7] to the global comment above, please also see **Appendix I** for detailed results and analysis. We can observe ChemGuide outperforms the evolutionary algorithm in terms of force RMS and energy above the ground state.
> - **Neural network guidance**: we replaced the non-differentiable oracle with a neural network potential AIMNet2, the results are reported in the global comment above, please also see **Appendix H3** for detailed results and analysis. It turns out that using AIMNet2 for guidance negatively impacted performance.
>
> ### **References**
> [1] Schneuing, A., Du, Y., Harris, C., Jamasb, A., Igashov, I., Du, W., Blundell, T., Lió, P., Gomes, C., Welling, M. and Bronstein, M., 2022. Structure-based drug design with equivariant diffusion models. arXiv preprint arXiv:2210.13695.
>
> [2] Wu, L., Trippe, B., Naesseth, C., Blei, D. and Cunningham, J.P., 2024. Practical and asymptotically exact conditional sampling in diffusion models. Advances in Neural Information Processing Systems, 36.
>
> [3] Prafulla Dhariwal and Alex Nichol. Diffusion models beat gans on image synthesis, 2021.
>
> [4] Clement Vignac, Igor Krawczuk, Antoine Siraudin, Bohan Wang, Volkan Cevher, and Pascal Frossard. Digress: Discrete denoising diffusion for graph generation. arXiv preprint arXiv:2209.14734, 2022.
>
> [5] James C. Spall. Multivariate stochastic approximation using a simultaneous perturbation gradient approximation. IEEE Transactions on Automatic Control, 37:332–341, 1992. URL https://api.semanticscholar.org/CorpusID:122365276.
>
> [6] Yurii Nesterov and Vladimir Spokoiny. Random gradient-free minimization of convex functions. Foundations of Computational Mathematics, 17(2):527–566, 2017
>
> [7] Huang, Y., Ghatare, A., Liu, Y., Hu, Z., Zhang, Q., Sastry, C.S., Gururani, S., Oore, S. and Yue, Y., Symbolic Music Generation with Non-Differentiable Rule Guided Diffusion. In Forty-first International Conference on Machine Learning, 2024.

---

> > ### Author Response · Authors · 2024-11-23
> >
> > We hope the above information helps address your concerns and questions. We are happy to answer any future questions you may have. Thank you!
> >
> > We also included additional results and summaries in <https://openreview.net/forum?id=4dAgG8ma3B&noteId=UrUPt5fBhg>.

---

> > ### Comment · Reviewer_tzoc · 2024-11-25
> > **Thank you for the discussion**
> >
> > Thank the authors for writing the rebuttal with additional experimental results.
> >
> > In general, I agree with the authors on the nuances of the settings of different methods, but inherently these methods are connected, for twisted SMC, they used a proposal with an additional gradient from the target but it is not necessary.
> >
> > I hate to say this but for such a popular problem with many works (either on proteins/molecules/materials or images/music/language/etc), I don't see major technical advancements this paper brings. In that sense, I am still okay with a more application-oriented paper if the authors can compare and discuss other methods well with a decent performance improvement.
> >
> > I appreciate again the effort the authors made during the rebuttal, but I will maintain my score.

---

> > > ### Author Response · Authors · 2024-12-04
> > >
> > > Thank you for your initial reviews on suggesting adding more baselines (evolutionary algorithms, neural network for force prediction, twisted SMC). In our revised paper, we have added 1. evolutionary algorithm [result](https://openreview.net/forum?id=4dAgG8ma3B&noteId=8BLGcJPJhX); 2. neural network for force prediction [result](https://openreview.net/forum?id=4dAgG8ma3B&noteId=hk1C9qcZok).
> > >
> > > In our previous rebuttal, we have pointed out tSMC [1] requires $p_{y|x^0}(y|x^0)$ and its derivative and thus not directly comparable to our works, and their task in protein binding is intrinsically different from ours in molecule optimization.
> > >
> > > Given your most recent comments on **“for twisted SMC, …, additional gradient … is not necessary”**, we would like to refer to Algorithm 1 in [1], where in line 6 they explicitly requires $\nabla_{x_k^{t+1}} p_k^{t+1}$, and $p_k^{t}=p_{\theta}(y|x_k^t)=p_{y|x^0}(y|\hat{x}_{\theta}(x^t))$ is defined in eq. (8). Without this gradient, it is not the **correct** twisted proposal function defined in eq. (9) and (10), so **it's necessary** to compute the gradient in tSMC.
> > >
> > > Furthermore, to calculate $p_{y|x^0}(y|x^0)$ is impossible for stability, because **y** here will be a #atoms$\times 3$ tensor in $R^d$, and just for the same molecule, there are infinitely many **y**s given the different 3D structure of the same set of atoms. Hence, even we are given enough resources to explore the entire molecule space, it is impractical to estimate this $p_{y|x^0}(y|x^0)$ for stability.
> > >
> > > To summarize, tSMC is not comparable in our setting because 1. The **correct** twisted proposal requires gradients, which is a bottleneck of introducing quantum chemistry to diffusion models before our method; 2. even if we are to use the **wrong** twisted proposal without gradients, the twisted weighting function requires $p_{y|x^0}(y|x^0)$, which is impossible to compute.
> > >
> > > Finally, as we explained in the Introduction section and re-emphasized in 1. Motivation of our rebuttal [here](https://openreview.net/forum?id=4dAgG8ma3B&noteId=UrUPt5fBhg), the **innovation** is to introduce **quantum chemistry for diffusion guidance**, instead of gradient estimation:
> > >
> > > “for molecule optimization, we aim to achieve controllable generation at inference time with quantum chemistry software to bypass the requirement to train a neural property predictor or a conditional diffusion model that needs labels.” and we use gradient estimation “to overcome the difficulty that we can not backpropagate through the chemistry software.”
> > >
> > > [1] Wu, L., Trippe, B., Naesseth, C., Blei, D. and Cunningham, J.P., 2024. Practical and asymptotically exact conditional sampling in diffusion models. Advances in Neural Information Processing Systems, 36.

---

### Official Review · Reviewer_vmGX · 2024-10-31

**Soundness:** 2
**Presentation:** 2
**Contribution:** 2
**Rating:** 6
**Confidence:** 3

**Summary:**

The authors propose CHEMGUIDE, a sampling algorithm to guide pre-trained (latent) diffusion models for molecule design with the goal to optimize for stable molecules indicated by smaller force norms when evaluated using the xTB oracle functions. As the xTB oracle function, which outputs the forces per atom in a molecule, is non-differentiable, the authors make use of known gradient approximation from random pertubation theory to approximate the gradients suitable for guidance during the diffusion sampling trajectory. The authors also suggest how their non-differentiable guidance can be combined with neural regressors as commonly done in the diffusion models literature. The authors show that they non-differentiable guidance when applied on GeoLDM leads to generated molecules with lower force norms compared to the samples when GeoLDM is used without the proposed guidance, indicating that their method works for the models trained on common benchmark datasets such as QM9 and GEOM-Drugs.

**Strengths:**

The usage of approximate gradients from non-differentiable oracles in diffusion guidance for molecule design is novel and interesting.
To evaluate their proposed method, the authors run a suite of multiple experiments showing that on the two common benchmarks the generated samples based on their sampling algorithm have improved evaluation metrics compared to the baselines.

**Weaknesses:**

The guidance component from the non-differentiable oracle shows small improvement on the QM9 and GEOM-Drugs dataset. While the idea is interesting, a stronger baseline to compare against is to use the samples generated from Baseline GeomLDM and perform a relaxation using xTB. As the authors mention in their appendix A - Implementation Details, the sampling time for 100 molecules is quite slow with 6 hours and 18 minutes if they perform their proposed guidance in the last 400 diffusion timesteps using xTB as oracle.
How does the GeoLDM baseline (right column) in Table 1 and 2 compare if xTB is used using a pre-defined number of relaxation steps?

Furthermore, I found it hard to read the paper as Section 3 Methodology contains sections 3.1 and 3.2 which are not the author's contribution but already existing methods. I would move these Section 2 within the preliminaries.

**Questions:**

Is there a typo in the numerator in Eq. 15 when approximating the gradient? It should say $\frac{ \mathcal{F}[z_{x,t} + c U, z_{h,t}] - \mathcal{F}[z_{x,t} - c U, z_{h,t}] )}{2c}$.

In Algorithm 1, the approximated gradient g_{t-1} has a dependency to the state at time $t=0$. Is this is a typo, since Eq. 15 does not refer to the clean data prediction.

---

> ### Author Response · Authors · 2024-11-23
>
> Thank you for taking the time to review! We appreciate your feedback and address the weaknesses and questions as follows.
>
> ### **1. Relaxation with xTB**
>
> - **Comparing to GeoLDM with xTB relaxation** We have included this baseline in tables 1-3 by reporting “energy above ground state”, which is defined in Appendix B as “ the difference between the energy before and after using xTB or DFT (specified in the context) to optimize the geometry of the generated molecules”. We can observe that **ChemGuide consistently achieves the smallest energy difference before and after quantum chemistry optimization** at different levels (xTB, DFT) across datasets (QM9, GEOM).
>
> - **Relaxation inside xTB** The number of geometric optimization steps is decided by xTB given the specific input molecule, which is a built-in and required process that can not be reduced or relaxed to achieve speed-up.
>
> - **Relaxation on the guidance steps of ChemGuide** There are two ways to accelerate ChemGuide, 1. add guidance for every $n$ step; 2. add less consecutive guidance steps.
>
>   - **Every $n$ step guidance**, the results are in **Appendix H1** (previously Appendix F1), where we choose $n=1, 3, 5$. We can observe that **a**. $n=1$ (no relaxation) performs the best, suggesting a trade-off between time and performance, and **b**. ChemGuide with $n=3, 5$ skip steps performs better than the baselines with smaller forces and energy above the ground state, demonstrating the effectiveness of our method.
>
>   - **Less consecutive guidance steps** In the global official comment above, we conduct experiments that apply the guidance for 100, 200, and 300 steps to generate 500 molecules, and report the results (updated in **Appendix H2** in our current revised submission). Similar trends can be observed that reducing guidance steps lowers performance.
>
> - **Relaxation on replacing xTB with neural networks**  In the global official comment in Table 3-8 above, we also relax the dependency on xTB and replace xTB with a neural network AIMNet2 [1], and report the results (detailed results and analysis are in H.3 of our current revised submission). We can observe that using a neural network as guidance hurts the performance, and further, significantly increases GPU memory demands.
>
> - **Hardware level relaxation**, we can increase the number of CPU cores assigned to xTB when calculating force, and this functionality is implemented in our code.
>
> ### **2. Order of Section 3.1**
>
> We changed the order accordingly and moved Section 3.1 on noisy and clean guidance into Section 2 Preliminary. Thank you for your constructive suggestion to present our paper clearly!
>
> ### **3. Eq. 15 typo**
> We fixed the typo accordingly. Thank you for your constructive suggestion to present our paper clearly!
>
> ### **Reference**
> [1] Dylan Anstine, Roman Zubatyuk, and Olexandr Isayev. Aimnet2: a neural network potential to
> meet your neutral, charged, organic, and elemental-organic needs. 2024.

---

> > ### Author Response · Authors · 2024-11-23
> >
> > We hope the above information helps address your concerns and questions. We are happy to answer any future questions you may have. Thank you!
> >
> > We also included additional results and summaries in <https://openreview.net/forum?id=4dAgG8ma3B&noteId=UrUPt5fBhg>.

---

> > > ### Comment · Reviewer_vmGX · 2024-11-25
> > >
> > > Thank you for addressing my concerns in your revised manuscript. I acknowledge the improvements made, particularly in adding new experiments in the appendix. I appreciate your efforts and will increase my score to 6.

---

> > > > ### Author Response · Authors · 2024-12-04
> > > >
> > > > Thank you very much for your engagement in the discussion and valuable suggestions to make our paper better.

---

### Official Review · Reviewer_V6At · 2024-11-04

**Soundness:** 3
**Presentation:** 3
**Contribution:** 2
**Rating:** 5
**Confidence:** 4

**Summary:**

The paper proposes CHEMGUIDE, a approach that uses non-differentiable guidance for the conditional generation of molecular structures in diffusion models. CHEMGUIDE use quantum chemistry oracles as guidance, providing gradients for sampling distributions. The method applies zeroth-order optimization to enhance molecule stability by minimizing atomic forces. Experiments demonstrate CHEMGUIDE’s effectiveness on two datasets.

**Strengths:**

1. CHEMGUIDE’s use of quantum chemistry as a non-differentiable oracle in conditional diffusion models is meaningful.
2. The paper reports improvements in stability and force metrics over baselines.
3. Implementing zeroth-order optimization in a diffusion context is well-justified.

**Weaknesses:**

1. The paper could enhance its rigor by comparing CHEMGUIDE with more baseline models, such as MolGAN, and more importantly, other existing guidance methods. The current comparisons seem limited. It would also be comprehensive to experiment with more datasets.
2. While GFN2-xTB is a reasonable compromise, comparing CHEMGUIDE results against high-accuracy methods like DFT more extensively could help validate the chemical accuracy of generated molecules.
3. The paper lacks a thorough discussion on the limitations of using a non-differentiable oracle, such as the potential difficulty in handling certain molecular configurations or diverse chemical spaces.
4. The use of the GFN2-xTB method and bilevel optimization adds computational complexity, which could restrict practical usage. And The guidance scale parameter lacks an adaptive mechanism. Exploring automated scale scheduling would improve usability.
5. Code is not provided.

**Questions:**

Please see Weaknesses

---

> ### Author Response · Authors · 2024-11-23
>
> Thank you for taking the time to review and provide feedback! We address the weaknesses and questions as follows.
>
> ### **1. More baselines, guidance methods & datasets**
> #### - **Additional baselines**
> Our ChemGuide model aims to generate **stable 3D** molecules (atoms + coordinates) with geometries close to their ground states. In our model, a generated 3D molecule is represented by its atoms and their coordinates. In comparison, MolGAN [1] was designed to generate **valid 2D** molecules with correct valency requirements. In MolGAN, a molecule is represented by a graph, with atoms as the vertices and bonds as the edges. **For molecule stability, it can be only defined on 3D structures over 2D graphs**.
>
> While MolGAN shows high potential in generating valid 2D molecules, generating **valid** 3D molecules is inherently more challenging. On the other hand, we are also targeting generating **stable** 3D molecules. To start, a valid 3D molecule must also meet the requirements for stable 2D molecules - correct valencies. In the 3D scenario, a bond is inferred from the generated coordinates by quantum mechanics. The model needs to learn the underlying quantum chemistry rules to generate a valid 3D molecule, such as the correct distance for desired bonds, the correct dihedral angles for correct configurations, etc.
>
> We designed our model to generate 3D molecules that are chemically correct but also thermodynamically stable with close-to-ground-state geometries. Additionally, we tasked our model to perform conditional generation of 3D molecules with desired quantum chemical properties such as HOMO-LUMO gap and polarizability - these properties are also more intricate than the properties used in MolGAN, such as drug likeliness and solubility.
>
> From the above considerations, MolGAN cannot be used as a 3D benchmark, but we are happy to consider other 3D benchmarks if more related models can be suggested. For a more robust evaluation of ChemGuide, we selected EDM [2] and GeoEDM [3] as the baseline models - they are most relevant to our work in generating valid 3D molecules (with conditioned properties) and their results were the recent state of the art. Our stability criteria are a proper superset of the stability definitions in the baselines and thus more robust and strict (see tables 1-3). We also conducted a comprehensive comparison of conditional 3D molecule generation (see tables 4-6).
>
> #### - **Other existing guidance methods**
> In the [global official comment](https://openreview.net/forum?id=4dAgG8ma3B&noteId=UrUPt5fBhg) above, we provide more guidance results for stability optimization with **1**. neural networks for force prediction (**Appendix H3**) and **2**. evolutionary algorithms (**Appendix I**). We can observe that ChemGuide performs the best compared to different guidance methods, with mostly decreased energy above ground state percentages, which demonstrates the effectiveness of our method.
>
> We would also like to note that ChemGuide is to achieve guidance in the form of gradients **directly** from quantum chemistry, there is no equivalent counterpart to strictly compare with, since neural networks are surrogate models for force prediction (trained on the labels produced by quantum chemistry) and evolutionary algorithms incur uncontrollable randomness during the evolution process.
>
> #### - **Additional datasets**
> In our results, we presented the results with the QM9 dataset and the GEOM dataset. They are two of the most widely recognized large datasets in the community and they are large: QM9 has 134K ground-state molecules and GEOM has 37M conformational molecules. Considering the scale of the dataset and the level of quantum chemical theory, QM9 and GEOM are two of the best benchmark datasets in the field, on which existing ML algorithms often struggle. Considering the computational cost, we focused more in-depth on the current two datasets to provide more comprehensive results on them rather than going in breadth across a multitude of datasets.
>
> If the reviewer has recommendations for additional datasets, we would be happy to look into them.

---

> ### Author Response · Authors · 2024-11-23
>
> ### **2. Comparing ChemGuide against DFT**
>
> The results against DFT are provided in Table 3 - for every generated 3D geometry by our model and the baseline models, we used DFT/B3LYP/6-31G(2df,p), the same level of theory as the QM9 dataset, to get its optimized geometry (ground state) and compared their energy difference. We showed that ChemGuide-generated 3D geometries are more stable and closer to the ground-state geometries than the baseline models while also being more successful in generating valid molecules.
>
> In other words, **we were able to achieve improved performance even in DFT accuracy by leveraging a lower-level GFN2-xTB theory with a reasonable cost**.
>
> ### **3. Handle certain molecular configurations**
>
> Our choice of using GFN2-xTB as the non-differentiable oracle would help address the potential difficulties in handling certain molecular configurations such as large molecules. It is a semi-empirical density functional-based tight-binding method with reasonable accuracy and computational cost, and unlike neural networks, does not suffer from out-of-distribution generalization problems (see Figure 7 in Appendix H.4). When a decoded 3D geometry from step $t$ is invalid during the generative process, GFN2-xTB will catch such undesired situations. Since the estimated guidance is not informative in such cases and calculating properties for invalid conformations is theoretically not correct, we add no guidance for the invalid molecules in such cases at step $t$ and allow more steps for the model to improve by itself.
>
> ### **4. Automated scale scheduling**
>
> We noticed the need for automated scale scheduling. However, since our paper mainly focuses on introducing quantum chemistry to the generative process for molecules, we mentioned this in our future work section on page 10 line 530:
>
> “Additionally, we observe that the guidance scale plays a critical role in the generative process of guided diffusion for molecules, but it is generally difficult to determine in advance. This opens the possibility for research into more effective methods for selecting guidance scales, including automated scale scheduling that optimizes guidance strength.”
>
> We expect that automated scale scheduling will continue to improve ChemGuide from the results shown in Tables 1 and 2. This would require extra designing, tuning, and computational costs and is beyond the scope of this work.
>
> ### **5. Code availability**
>
> **We provided access to our code in Appendix A: <https://anonymous.4open.science/r/ChemGuide> in our initial submission**, and added the link in the abstract in our current revised version.
>
> ### **References**
> [1] De Cao, Nicola, and Thomas Kipf. "MolGAN: An implicit generative model for small molecular graphs." arXiv preprint arXiv:1805.11973 (2018).
>
> [2] Hoogeboom, Emiel, et al. "Equivariant diffusion for molecule generation in 3d." International conference on machine learning. PMLR, 2022.
>
> [3] Xu, Minkai, et al. "Geometric latent diffusion models for 3d molecule generation." International Conference on Machine Learning. PMLR, 2023.

---

> ### Author Response · Authors · 2024-11-23
>
> We hope the above information helps address your concerns and questions. We are happy to answer any future questions you may have. Thank you!
>
> We included additional results and summaries in <https://openreview.net/forum?id=4dAgG8ma3B&noteId=UrUPt5fBhg>.

---

> > ### Comment · Reviewer_V6At · 2024-12-02
> > **Response to rebuttal**
> >
> > I appreciate the rebuttal from the authors and the experiments extended. However, the guidance baselines are not comprehensive.  Works such as [1,2] and those brought up by Reviewer tzoc are related and can be compared. I will keep the score due to this and understandings of the issues other reviewers discussed.
> >
> > [1] Amortizing intractable inference in diffusion models for vision, language, and control
> >
> > [2] Steering Masked Discrete Diffusion Models via Discrete Denoising Posterior Prediction

---

> > > ### Author Response · Authors · 2024-12-04
> > >
> > > Thank you for your response.
> > > -  While [1, 2] can accommodate non-differentiable reward functions, their methods are fundamentally different from ours. As described in Algorithm 1 of [2] and Section 4.2 of [1], these methods use the reward function to **fine-tune** their pre-trained models. In contrast, our approach is **training-free**, as we only apply xTB during inference, as highlighted in our introduction. This makes our method both simpler and more computationally efficient.
> > >
> > > - **Furthermore, since [2] comes after the ICLR deadline, it's impossible for us to know it in advance**.
> > >
> > > - For the questions raised by review tzoc, regarding more baselines, such as tSMC, to compare, please refer to [here](https://openreview.net/forum?id=4dAgG8ma3B&noteId=iMspT6LkPn). In summary, tSMC is different from ours because it requires gradient calculation.
> > >
> > > [1] Amortizing Intractable Inference in Diffusion Models for Vision, Language, and Control
> > >
> > > [2] Steering Masked Discrete Diffusion Models via Discrete Denoising Posterior Prediction

---

### Author Response · Authors · 2024-11-23

We thank all reviewers for taking the time to review and for their constructive feedback! We appreciate their effort and suggestions to make our paper a better work! We summarize the changes of our current revised paper (in blue text) as follows.

- **List of content (Appendix A)**: for convenience of reading Appendix, we added a list of content to Appendix A.
- **More experiments**:
    - **Effect of guidance steps (Appendix H2)**: we explored reducing guidance steps from 400 to 100, 200, and 300 in Appendix H.2.
    - **AIMNet2 (Appendix H3)**: we replaced the non-differentiable oracle xTB with a differentiable neural network potential AIMNet2 for guiding force optimization, and compared the gradients from xTB and AIMNet2 by plotting their cosine similarities.
    - **Other optimization algorithms (Appendix I)**: we used an evolutionary algorithm for force optimization in Appendix I.


We would like to summarize our motivations as follows.
### **1. Motivation**
for molecule optimization, we aim to achieve controllable generation at inference time with quantum chemistry software to bypass the requirement to train a neural property predictor or a conditional diffusion model that needs labels.

We choose stability (defined by 3D molecule structure) as the molecule property to explore, hence using a 3D diffusion backbone.

Our method builds on previously proposed guided diffusion methods, and the use of zeroth-order optimization techniques for gradient estimation is employed to overcome the difficulty that we can not backpropagate through the chemistry software.

### **2. Importance of stability optimization**
- To generate **3D** molecules (atoms and coordinates) that are valid and **stable** with close-to-ground-state geometries.
  - We introduced a more strict, robust, and challenging stability metric, energy above ground state, than the current mainstream literature metric with chemical valencies.
- To generate **stable** 3D molecules of **conditioned quantum chemical properties**.
- The motivation behind generating stable close-to-ground-state molecules is that molecular properties are in many cases geometry-related. And ground state geometries are important.
  - Take the example in Appendix D - say we need to generate 3D molecules that are non-polar, a generative model can very well generate a line-shaped H-O-H water molecule, which is non-polar and correct in valencies. But a line-shaped water molecule is not considered stable and thus this generated 3D molecule can be misleading in real-life applications. Our work aims to avoid such scenarios and generate **stable** molecules by introducing and estimating quantum mechanical (QM) guidance.

---

> ### Author Response · Authors · 2024-11-23
>
> We address common weaknesses and questions as follows.
>
> ### **1. More baselines**
> #### **- AIMNet2**
>
> we replaced xTB with the neural network potential AIMNet2 for guiding force optimization. Detailed results and analysis are presented in **Appendix H.3**. We evaluated its performance by sampling 500 molecules from QM9 and 50 molecules from GEOM. We found two significant drawbacks of using neural networks as guidance:
>
> **Performance**: Using AIMNet2 for guidance negatively impacted performance. Specifically, for QM9, ChemGuide consistently outperforms AIMNet2 across all metrics. For GEOM, while AIMNet2 achieves similar results to ChemGuide in terms of force, validity, and stability, it struggles with the energy above the ground state. This indicates the optimization process is prone to being trapped in local minima, far from the global minima.
>
> To understand the difference, we analyzed the cosine similarity between the guidance provided by ChemGuide and AIMNet2. The results showed that their gradients are nearly orthogonal, which accounts for the observed performance differences.
>
> **Memory Constraints**: AIMNet2 outputs forces in the shape [N, 3]. Backpropagating through these forces to compute guidance requires the Hessian matrix, resulting in a memory complexity of [N, 3, N, 3]. This complexity increases by a factor of $3N$ which significantly limits scalability. For GEOM, we could only sample 5 molecules at a time on a 48GiB GPU. In contrast, ChemGuide allowed us to sample 100 molecules per batch on the same device.
>
> We report the performances below. * and **bold** denote the overall best result and our best result, respectively. Percentage changes between our results and GeoLDM are shown in parentheses.
>
> Table 1: AIMNet neural force guidance on 500 molecules sampled on QM9
>
> | Metric          	| 0.0001           	| 0.001            	| 0.01             	| 0.1              	| 1.0               	| ChemGuide	| EDM    	| GeoLDM 	|
> |---------------------|----------------------|----------------------|----------------------|----------------------|-----------------------|---------------------|------------|------------|
> | **Force RMS (Eh/Bohr)**   	| 0.0113 	         | 0.0113          	| 0.0112          	| 0.0114          	| 0.0110 (-1.00%↓) 	| **0.0104*** (-6.76%↓) | 0.0114 	| 0.0111 	|
> | **Validity**    	| 89.20% (-0.60%↓)	| 89.20% (-0.60%↓)	| 89.00%          	| 88.60%          	| 89.00%           	| **91.40%*** (1.60%↑)  | 86.60% 	| 89.80% 	|
> | **Uniqueness**  	| **100.00%***    	| **100.00%***    	| **100.00%***    	| **100.00%***    	| **100.00%***     	| **100.00%***     	| 100.00%* | 100.00%* |
> | **Atom Stability**  | 98.72%          	| 98.72%          	| 98.71%          	| 98.74%     	     | 98.75% (-0.18%↓) 	| **99.02%*** (0.09%↑)  | 98.53% 	| 98.93% 	|
> | **Molecule Stability** | 87.80%          	| 87.80%          	| 87.60%          	| 87.80%          	| 88.00% (-0.80%↓) 	| **90.60%*** (1.80%↑)  | 83.60% 	| 88.80% 	|
> | **Energy above ground state (Eh)**   | 0.0056         	| 0.0057         	| 0.0056         	| 0.0059         	| 0.0055 (9.30%↑) 	| **0.0042*** (-15.78%↓) | 0.0072 	| 0.0050	|
>
> Table 2: AIMNet neural force guidance on 50 molecules sampled on GEOM
>
> | Metric                    	| 0.0001          	| 0.001           	| 0.01            	| 0.1             	| 1.0              	| ChemGuide    	| EDM    	| GeoLDM 	|
> |-------------------------------|---------------------|---------------------|---------------------|---------------------|----------------------|---------------------|------------|------------|
> | **Force RMS (Eh/Bohr)**   	| 0.0418         	| 0.0418         	| 0.0418         	| 0.0409         	| **0.0406*** (-15.14%↓) | 0.0411 (-14.16%↓)  | 0.0742 	| 0.0478 	|
> | **Validity (xTB)**        	| 50.00% (1.00%↑)	| 50.00%         	| 50.00%***      	| 50.00%         	| 50.00%          	| **50.40%*** (1.40%↑) | 46.40% 	| 49.00% 	|
> | **Uniqueness**            	| **100.00%***   	| **100.00%***   	| **100.00%***   	| **100.00%***   	| **100.00%***    	| **100.00%***    	| 100.00%* | 100.00%* |
> | **Atom Stability**        	| 85.01%         	| 85.01%         	| 85.01%         	| 85.13%         	| **85.26%*** (0.73%↑) | 84.36% (-0.17%↓)	| 81.22% 	| 84.53% 	|
> | **Energy above ground state (Eh)** | 0.2392        	| 0.2392        	| 0.2392        	| 0.2356        	| 0.2299 (2.26%↑)	| **0.1935*** (-13.92%↓) | 0.3742 	| 0.2248	|

---

> ### Author Response · Authors · 2024-11-23
>
> #### - **Evolutionary algorithm**
>
> We explore using evolutionary algorithms as optimization. Details can be found in **Appendix I**. We follow [1], and the parameters of the algorithm are variant size $k$ and evolution interval $E$. During the diffusion process, the population is preserved, and evolution is performed at fixed intervals. At each evolution step, $k-1$ noise perturbations are added to the population, resulting in $k$ variants (we treat the non-perturbed molecule as a variant). The best variant is then selected as the new population based on evaluations from the non-differentiable oracle (i.e. xTB). Specifically, the oracle calculates the force RMS for each variant, and the variant with the lowest force RMS is selected. We explore several combinations of variant size and evolution interval in the below table. ChemGuide significantly outperforms it in terms of force RMS and energy above the ground state.
>
> Table 3: Evolutionary algorithm results of 500 molecules on QM9
>
> | Metric                    	| ($k=3$, $E=20$)       	| ($k=3$, $E=50$)       	| ($k=5$, $E=20$)       	| ($k=5$, $E=50$)      	| ChemGuide   	| EDM    	| GeoLDM 	|
> |-------------------------------|-------------------|-------------------|-------------------|--------------------|----------------------|------------|------------|
> | **Force RMS (Eh/Bohr)**   	| 0.0112       	| 0.0111       	| 0.0110       	| 0.0109 (-1.75%↓)  | **0.0104*** (-6.76%↓) | 0.0114 	| 0.0111 	|
> | **Validity**              	| 91.60%       	| 87.60%       	| 90.00%       	| **92.40%*** (2.60%↑) | 91.40% (1.60%↑)   	| 86.60% 	| 89.80% 	|
> | **Uniqueness**            	| **100.00%*** 	| **100.00%*** 	| 99.78%       	| **100.00%***   	| **100.00%***     	| 100.00%* | 100.00%* |
> | **Atom Stability**        	| 98.95%       	| 98.76%       	| 98.66%       	| **99.14%*** (0.21%↑) | 99.02% (0.09%↑)   	| 98.53% 	| 98.93% 	|
> | **Molecule Stability**        | 90.00%       	| 87.60%       	| 88.00%       	| **91.20%*** (2.40%↑) | 90.60% (1.80%↑)   	| 83.60% 	| 88.80% 	|
> | **Energy above ground state (Eh)** | 0.0048 (-4.45%↓) | 0.0053      	| 0.0056      	| 0.0054       	| **0.0042*** (-15.78%↓) | 0.0072 	| 0.0050	|
>
> ### **Reference**
> [1] Huang, Y., Ghatare, A., Liu, Y., Hu, Z., Zhang, Q., Sastry, C.S., Gururani, S., Oore, S. and Yue, Y., Symbolic Music Generation with Non-Differentiable Rule Guided Diffusion. In Forty-first International Conference on Machine Learning, 2024.

---

> > ### Author Response · Authors · 2024-11-23
> >
> > ### **2. Relaxation of xTB: Fewer guidance steps**
> > - **Non-consecutive guidance**: this is called skip-step, as an acceleration method, where we add guidance every $N$ steps. Details can be found in **Appendix H1** (previously F1). We explore adding guidance every 3 and 5 steps. It turns out that in general more guidance steps, better results. The results are reported in below table:
> >
> > Table 4 xTB force guidance using skip-step acceleration on 500 molecules sampled on QM9
> >
> > | Metric | Every 1 Step (s) | Every 3 Steps (s) | Every 5 Steps (s) | EDM | GeoLDM |
> > |-------------------------------|--------------------|---------------------|---------------------|------------|------------|
> > | **Force RMS (Eh/Bohr)** | **0.0104*** (-6.76%↓) | 0.0106 | 0.0011 | 0.0114 | 0.0111 |
> > | **Validity** | **91.40%*** (1.60%↑) | 91.40% | 89.40% | 86.60% | 89.80% |
> > | **Uniqueness** | **100.00%*** | 99.79% | 100.00% | 100.00% | 100.00% |
> > | **Atom Stability** | 99.02% | 99.01% | 98.63% | 98.53% | 98.93% |
> >  | **Molecule Stability** | 90.60% | 90.60% | 87.20% | 83.60% | 88.80% |
> >  | **Energy above ground state (Eh)** | **0.0042*** (-15.78%↓) | 0.0045 | 0.0055 | 0.0072 | 0.0050|
> >
> > - **Consecutive yet delayed guidance**: instead of adding 400 guidance steps, we explore adding 100, 200, and 300 guidance steps. Details can be found in **Appendix H.2**. It turns out that in general more guidance steps, better results. The results are reported in below table:
> >
> > Table 5 xTB force guidance using 100 guidance steps on 500 molecules sampled on QM9
> >
> > | Metric                    	| 0.0001           	| 0.001            	| 0.01             	| 0.1              	| 1.0              	| EDM    	| GeoLDM 	|
> > |-------------------------------|----------------------|----------------------|----------------------|----------------------|----------------------|------------|------------|
> > | **Force RMS (Eh/Bohr)**   	| **0.0110*** (-0.88%↓) | **0.0110*** (-0.88%↓) | 0.0110          	| 0.0113          	| 0.0120          	| 0.0114 	| 0.0111 	|
> > | **Validity**              	| 90.40%          	| 90.40%          	| 90.40%          	| **90.80%*** (1.00%↑) | 90.60%          	| 86.60% 	| 89.80% 	|
> > | **Uniqueness**            	| **100.00%***    	| **100.00%***    	| **100.00%***    	| **100.00%***    	| **100.00%***    	| 100.00%* | 100.00%* |
> > | **Atom Stability**        	| **98.94%*** (0.01%↑) | **98.94%*** (0.01%↑) | **98.94%*** (0.01%↑) | 98.91%          	| 98.92%          	| 98.53% 	| 98.93% 	|
> > | **Molecule Stability**    	| 89.40%          	| 89.40%          	| 89.40%          	| **89.60%*** (0.80%↑) | **89.60%*** (0.80%↑) | 83.60% 	| 88.80% 	|
> > | **Energy above ground state (Eh)** | 0.0051         	| **0.0051*** (0.87%↑) | 0.0051         	| 0.0055         	| 0.0057         	| 0.0072 	| 0.0050	|

---

> > > ### Author Response · Authors · 2024-11-23
> > >
> > > Table 6 xTB force guidance using 200 guidance steps on 500 molecules sampled on QM9
> > >
> > > | Metric                    	| 0.0001           	| 0.001            	| 0.01             	| 0.1              	| 1.0              	| EDM    	| GeoLDM 	|
> > > |-------------------------------|----------------------|----------------------|----------------------|----------------------|----------------------|------------|------------|
> > > | **Force RMS (Eh/Bohr)**   	| 0.0119          	| **0.0106*** (-4.28%↓) | 0.0119          	| 0.0114          	| 0.0116          	| 0.0114 	| 0.0111 	|
> > > | **Validity**              	| 90.00%          	| **91.00%*** (1.20%↑) | 90.00%          	| 90.60%          	| 90.60%          	| 86.60% 	| 89.80% 	|
> > > | **Uniqueness**            	| **100.00%***    	| **100.00%***    	| **100.00%***    	| **100.00%***    	| **100.00%***    	| 100.00%* | 100.00%* |
> > > | **Atom Stability**    	    | 98.87%          	| 98.72%          	| 98.87%          	| **98.90%*** (-0.03%↓) | 98.86%          	| 98.53% 	| 98.93% 	|
> > > | **Molecule Stability**    	| 89.00%          	| 88.40%          	| 89.00%          	| **89.40%*** (0.60%↑) | 89.00%          	| 83.60% 	| 88.80% 	|
> > > | **Energy above ground state (Eh)** | 0.0054         	| **0.0045*** (-9.32%↓) | 0.0054         	| 0.0049         	| 0.0050         	| 0.0072 	| 0.0050	|
> > >
> > > Table 7 xTB force guidance using 300 guidance steps on 500 molecules sampled on QM9
> > >
> > > | Metric                    	| 0.0001           	| 0.001            	| 0.01             	| 0.1              	| 1.0              	| EDM    	| GeoLDM 	|
> > > |-------------------------------|----------------------|----------------------|----------------------|----------------------|----------------------|------------|------------|
> > > | **Force RMS (Eh/Bohr)**   	| 0.0109          	| 0.0109          	| **0.0107*** (-3.98%↓) | 0.0108          	| 0.0115          	| 0.0114 	| 0.0111 	|
> > > | **Validity**              	| **92.80%*** (3.00%↑) | **92.80%*** (3.00%↑) | 92.60%          	| 92.20%          	| 88.80%          	| 86.60% 	| 89.80% 	|
> > > | **Uniqueness**            	| 99.57%          	| **100.00%***    	| 99.58%          	| 99.36%          	| **100.00%***    	| 100.00%* | 100.00%* |
> > > | **Atom Stability**        	| 98.93%          	| **99.09%*** (0.15%↑) | 99.00%          	| 98.99%          	| 98.88%    	      | 98.53% 	| 98.93% 	|
> > > | **Molecule Stability**    	| 89.80%          	| **90.80%*** (2.00%↑) | 89.60%          	| 89.60%          	| 88.00%          	| 83.60% 	| 88.80% 	|
> > > | **Energy above ground state (Eh)** | 0.0049         	| 0.0052         	| **0.0045*** (-10.64%↓) | 0.0046         	| 0.0056         	| 0.0072 	| 0.0050	|
> > >
> > >
> > > Table 8 xTB force guidance using 400 guidance steps on 500 molecules sampled on QM9
> > >
> > > | Metric                    	| 0.0001           	| 0.001            	| 0.01             	| 0.1              	| 1.0              	| EDM    	| GeoLDM 	|
> > > |-------------------------------|----------------------|----------------------|----------------------|----------------------|----------------------|------------|------------|
> > > | **Force RMS (Eh/Bohr)**   	| **0.0104*** (-6.76%↓) | 0.0104          	| 0.0107          	| 0.0108          	| 0.0125          	| 0.0114 	| 0.0111 	|
> > > | **Validity**              	| **91.40%*** (1.60%↑) | 91.20%          	| 91.20%    	      | 90.00%          	| 89.40%          	| 86.60% 	| 89.80% 	|
> > > | **Uniqueness**            	| **100.00%***    	| **100.00%***    	| **100.00%***    	| **100.00%***    	| **100.00%***    	| 100.00%* | 100.00%* |
> > > | **Atom Stability**        	| 99.02%          	| 98.97%          	| 98.98%          	| **99.03%*** (0.10%↑) | 98.67%          	| 98.53% 	| 98.93% 	|
> > > | **Molecule Stability**    	| 90.60%          	| 90.40%          	| 90.20%          	| **91.00%*** (2.20%↑) | 87.80%          	| 83.60% 	| 88.80% 	|
> > > | **Energy above Ground State (Eh)** | **0.0042*** (-15.78%↓) | 0.0042          	| 0.0045          	| 0.0050          	| 0.0061          	| 0.0072 	| 0.0050     |

---

### Meta-Review · Area_Chair_aer3 · 2024-12-20

**Metareview:**

The paper proposes a novel guidance technique for molecule generation. Namely, it targets generating molecules closer to their equilibrium state (with lower force magnitude) by adding the proposed guidance term to the diffusion model. The main challenge of this problem is that the objective function (evaluating the force) is not differentiable. The authors propose to address this challenge by evaluating the finite difference of the objective along a random direction. Reviewers tzoc and V6At found the empirical study insufficient but still up to the ICLR standards. On the contrary, Reviewer wXP8 extensively argued for the acceptance of the paper.

**Additional Comments On Reviewer Discussion:**

Several reviewers raised concerns regarding evaluation, e.g. comparison with evolutionary algorithms (as suggested by Reviewer tzoc). The authors provided the requested experiments during the rebuttal which partially addressed the reviewers' concerns. Finally, Reviewer wXP8 championed the acceptance of the paper in the final discussion, which is an important indicator of the paper's relevance to the ICLR community.

---

### Decision · Program_Chairs · 2025-01-22

Accept (Poster)